# PRISM: Gauge-Invariant Tangent-Space Differentially Private LoRA

**Shihao Wang**[1]  **Xueru Zhang**[1]

## Abstract

Applying differential privacy (DP) via DP-SGD to Low-Rank Adaptation (LoRA) is a natural approach for privacy-preserving fine-tuning. However, LoRA's low-rank parameterization poses a fundamental challenge. In LoRA, each trainable update is represented as a low-rank matrix $Z = AB^\top$, but this factorization is inherently *non-identifiable*: many factor pairs $(A, B)$ represent the same update $Z$. As a result, applying DP-SGD directly to the factors induces *gauge-dependent* perturbations on $Z$, and we show that this naive DP-LoRA can lead to unbounded noise amplification. We propose **PRISM**, an intrinsic DP mechanism for LoRA that is gauge invariant by construction, avoids bilinear noise amplification, and admits an efficient low-dimensional noise sampler. Moreover, PRISM yields a closed-form characterization of the effective intrinsic noise induced on $Z$, enabling stable privacy–utility trade-offs through bounded, gauge-invariant perturbations. We establish standard $(\varepsilon, \delta)$-DP guarantees for PRISM and introduce a DP-aware, gauge-invariant adaptive update rule that prevents adaptive optimization from amplifying injected privacy noise, improving numerical stability in practice.

## 1. Introduction

Foundation models are routinely adapted to domain-specific tasks using *private* corpora. Because full-model fine-tuning is costly, practitioners often adopt parameter-efficient fine-tuning (PEFT) methods, which update only a small subset of parameters while keeping the backbone frozen. Common methods include adapters (Houlsby et al., 2019; Hu et al., 2023), prefix/prompt tuning (Li & Liang, 2021; Lester et al., 2021), bias-only updates (Ben Zaken et al., 2022), and

lightweight reparameterizations (Liu et al., 2022). Among these, Low-Rank Adaptation (LoRA) (Hu et al., 2022) has emerged as a dominant method due to its drop-in compatibility with existing linear layers, strong performance under low-precision training with a quantized backbone (Dettmers et al., 2023; Li et al., 2024), and the non-identifiability induced by its low-rank parameterization. Formally, LoRA adapts a frozen pretrained weight matrix $W_0 \in \mathbb{R}^{m \times n}$ by adding a low-rank update $Z$:

$$W = W_0 + Z, \qquad Z = AB^\top \tag{1}$$

where $A \in \mathbb{R}^{m \times r}$, $B \in \mathbb{R}^{n \times r}$ with rank $r \ll \min\{m, n\}$. Rather than updating $W_0$ directly, optimization proceeds over $(A, B)$, which implicitly define the intrinsic update $Z$ applied to the backbone. This low-rank factorization substantially reduces the number of trainable parameters.

Despite its efficiency, PEFT on sensitive data raises significant privacy concerns. Prior work has shown that trained models can leak information about training data through membership, inversion, or extraction attacks (Fredrikson et al., 2015; Shokri et al., 2017; Ganju et al., 2018; Carlini et al., 2019; 2021). These risks are often exacerbated in PEFT settings, where fine-tuning datasets are small, domain-specific, and contain rare or uniquely identifying records.

Various approaches have been proposed to mitigate privacy risks in large models (Bourtoule et al., 2021). We focus on differential privacy (DP) (Dwork et al., 2006; Dwork & Roth, 2014), which provides an attack-agnostic, *ex ante* guarantee by bounding the influence of any individual record and remaining robust to arbitrary post-processing. In practice, DP is most commonly instantiated via DP-SGD (Abadi et al., 2016), which clips per-example gradients and injects calibrated Gaussian noise before each update.

A natural approach to obtaining DP in LoRA fine-tuning is to apply DP-SGD directly to the low-rank factors $(A, B)$ (Yu et al., 2022; Liu et al., 2025; Xu et al., 2025). Some variants further freeze one of the two factors to improve numerical stability (Sun et al., 2024). While straightforward, this strategy is fundamentally misaligned with the structure of LoRA and leads to ill-defined private updates.

The core issue is that DP-SGD is defined relative to a *parameterization*, whereas in LoRA the factors $(A, B)$ are only an auxiliary representation of the intrinsic update applied to

---

[1]Department of Computer Science and Engineering, The Ohio State University, Columbus, OH, USA. Correspondence to: Shihao Wang <wang.17571@osu.edu>, Xueru Zhang <zhang.12807@osu.edu>.

*Proceedings of the 43rd International Conference on Machine Learning*, Seoul, South Korea. PMLR 306, 2026. Copyright 2026 by the author(s).

*Table 1.* **Comparison with other DP-LoRA design choices.** We evaluate each variant against three desiderata: (a) gauge-invariant randomized mechanism, (b) additive perturbations on $Z$ (i.e., no bilinear DP term), and (c) LoRA-scale efficiency. The naive factor-space variant violates (a) and (b), yielding unbounded effective intrinsic noise $\mathcal{E}_Z$. The *one-sided* variant, which updates $B$ only while freezing $A$, satisfies (b) and (c) but not (a). PRISM satisfies all three and admits a closed-form expression for $\mathcal{E}_Z$.

| METHOD | PARAMS | $\mathcal{E}_Z$ | (a) | (b) | (c) |
|---|---|---|---|---|---|
| DP-LoRA | $(m+n)r$ | UNBOUNDED | ✗ | ✗ | ✓ |
| ONE-SIDE | $nr$ | $(\sigma C/b)\sqrt{n}\,\|A\|_F$ | ✗ | ✓ | ✓ |
| **PRISM** | $(m+n)r$ | $(\sigma C/b)\sqrt{r(m+n-r)}$ | ✓ | ✓ | ✓ |

the frozen backbone. It is the effective update $Z = AB^\top$, rather than the factors themselves, that ultimately determines model behavior. As a consequence, naively applying DP-SGD in factor space (i) induces *gauge-dependent* perturbations on the intrinsic update $Z$, meaning that different factor pairs corresponding to the same intrinsic update can induce significantly different clipping and noise effects; (ii) introduces spurious higher-order noise terms caused by independently noising the two factors, resulting in quadratic noise effects that do not arise in standard DP-SGD on linear parameters; and (iii) interacts poorly with optimization dynamics, where adaptive preconditioning can amplify stochasticity and lead to numerical instability. We discuss these issues in detail in Section 2.

These issues motivate a key design principle: the randomized DP mechanism should operate on the space of *intrinsic* model updates, rather than on a gauge-redundant factorization. Based on this, we propose the **P**rojected **R**iemannian **I**nvariant **S**ubspace **M**echanism (**PRISM**), which performs DP-SGD directly on the rank-$r$ manifold of LoRA updates. Specifically, PRISM projects per-example gradients to the tangent space $\Delta Z \in T_Z\mathcal{M}_r$, applies global Frobenius-norm clipping and isotropic tangent Gaussian noise in this intrinsic space, and retracts back to rank $r$. By aligning the DP mechanism with the intrinsic update $Z$, PRISM ensures that the *effective intrinsic noise* on $Z$ is deterministic and independent of the particular factorization $(A, B)$. Moreover, operating in intrinsic coordinates keeps updates additive and avoids the spurious bilinear second-order noise terms induced by independently noising the factors. Importantly, PRISM achieves these guarantees with LoRA-scale computational cost. Table 1 compares PRISM with existing DP-LoRA design choices.

Our contributions can be summarized below.

- We identify issues with factor-space DP-LoRA and show that they can lead to (i) gauge-dependent clipping and noise injection, (ii) unbounded amplification of intrinsic noise, and (iii) spurious bilinear second-order noise terms arising from independently noising the two factors.

- We propose PRISM, a gauge-invariant DP mechanism that projects per-example gradients on rank-$r$ tangent space, applies *global* intrinsic clipping across all LoRA modules, injects isotropic tangent noise using an $O((m + n)r^2)$ sampler, and retracts updates back to rank $r$.

- We develop a DP-aware gauge-invariant adaptive update rule that floors rank-space preconditioners based on the DP noise level, mitigating optimizer-induced noise amplification while preserving LoRA-scale efficiency.

- We provide theoretical guarantees showing that PRISM produces gauge-invariant updates and noise distributions, satisfies $(\varepsilon, \delta)$-DP under subsampled Gaussian accounting, and injects tangent noise with gauge-invariant covariance and energy proportional to $r(m + n - r)$. The project code is available at github.com/osu-srml/PRISM-DP-LoRA.

## 2. Problem Formulation

We study differentially private (DP) parameter-efficient fine-tuning using LoRA. Given a frozen weight matrix $W_0 \in \mathbb{R}^{m \times n}$, LoRA learns a rank-$r$ update $Z$ such that $W = W_0 + Z$ with $Z = AB^\top$ (Eq. (1)), by minimizing the empirical risk over a private dataset $\mathcal{D} = \{x_i\}_{i=1}^N$:

$$\min F(A, B) \triangleq \tfrac{1}{N} \sum_{i=1}^N \ell_i(W_0 + AB^\top).$$

Our goal is to design a randomized training procedure whose released adapter $Z$ satisfies $(\varepsilon, \delta)$-DP (Dwork & Roth, 2014) with respect to $\mathcal{D}$, while preserving the utility of LoRA.

Formally, an algorithm $\mathcal{M}$ satisfies $(\varepsilon, \delta)$-DP if, for any adjacent datasets $\mathcal{D}, \mathcal{D}'$ and any measurable event $\mathcal{S}$,

$$\Pr[\mathcal{M}(\mathcal{D}) \in \mathcal{S}] \leq e^\varepsilon \Pr[\mathcal{M}(\mathcal{D}') \in \mathcal{S}] + \delta, \quad (2)$$

A standard approach for achieving DP in model training is DP-SGD (Abadi et al., 2016). At each iteration, DP-SGD computes per-example gradients $g_i = \nabla \ell_i$, clips them to a prescribed norm $C$ and aggregates them with added Gaussian noise $\xi \sim \mathcal{N}(0, I)$:

$$\tilde{g}_i = \frac{g_i}{\max\{1, \|g_i\|_2/C\}}, \qquad \widehat{g} = \frac{1}{b} \sum_{i=1}^b \tilde{g}_i + \frac{\sigma C}{b} \xi \quad (3)$$

where $b$ denotes the batch size and $\sigma$ is the noise multiplier. The noisy $\widehat{g}$ is then used to perform an optimizer update.

A dominant approach in DP-LoRA applies DP-SGD directly to the factor parameters $(A, B)$ (Yu et al., 2022; Liu et al., 2025; Xu et al., 2025). Specifically, let $g_{A,i}, g_{B,i}$ denote the per-example gradients with respect to $A$ and $B$, respectively. This approach applies (3) to the concatenated gradient $g_i = (g_{A,i}, g_{B,i})$ with $\|g_i\|_2^2 = \|g_{A,i}\|_F^2 + \|g_{B,i}\|_F^2$. Some works further consider variants with one-sided training that updates

only one LoRA factor while freezing the other (Sun et al., 2024). However, enforcing DP on the factors $(A, B)$ is fundamentally misaligned with the effective update $Z = AB^\top$ that governs model behavior, as we detail below.

**Issue I: Factor-space DP violates LoRA gauge symmetry.** LoRA factorization is *non-identifiable* (Hu et al., 2022): for any invertible $R \in \mathrm{GL}(r)$, the factor pairs $(A, B)$ and $(AR, BR^{-\top})$ induce the same intrinsic update $Z$. Under such a gauge transformation $(A, B) \mapsto (AR, BR^{-\top})$, the corresponding per-example gradients transform as

$$g'_{A,i} = g_{A,i} R^{-\top}, \qquad g'_{B,i} = g_{B,i} R.$$

As a result, the clipping norm, and hence the clipping coefficient, used in DP-SGD depend on the particular factorization chosen to represent $Z$. For example, under the simple rescaling gauge $(A, B) \mapsto (cA, c^{-1}B)$,

$$\|g'_{A,i}\|_F^2 + \|g'_{B,i}\|_F^2 \;=\; c^{-2}\|g_{A,i}\|_F^2 + c^2\|g_{B,i}\|_F^2. \quad (4)$$

which can vary arbitrarily with $c$. Consequently, the distribution of the clipped-and-noised update produced in factor space is gauge dependent, and the induced intrinsic update is not determined by $Z$ alone (Appendix A.2[1]).

This gauge dependence propagates from gradient clipping to the resulting update increments. Even when a per-example increment $(\Delta A_i, \Delta B_i)$ represents a fixed intrinsic direction $\Delta Z_i = \Delta A_i B^\top + A \Delta B_i^\top$, the same intrinsic direction admits gauge-related representatives $(\Delta A'_i, \Delta B'_i) = (\Delta A_i R, \Delta B_i R^{-\top})$ for any $R \in \mathrm{GL}(r)$. In general,

$$\|\Delta A'_i\|_F^2 + \|\Delta B'_i\|_F^2 \neq \|\Delta A_i\|_F^2 + \|\Delta B_i\|_F^2, \quad (5)$$

so any mechanism that clips and perturbs based on the Euclidean norms in factor space is inherently gauge dependent.

Formally, let $\Delta Z_{\mathrm{fac}}(A, B)$ denote the intrinsic update induced by a single factor-space DP-SGD step. A gauge-respecting mechanism would require

$$\Delta Z_{\mathrm{fac}}(A, B) \overset{d}{=} \Delta Z_{\mathrm{fac}}(AR, BR^{-\top}), \;\; \forall R \in \mathrm{GL}(r), \quad (6)$$

a condition already violated by Eq. (4). Importantly, this issue is distinct from, and not resolved by, deterministic transformation-invariant optimizers for LoRA (Yen et al., 2025), as DP requires invariance of the randomized clipping and noising procedure itself.

**Issue II: Noising both factors injects bilinear and gauge-amplified intrinsic noise.** When DP noise is injected into both factors, the intrinsic update inevitably contains a second-order noise term. Consider a single update step

---

[1]Throughout the paper, we state the main claims in the main text and defer detailed analysis and proofs to the appendix, with explicit appendix references provided after each claim.

$(A, B) \leftarrow (A, B) + (\Delta A, \Delta B)$, and let $\xi_A, \xi_B$ denote the Gaussian perturbations added by DP-SGD, scaled by the step size $\eta$. The induced intrinsic update then satisfies

$$
\begin{aligned}
Z^+ &= (A + \Delta A + \eta \xi_A)(B + \Delta B + \eta \xi_B)^\top \\
&= Z + \Delta Z + \eta(\xi_A B^\top + A \xi_B^\top) + \eta^2 \xi_A \xi_B^\top. \quad (7)
\end{aligned}
$$

The final term $\eta^2 \xi_A \xi_B^\top$ arises from multiplying independently noised factors and cannot be produced by any additive noise mechanism applied directly to $Z$ (Appendix A.3).

Even if this second-order term is ignored, the first-order intrinsic noise remains problematic, as its magnitude depends on the norms of the factors (see Proposition 2.2, Eq. (8)). Under rescaling gauge $(A, B) \mapsto (cA, c^{-1}B)$, Eq. (8) becomes $\tau^2(mc^{-2}\|B\|_F^2 + nc^2\|A\|_F^2)$, which can grow without bound as $c \to 0$ or $c \to \infty$. One-sided variants that freeze one LoRA factor (Sun et al., 2024) can suppress parts of Eq. (7), but do not eliminate the dependence of intrinsic noise energy on a representation-dependent scale, since the frozen factor still determines $\|A\|_F$ or $\|B\|_F$. This motivates enforcing DP directly in the intrinsic space of $Z$ (or its tangent space), rather than in factor coordinates.

To formalize the scale of randomized perturbations on $Z$, we introduce the following notion.

**Definition 2.1** (Effective intrinsic noise)**.** For a random intrinsic perturbation $\mathcal{N}_Z$, define $\mathcal{E}_Z \triangleq \sqrt{\mathbb{E}\|\mathcal{N}_Z\|_F^2}$.

**Proposition 2.2** (Intrinsic noise energy under factor noising)**.** *Let $\xi_A \in \mathbb{R}^{m \times r}$ and $\xi_B \in \mathbb{R}^{n \times r}$ be independent with i.i.d. entries $\mathcal{N}(0, \tau^2)$, and define $\mathcal{N}_Z = \xi_A B^\top + A \xi_B^\top + \xi_A \xi_B^\top$. Then the first-order term $\mathcal{N}_Z^{(1)} \triangleq \xi_A B^\top + A \xi_B^\top$ satisfies*

$$\mathbb{E}\|\mathcal{N}_Z^{(1)}\|_F^2 = \tau^2\Big(m\|B\|_F^2 + n\|A\|_F^2\Big), \quad (8)$$

*while the bilinear term $\mathcal{N}_Z^{(2)} \triangleq \xi_A \xi_B^\top$ satisfies*

$$\mathbb{E}\|\mathcal{N}_Z^{(2)}\|_F^2 = mnr\,\tau^4. \quad (9)$$

Proposition 2.2 reveals that factor-space noising induces intrinsic noise whose first-order component scales with $\|A\|_F, \|B\|_F$ and whose second-order bilinear component scales as $mnr\,\tau^4$, leading to gauge-dependent amplification and an unavoidable $\eta^2$ noise term.

**Corollary 2.3** (Unbounded gauge amplification)**.** *Fix $Z = AB^\top \neq 0$ and $\tau^2 > 0$. Along the scalar gauge family $(A_c, B_c) = (cA, c^{-1}B)$, the first-order effective intrinsic noise $\mathcal{E}_Z$ defined in Eq. (8) is unbounded over $c > 0$ even though $A_c B_c^\top = Z$ is constant.*

See Appendix A.4-A.5 for derivations. Corollary 2.3 shows that factor-space DP induces gauge-dependent and potentially unbounded intrinsic noise, even when the effective update $Z$ is fixed.

**Issue III: Adaptive preconditioning magnifies DP noise and stresses low-rank numerics.** Private PEFT methods typically rely on adaptive optimizers such as Adam and AdamW to maintain utility under noisy gradients (Kingma & Ba, 2015; Loshchilov & Hutter, 2019), and recent LoRA-specific invariant optimizers extend the same principle (Yen et al., 2025). Conceptually, these methods apply a (possibly low-dimensional) preconditioner to the DP gradient:

$$\theta^+ = \theta - \eta\, \mathsf{P}^{-1/2}\widehat{g}, \qquad \widehat{g} = g + \xi \qquad (10)$$

where $\xi \sim \mathcal{N}(0, \Sigma_\xi)$ denotes the injected DP noise. The resulting update noise is therefore $\eta\, \mathsf{P}^{-1/2}\xi$ with covariance $\eta^2\, \mathsf{P}^{-1/2}\Sigma_\xi \mathsf{P}^{-1/2}$. Because the preconditioner $\mathsf{P}$ is estimated from noisy gradients, adaptive optimizers inevitably adapt to the injected DP noise. When DP noise dominates the true gradient signal, the preconditioner is largely determined by noise statistics, yielding $\mathsf{P} \approx \mathbb{E}[\xi\xi^\top] = \Sigma_\xi$. In this regime, the update noise covariance becomes $\eta^2\, \mathsf{P}^{-1/2}\Sigma_\xi \mathsf{P}^{-1/2} \approx \eta^2 I$, i.e., adaptive preconditioning can *normalize/reshape* the injected DP noise so that the effective update noise is no longer scaled in a simple way by the base DP-SGD noise level. This noisy-moment-driven behavior is known to undermine the benefits of black-box combinations of DP with adaptive optimizers, and has motivated several DP-aware adaptive variants (e.g., leveraging side information, delayed/stale preconditioners, or bias-corrected moment estimation) (Li et al., 2022; 2023; Tang et al., 2024).

These issues are further exacerbated in LoRA setting. Adaptive and invariant LoRA optimizers often involve operations on small $r \times r$ Gram matrices such as $M = A^\top A$ and $N = B^\top B$ (e.g., via inverses, pseudoinverses, or inverse square roots). DP noise and gauge drift can drive these matrices toward ill-conditioning, where $\| M^{\dagger/2} \|_2 = 1/\sqrt{\lambda_{\min}^+(M)}$ becomes large. This both amplifies noise in the update and destabilizes numerical routines such as eigendecompositions. Consequently, a practical DP-LoRA mechanism must control not only intrinsic sensitivity and noise injection, but also the interaction between privacy noise and adaptive preconditioning under low-rank numerical constraints.

**Design target.** We aim to develop a DP-LoRA procedure whose randomized update of the intrinsic parameter is (i) gauge invariant in distribution, (ii) additive in an intrinsic (tangent) representation, thereby avoiding bilinear noise, and (iii) stable under adaptive optimization and preconditioning, without magnifying DP noise or destabilizing low-rank numerical operations.

## 3. Proposed Method: PRISM

To address the issues in Section 2, we propose the Projected Riemannian Invariant Subspace Mechanism (PRISM), a

---

**Algorithm 1** One PRISM update across all LoRA modules

1: **Input:** LoRA factors $\{(A_\ell, B_\ell)\}_{\ell=1}^L$, minibatch $\{x_i\}_{i=1}^b$, clip $C$, noise multiplier $\sigma$, learning rate $\eta$.
2: **Per-example intrinsic norms:**
3: **for** $i = 1$ to $b$ **do**
4:      For each module $\ell$, compute lifted tangent gradient $(\Delta A_{i,\ell}, \Delta B_{i,\ell})$ via Eq. (14).
5:      Compute $\|\Delta Z_{i,\ell}\|_F^2$ via Eq. (15), and $s_i$ via Eq. (16).
6:      Set clipping coefficient $\alpha_i = \min\{1, C/s_i\}$.
7: **end for**
8: **Module-wise DP tangent update:**
9: **for** each module $\ell$ **do**
10:      $\bar{\Delta} A_\ell = \frac{1}{b}\sum_i \alpha_i \Delta A_{i,\ell}$, $\bar{\Delta} B_\ell = \frac{1}{b}\sum_i \alpha_i \Delta B_{i,\ell}$.
11:      Sample tangent noise $(\Xi_{A,\ell}, \Xi_{B,\ell})$ via Eq. (19).
12:      $\Delta A_\ell^{\mathrm{dp}} = \bar{\Delta} A_\ell + \frac{\sigma C}{b}\Xi_{A,\ell}$, $\Delta B_\ell^{\mathrm{dp}} = \bar{\Delta} B_\ell + \frac{\sigma C}{b}\Xi_{B,\ell}$.
13:      Compute DP-aware invariant adaptive direction $(U_{A,\ell}, U_{B,\ell})$ using Eqs. (24)–(26).
14:      Retract via Eq. (22):
       $(A_\ell, B_\ell) \leftarrow \mathrm{Retr}_r\big(A_\ell, B_\ell; -\eta U_{A,\ell}, -\eta U_{B,\ell}\big)$.
15: **end for**
16: **Output:** updated LoRA factors.

---

DP-LoRA procedure that applies DP-SGD in the *intrinsic* geometry of low-rank updates. For simplicity, we previously described LoRA for a single weight matrix $W = W_0 + AB^\top$; in practice, LoRA is applied to multiple layers of a model. We therefore consider a LoRA model with $L$ *LoRA modules* (i.e., LoRA-augmented layers), indexed by $\ell$ (Hu et al., 2022). Each module $\ell$ has factor parameters $(A_\ell, B_\ell)$ and an intrinsic update $Z_\ell = A_\ell B_\ell^\top$. For each training example $i$, we denote the intrinsic gradient $G_{i,\ell} \triangleq \nabla_{Z_\ell}\ell_i$. By the chain rule, the corresponding factor gradients satisfy $g_{A,i,\ell} = G_{i,\ell}B_\ell$ and $g_{B,i,\ell} = G_{i,\ell}^\top A_\ell$.

**Overview.** Algorithm 1 summarizes one PRISM update applied across all LoRA modules. Rather than directly perturbing the non-identifiable factors, it performs DP-SGD on the intrinsic parameters $\{Z_\ell\}$ by operating on tangent directions of the fixed-rank manifold $\mathcal{M}_r$. For each sample $i$ and module $\ell$, PRISM computes a *lifted tangent gradient* $(\Delta A_{i,\ell}, \Delta B_{i,\ell})$ (line 4), which is a factor-space representation of an intrinsic tangent matrix $\Delta Z_{i,\ell} \in T_{Z_\ell}\mathcal{M}_r$ satisfying $\Delta Z_{i,\ell} = \Delta A_{i,\ell}B_\ell^\top + A_\ell \Delta B_{i,\ell}^\top$. Using these intrinsic tangent matrices, PRISM computes a per-example intrinsic norm aggregated across all modules and applies a single *global* clipping coefficient $\alpha_i$ (lines 3-7), yielding a unified sensitivity bound for the entire LoRA update.

PRISM then proceeds module-wise: it averages the clipped lifted tangents over the minibatch (line 10) and adds isotropic tangent Gaussian noise to form the DP tangent update $(\Delta A_\ell^{\mathrm{dp}}, \Delta B_\ell^{\mathrm{dp}})$ (lines 11–12). It applies a DP-aware, gauge-invariant adaptive transform to obtain $(U_{A,\ell}, U_{B,\ell})$

(line 13), and retracts back to rank $r$ to update $(A_\ell, B_\ell)$ (line 14). The remainder of this section details these components and explains how they address Issues I–III.

## 3.1. Tackle Issue I: Gauge-Invariant Tangent Projection

To eliminate the gauge dependence of factor-space updates, we treat the intrinsic update $Z_\ell = A_\ell B_\ell^\top$ as a point on the fixed-rank manifold $\mathcal{M}_r$ and operate directly in its tangent space. For full-column-rank $A_\ell, B_\ell$, the tangent space at $Z_\ell$ admits the characterization (Appendix A.6)

$$T_{Z_\ell}\mathcal{M}_r = \{\Delta Z_\ell = \Delta A_\ell B_\ell^\top + A_\ell \Delta B_\ell^\top\}, \quad (11)$$

where $\Delta A_\ell \in \mathbb{R}^{m \times r}$ and $\Delta B_\ell \in \mathbb{R}^{n \times r}$. We refer to any pair $(\Delta A_\ell, \Delta B_\ell)$ satisfying this relation as a (factor-space) *lift* of the intrinsic tangent matrix $\Delta Z_\ell$. While such lifts are not unique, the induced matrix $\Delta Z_\ell$ depends only on $Z_\ell$ (Appendix A.7).

To define a gauge-invariant intrinsic gradient, we introduce the orthogonal projectors onto the column spaces of factors,

$$\Pi_{A_\ell} \triangleq A_\ell(A_\ell^\top A_\ell)^\dagger A_\ell^\top, \quad \Pi_{B_\ell} \triangleq B_\ell(B_\ell^\top B_\ell)^\dagger B_\ell^\top. \quad (12)$$

Given a per-example intrinsic gradient $G_{i,\ell} \in \mathbb{R}^{m \times n}$, we project it onto the tangent space via

$$\mathcal{P}_{A_\ell, B_\ell}(G_{i,\ell}) \triangleq G_{i,\ell} - (I - \Pi_{A_\ell})G_{i,\ell}(I - \Pi_{B_\ell}) \quad (13)$$
$$= \Pi_{A_\ell}G_{i,\ell} + G_{i,\ell}\Pi_{B_\ell} - \Pi_{A_\ell}G_{i,\ell}\Pi_{B_\ell}.$$

which depends only on $\Pi_{A_\ell}, \Pi_{B_\ell}$, and hence is invariant to the gauge transformation. As a result, the projected tangent direction $\mathcal{P}_{A_\ell, B_\ell}(G_{i,\ell})$ represents an intrinsic update of $Z_\ell$ that is independent of the chosen factorization (Appendix A.8).

To obtain a concrete factor-space representation, we adopt a canonical horizontal lift that maps the intrinsic tangent direction back to factor space in a gauge-consistent manner. Let $g_{A,i,\ell} = G_{i,\ell}B_\ell$, $g_{B,i,\ell} = G_{i,\ell}^\top A_\ell$, and define $M_\ell = A_\ell^\top A_\ell$, $N_\ell = B_\ell^\top B_\ell$. We set

$$\Delta A_{i,\ell} = g_{A,i,\ell}N_\ell^\dagger - \tfrac{1}{2}\Pi_{A_\ell}(g_{A,i,\ell}N_\ell^\dagger),$$
$$\Delta B_{i,\ell} = g_{B,i,\ell}M_\ell^\dagger - \tfrac{1}{2}\Pi_{B_\ell}(g_{B,i,\ell}M_\ell^\dagger). \quad (14)$$

which satisfies $\Delta A_{i,\ell}B_\ell^\top + A_\ell \Delta B_{i,\ell}^\top = \mathcal{P}_{A_\ell, B_\ell}(G_{i,\ell})$ (Appendix A.9).

## 3.2. Tackle Issue II: Global Intrinsic DP Mechanism

We next design a DP mechanism that operates directly on intrinsic tangent updates, thereby avoiding amplified noise inherent to factor-space perturbations. Given lifted tangent directions $(\Delta A_{i,\ell}, \Delta B_{i,\ell})$ obtained from Eq. (14), we measure the magnitude of tangent directions using the Frobenius

norm $\|\Delta Z_{i,\ell}\|_F^2 = \|\Delta A_{i,\ell}B_\ell^\top + A_\ell \Delta B_{i,\ell}^\top\|_F^2$, which can be computed efficiently (Appendix A.10):

$$\|\Delta Z_{i,\ell}\|_F^2 = \operatorname{tr}(\Delta A_{i,\ell}^\top \Delta A_{i,\ell} N_\ell) + \operatorname{tr}(\Delta B_{i,\ell}^\top \Delta B_{i,\ell} M_\ell)$$
$$+ 2\operatorname{tr}((A_\ell^\top \Delta A_{i,\ell})(B_\ell^\top \Delta B_{i,\ell})). \quad (15)$$

In the common per-example gradient setting, Eq. (15) further simplifies (Appendix A.11).

To control sensitivity across all LoRA modules, we aggregate intrinsic norms and define the *global intrinsic norm*

$$s_i \triangleq \left(\sum_{\ell=1}^L \|\Delta Z_{i,\ell}\|_F^2\right)^{1/2} \quad (16)$$

We then compute per-example clipping coefficients $\alpha_i \triangleq \min\{1, C/s_i\}$. Each module aggregates the clipped lifts as $\bar{\Delta}A_\ell = \frac{1}{b}\sum_i \alpha_i \Delta A_{i,\ell}$ and $\bar{\Delta}B_\ell = \frac{1}{b}\sum_i \alpha_i \Delta B_{i,\ell}$. This mirrors DP-SGD sensitivity control (Eq. (3)), but crucially operates in the intrinsic geometry of LoRA.

**Isotropic tangent noise.** For each module $\ell$, PRISM adds Gaussian noise directly in the tangent space,

$$(\Delta A_\ell^{\mathrm{dp}}, \Delta B_\ell^{\mathrm{dp}}) = (\bar{\Delta}A_\ell, \bar{\Delta}B_\ell) + \frac{\sigma C}{b}(\Xi_{A,\ell}, \Xi_{B,\ell}), \quad (17)$$

where the random pair $(\Xi_{A,\ell}, \Xi_{B,\ell})$ is constructed so that $\Xi_{A,\ell}B_\ell^\top + A_\ell \Xi_{B,\ell}^\top \sim \mathcal{P}_{A_\ell, B_\ell}(\Xi_\ell)$ for $\Xi_\ell \sim \mathcal{N}(0, I_{m_\ell \times n_\ell})$ (Appendix A.12). PRISM uses factor-space lifts for efficiency, but the *released* update is intrinsic and invariant to the factor lift; hence it admits an equivalent lift-free form:

$$\widehat{\Delta Z}_\ell = \frac{1}{b}\sum_{i=1}^b \alpha_i \Delta Z_{i,\ell} + \frac{\sigma C}{b}\mathcal{P}_{A_\ell, B_\ell}(\Xi_\ell), \quad (18)$$

where $\Delta Z_{i,\ell} = \mathcal{P}_{A_\ell, B_\ell}(G_{i,\ell}) \in T_{Z_\ell}\mathcal{M}_r$. This intrinsic form is convenient for stating gauge invariance and privacy guarantees; in implementation, to sample $\mathcal{P}_{A_\ell, B_\ell}(\Xi_\ell)$ efficiently, we avoid drawing a full $m_\ell \times n_\ell$ Gaussian matrix and instead use a low-dimensional factor sampler (Appendix A.13-A.15):

$$\Xi_{A,\ell} = (I - \Pi_{A_\ell})\Omega_{A,\ell} N_\ell^{-\frac{1}{2}}, \quad \Xi_{B,\ell} = \Omega_{B,\ell} M_\ell^{-\frac{1}{2}}. \quad (19)$$

with $\Omega_{A,\ell} \sim \mathcal{N}(0, I_{m_\ell \times r})$ and $\Omega_{B,\ell} \sim \mathcal{N}(0, I_{n_\ell \times r})$.

**Theorem 3.1** (Isotropic tangent noise and closed-form intrinsic energy). *Let $\Xi_\ell \in \mathbb{R}^{m_\ell \times n_\ell}$ have i.i.d. $\mathcal{N}(0,1)$ entries. Then $\mathcal{P}_{A_\ell, B_\ell}(\Xi_\ell)$ is an isotropic Gaussian supported on $T_{Z_\ell}\mathcal{M}_r$ and*

$$\mathbb{E}\|\mathcal{P}_{A_\ell, B_\ell}(\Xi_\ell)\|_F^2 = r(m_\ell + n_\ell - r). \quad (20)$$

*Therefore, the effective intrinsic noise of PRISM perturbation $\mathcal{N}_{Z_\ell}^{PRISM} = \frac{\sigma C}{b}\mathcal{P}_{A_\ell, B_\ell}(\Xi_\ell)$ is*

$$\mathcal{E}_{Z_\ell}^{PRISM} = \frac{\sigma C}{b}\sqrt{r(m_\ell + n_\ell - r)}. \quad (21)$$

See Appendix A.16 for the proof, with supporting results in Appendix A.17-A.18 and concentration bounds in Appendix A.19. Theorem 3.1 shows that projecting a dense Gaussian matrix onto the rank-$r$ tangent space yields an isotropic Gaussian supported on that subspace with expected squared norm $r(m + n - r)$. Consequently, PRISM induces intrinsic noise on $Z$ whose magnitude depends only on $(\sigma, C, b)$ and layer dimensions, and is independent of gauge-dependent quantities $\|A\|_F, \|B\|_F$.

Retraction without bilinear noise. Given the noisy tangent direction, PRISM updates the intrinsic parameter via a retraction onto the fixed-rank manifold $\mathcal{M}_r$,

$$Z_\ell^+ = \text{Retr}_r\big(Z_\ell - \eta(\Delta A_\ell^{\text{dp}} B_\ell^\top + A_\ell (\Delta B_\ell^{\text{dp}})^\top)\big), \quad (22)$$

where $\text{Retr}_r(\cdot)$ denotes the best rank-$r$ approximation in Frobenius norm (Appendix A.20). Because (22) is additive in the intrinsic tangent perturbation, this step avoids the bilinear second-order noise term $\eta^2 \xi_{A,\ell} \xi_{B,\ell}^\top$ that arises when independently noising both factors (Eq. (7); Appendix A.21). Consequently, the effective intrinsic noise induced by PRISM admits a closed-form characterization (Eq. (21)), and retraction introduces only second-order distortion through the lifted factor-product residual.

Proposition 3.2 (Retraction distortion is second order). *Let* $Z = AB^\top \in \mathcal{M}_r$, *with* $A, B$ *full column rank, and let* $\Delta Z = \Delta AB^\top + A\Delta B^\top \in T_Z\mathcal{M}_r$ *be a lifted tangent perturbation. For the truncated-SVD retraction* $\text{Retr}_r$ *and any* $\eta \geq 0$,

$$\|\text{Retr}_r(Z - \eta\Delta Z) - (Z - \eta\Delta Z)\|_F \leq \eta^2 \|\Delta A\Delta B^\top\|_F. \quad (23)$$

*Since* $\|\Delta A\Delta B^\top\|_F \leq \|\Delta A\|_F \|\Delta B\|_F$, *the distortion is* $O(\eta^2)$ *for fixed* $\Delta A, \Delta B$; *equivalently,* $\text{Retr}_r(Z - \eta\Delta Z) = Z - \eta\Delta Z + O(\eta^2)$ *as* $\eta \to 0$.

Proposition 3.2 shows that retraction is first-order exact for the lifted tangent step used by PRISM. Thus, the DP perturbation remains additive to first order; the only discrepancy is the second-order residual in Eq. (23), rather than an explicit factor-space bilinear noise term.

Theorem 3.3 (Gauge invariance of PRISM). *Fix* $(\sigma, C, b)$ *and consider one PRISM step at intrinsic state* $Z_\ell = A_\ell B_\ell^\top$. *For any* $R \in \text{GL}(r)$ *and gauge-equivalent factors* $(A_\ell', B_\ell') = (A_\ell R, B_\ell R^{-\top})$, *the distribution of the intrinsic DP increment in Eq. (18) is invariant:*

$$\widehat{\Delta Z}_\ell(A_\ell, B_\ell) \overset{d}{=} \widehat{\Delta Z}_\ell(A_\ell', B_\ell').$$

*Since retraction in Eq. (22) is deterministic post-processing,* $Z_\ell^+$ *is also gauge invariant in distribution.*

Theorem 3.3 implies that PRISM is a well-defined randomized mechanism on the rank-$r$ manifold: the law of the clipped-and-noised increment is determined by $Z_\ell$ alone, not by the particular factor gauge $(A_\ell, B_\ell)$. See Appendix A.22 for the proof.

Privacy guarantee. Eq. (18) is a (subsampled) Gaussian mechanism on a linear space, and all subsequent operations—adaptive post-processing, factorization, alignment, and retraction—are DP-preserving by post-processing (Appendix A.23 and Appendix A.24).

Theorem 3.4 (DP guarantee of PRISM). *Assume Poisson subsampling with rate* $q = b/N$ *and per-example intrinsic clipping at threshold* $C$ *(Eq. (16)). Each PRISM iteration is a subsampled Gaussian mechanism with noise multiplier* $\sigma$. *Consequently, for any target* $\delta \in (0, 1)$, *after* $T$ *iterations PRISM satisfies* $(\varepsilon, \delta)$-*DP, where* $\varepsilon$ *is determined by composing* $T$ *subsampled Gaussian mechanisms and can be computed numerically using the privacy loss random variable (PRV) accountant (Gopi et al., 2021; Yousefpour et al., 2022; Opacus Contributors, 2026).*

Theorem 3.4 shows that each PRISM iteration is a subsampled Gaussian mechanism on intrinsic tangent updates; the remaining operations are DP-preserving post-processing. Hence, standard DP-SGD accounting applies; see Appendix A.25 for the composition analysis.

### 3.3. Tackle Issue III: DP-Aware Gauge-Invariant Adaptivity and Numerical Stability

The mechanism in Eq. (17) produces a DP-sanitized tangent direction; by the post-processing property of DP, any subsequent transformation preserves the DP guarantee. We leverage this property to design a gauge-invariant adaptive update that is robust to privacy noise. For clarity, we describe the computation for a single LoRA module $\ell$.

Right-invariant preconditioning in rank space. For each module $\ell$, we track first moments $m_{A,\ell}, m_{B,\ell}$ and rank-space second moments $V_{A,\ell}, V_{B,\ell} \in \mathbb{R}^{r \times r}$ defined as,

$$m_{A,\ell} \leftarrow \beta_1 m_{A,\ell} + (1 - \beta_1) \Delta A_\ell^{\text{dp}},$$

$$V_{A,\ell} \leftarrow \beta_2 V_{A,\ell} + (1 - \beta_2) \frac{(\Delta A_\ell^{\text{dp}})^\top \Delta A_\ell^{\text{dp}}}{m_\ell}. \quad (24)$$

with analogous updates for $m_{B,\ell}, V_{B,\ell}$ (replacing $m_\ell$ by $n_\ell$). We precondition on the *right* by inverse square roots and set the adaptive direction

$$U_{A,\ell} = m_{A,\ell} (V_{A,\ell} + \lambda_{A,\ell} I)^{-1/2},$$

$$U_{B,\ell} = m_{B,\ell} (V_{B,\ell} + \lambda_{B,\ell} I)^{-1/2}. \quad (25)$$

Under a gauge action $(A_\ell, B_\ell) \mapsto (A_\ell R, B_\ell R^{-\top})$, $V_{A,\ell}$ and $V_{B,\ell}$ transform by congruence and (25) yields the same intrinsic update $U_{A,\ell} B_\ell^\top + A_\ell U_{B,\ell}^\top$.

DP-aware floors and conditioning control. Adaptive preconditioners can amplify DP noise when $V_{A,\ell}$ or $V_{B,\ell}$

*Table 2.* **Utility on GLUE8 and Math-10K (higher is better).** "Non-DP" uses the same setup without DP clipping/noise; $\varepsilon \in \{6, 3\}$ uses DP-SGD with $\delta = 10^{-5}$. **Avg** is the unweighted mean over the 12 tasks; bold is best per column. *Takeaway:* Under DP, PRISM attains the best Avg and wins most tasks, especially on multi-step reasoning (GSM8K/MAWPS/SVAMP).

| SETTING | METHOD | GLUE8 | | | | | | | | MATH-10K | | | | AVG |
|---|---|---|---|---|---|---|---|---|---|---|---|---|---|---|
| | | CoLA | SST-2 | MRPC | STS-B | QQP | MNLI | QNLI | RTE | GSM8K | AQuA | MAWPS | SVAMP | |
| NON-DP | FFA | 0.456 | 0.935 | 0.759 | 0.821 | 0.713 | 0.736 | 0.809 | 0.809 | 0.513 | 0.476 | 0.836 | 0.678 | 0.712 |
| | RITE | 0.515 | 0.947 | **0.883** | **0.873** | **0.821** | **0.846** | **0.889** | **0.895** | **0.595** | **0.488** | **0.899** | **0.736** | **0.782** |
| | ADAMW | 0.504 | **0.954** | 0.831 | 0.863 | 0.766 | 0.813 | 0.846 | 0.848 | 0.561 | 0.476 | 0.870 | 0.698 | 0.752 |
| | LORA+ | **0.578** | 0.950 | 0.840 | 0.862 | 0.807 | 0.845 | 0.851 | 0.838 | 0.592 | 0.465 | 0.891 | 0.712 | 0.769 |
| | LAMB | 0.468 | 0.939 | 0.860 | 0.872 | 0.776 | 0.842 | 0.868 | 0.856 | 0.559 | 0.449 | 0.878 | 0.708 | 0.756 |
| | PRISM | 0.392 | 0.921 | 0.857 | 0.822 | 0.797 | 0.814 | 0.834 | 0.798 | 0.552 | 0.472 | 0.895 | 0.693 | 0.737 |
| $\epsilon = 6$ | FFA | 0.355 | 0.907 | 0.738 | 0.465 | 0.479 | 0.579 | 0.684 | 0.755 | 0.375 | 0.390 | 0.735 | 0.611 | 0.589 |
| | RITE | 0.235 | 0.787 | 0.635 | 0.268 | 0.500 | 0.482 | 0.562 | 0.657 | 0.282 | 0.366 | 0.597 | 0.503 | 0.490 |
| | ADAMW | 0.407 | 0.915 | 0.770 | 0.659 | 0.493 | 0.651 | 0.716 | 0.798 | 0.441 | **0.465** | 0.761 | 0.615 | 0.641 |
| | LORA+ | 0.436 | 0.897 | 0.787 | 0.691 | 0.739 | **0.721** | 0.747 | **0.823** | 0.446 | 0.409 | 0.786 | 0.611 | 0.674 |
| | LAMB | 0.414 | **0.920** | 0.756 | 0.544 | 0.521 | 0.602 | 0.709 | 0.776 | 0.425 | 0.437 | 0.761 | 0.592 | 0.621 |
| | PRISM | **0.444** | 0.919 | **0.798** | **0.718** | **0.770** | 0.707 | **0.776** | 0.791 | **0.469** | 0.445 | **0.819** | **0.626** | **0.690** |
| $\epsilon = 3$ | FFA | 0.337 | 0.890 | 0.730 | 0.406 | 0.466 | 0.561 | 0.662 | 0.740 | 0.350 | 0.374 | 0.718 | 0.598 | 0.569 |
| | RITE | 0.221 | 0.713 | 0.636 | 0.260 | 0.485 | 0.463 | 0.548 | 0.606 | 0.255 | 0.362 | 0.525 | 0.474 | 0.462 |
| | ADAMW | 0.410 | 0.903 | 0.778 | 0.622 | 0.555 | 0.633 | 0.718 | **0.812** | 0.446 | 0.413 | 0.731 | 0.591 | 0.634 |
| | LORA+ | **0.434** | 0.906 | **0.798** | 0.668 | 0.730 | 0.708 | 0.740 | **0.812** | 0.419 | 0.386 | 0.765 | 0.609 | 0.665 |
| | LAMB | 0.396 | **0.909** | 0.759 | 0.517 | 0.486 | 0.586 | 0.708 | 0.783 | 0.393 | **0.425** | 0.744 | 0.608 | 0.609 |
| | PRISM | 0.406 | 0.884 | 0.784 | **0.729** | **0.770** | **0.732** | **0.791** | 0.780 | **0.456** | 0.406 | **0.807** | **0.614** | **0.680** |

has small or ill-conditioned eigenvalues. To mitigate this, PRISM introduces DP-aware floors $\lambda_{A,\ell}, \lambda_{B,\ell}$, scaled according to the known DP noise level $\left(\frac{\sigma C}{b}\right)^2$ and the geometry of the current LoRA module. For isotropic tangent noise, the rank-space noise covariances satisfy $\mathbb{E}[\Xi_{A,\ell}^\top \Xi_{A,\ell}/m_\ell] = \frac{m_\ell - r}{m_\ell} N_\ell^{-1}$ and $\mathbb{E}[\Xi_{B,\ell}^\top \Xi_{B,\ell}/n_\ell] = M_\ell^{-1}$ (Appendix A.26). Small eigenvalues of $M_\ell$ or $N_\ell$ therefore simultaneously increase DP noise seen by the preconditioner and degrade numerical stability. Motivated by this, we set

$$\lambda_{A,\ell} \asymp \left(\frac{\sigma C}{b}\right)^2 \frac{\mathrm{tr}(N_\ell^{-1})}{r}, \quad \lambda_{B,\ell} \asymp \left(\frac{\sigma C}{b}\right)^2 \frac{\mathrm{tr}(M_\ell^{-1})}{r}. \quad (26)$$

These operations yield a uniform bound on DP noise amplification under the adaptive post-processing.

**Theorem 3.5** (Bounding DP noise amplification under adaptive preconditioning). *Let $V \succeq 0$ and $\lambda > 0$, and define $\mathsf{P} = V + \lambda I$. Then $\|\mathsf{P}^{-1/2}\|_2 \leq \lambda^{-1/2}$ and for any matrix $X$,*

$$\|X\mathsf{P}^{-1/2}\|_F^2 \leq \lambda^{-1}\|X\|_F^2. \quad (27)$$

Theorem 3.5 formalizes Issue III (Eq. (10)): DP-noise amplification under inverse-square-root right-preconditioning is governed by the preconditioner's spectral gain $g(\mathsf{P}) \triangleq \|\mathsf{P}^{-1/2}\|_2 = 1/\sqrt{\lambda_{\min}^+(\mathsf{P})}$. A floor $\mathsf{P} = V + \lambda I$ forces $g(\mathsf{P}) \leq 1/\sqrt{\lambda}$, hence the Frobenius energy of any perturbation can increase by at most $1/\lambda$.

In PRISM, Eq. (17) injects DP noise into $(\Delta A_\ell^{\mathrm{dp}}, \Delta B_\ell^{\mathrm{dp}})$, and Eq. (25) post-processes it by $(V_{A,\ell} + \lambda_{A,\ell}I)^{-1/2}$ and

$(V_{B,\ell} + \lambda_{B,\ell}I)^{-1/2}$. The DP-aware floors in Eq. (26) keep $\lambda_{A,\ell}, \lambda_{B,\ell}$ from collapsing when $V_{A,\ell}, V_{B,\ell}$ (or $M_\ell, N_\ell$) are ill-conditioned, so Eq. (27) yields controlled DP-noise amplification (bounded by $1/\lambda_{A,\ell}$ and $1/\lambda_{B,\ell}$), and intrinsic clipping caps the final step size (Appendix A.27).

## 4. Experiments

We benchmark PRISM for private LoRA fine-tuning on two multi-task instruction suites: **GLUE8** (NLU) and **Math-10K** (multi-step numerical reasoning), spanning diverse linguistic phenomena and compositional reasoning tasks to assess robustness across settings.

**Setup.** We fine-tune the `Gemma-3-4B-pt` backbone (Gemma Team et al., 2025) with LoRA (Hu et al., 2022). Our implementation follows LLM-Adapters (Hu et al., 2023; AGI-Edgerunners, 2023): Math-10K is used in its original form, while GLUE8 is constructed from GLUE (Wang et al., 2018) in the same instruction-format interface. We report both **non-private** results and $(\varepsilon, \delta)$-**DP** results using DP-SGD (Abadi et al., 2016) with $\varepsilon \in \{3, 6\}$ and $\delta = 10^{-5}$. We use Opacus (Yousefpour et al., 2022; Opacus Contributors, 2026) with the default PRV accountant (Gopi et al., 2021). Full details are provided in Appendix B.1.

**Datasets and metrics. GLUE8** consists of eight GLUE tasks (excluding WNLI) (Wang et al., 2018). We evaluate on the official validation splits using standard GLUE metrics. **Math-10K** combines GSM8K (Cobbe et al., 2021), AQuA (Ling et al., 2017), MAWPS (Koncel-Kedziorski

*Table 3.* Math-10K results on `Gemma-3-4B-pt`, `Gemma-2-9B`, and `Gemma-3-12B-pt` under DP ($r = 16$, $\varepsilon = 6$, $\delta = 10^{-5}$). The "Type" column indicates the backbone family; all experiments use text-only inputs. Bold indicates the best result within each backbone.

| Backbone | Type | Method | GSM8K | AQuA | MAWPS | SVAMP | Avg |
|---|---|---|---|---|---|---|---|
| `Gemma-3-4B-pt` | Multimodal | AdamW | 0.441 | **0.465** | 0.761 | 0.615 | 0.571 |
| | | LoRA+ | 0.446 | 0.409 | 0.786 | 0.611 | 0.563 |
| | | PRISM | **0.469** | 0.445 | **0.819** | **0.626** | **0.590** |
| `Gemma-2-9B` | Text-only | AdamW | 0.6473 | 0.4979 | 0.8093 | 0.7570 | 0.6779 |
| | | LoRA+ | 0.6293 | 0.4409 | 0.8067 | 0.6970 | 0.6435 |
| | | PRISM | **0.6603** | **0.5197** | **0.8487** | **0.7790** | **0.7019** |
| `Gemma-3-12B-pt` | Multimodal | AdamW | 0.5807 | 0.4764 | 0.7311 | 0.6870 | 0.6188 |
| | | LoRA+ | 0.6346 | 0.5039 | **0.8193** | 0.7460 | 0.6760 |
| | | PRISM | **0.6535** | **0.5315** | **0.8193** | **0.7820** | **0.6966** |

et al., 2016), and SVAMP (Patel et al., 2021) via LLM-Adapters (Hu et al., 2023). Performance is measured by exact answer accuracy using the LLM-Adapters protocol.

**Baselines.** We compare PRISM against **FFA** (Sun et al., 2024), **LoRA-RITE** (Yen et al., 2025), **AdamW** (Kingma & Ba, 2015; Loshchilov & Hutter, 2019), **LoRA+** (Hayou et al., 2024), and **LAMB** (You et al., 2020).

**Main Results and Interpretation.** Table 2 reports utility under non-private training and DP training with $\varepsilon \in \{6, 3\}$. Without DP, LoRA-RITE achieves the best average performance, which is expected since PRISM is designed to address *DP-specific* issues rather than to improve non-private optimization. Under DP, PRISM achieves the best average performance at both privacy budgets (0.690 at $\varepsilon = 6$; 0.680 at $\varepsilon = 3$) and wins the majority of tasks (8/12 and 7/12, respectively). PRISM is not best on every task (e.g., SST-2, RTE, and AQuA). This is expected because GLUE8 and Math-10K are trained as task suites with shared hyperparameters, while individual tasks have different convergence rates and sensitivities to DP noise.

The mechanism-level distinction is where the DP perturbation is applied. Factor-space DP perturbs the factors $(A, B)$; after multiplication, the effective perturbation on $Z = AB^\top$ is scaled by the current factor norms and therefore depends on the chosen gauge. Thus, under the same nominal privacy budget, factor-space DP can induce uneven intrinsic noise across layers and tasks, causing some components to be over-noised. PRISM instead clips and noises the gauge-invariant tangent update of $Z$, so the induced intrinsic noise is bounded and independent of the factorization. This makes the private updates more stable across heterogeneous task suites and improves the average DP utility, even when another optimizer is best on a few individual tasks.

**Additional Scaling and Overhead Results.** To assess robustness beyond the main setting, we further evaluate PRISM on larger backbones, across varying LoRA ranks, and with explicit runtime and memory profiling. For the backbone type, `Gemma-2-9B` is a text-

only language model (Gemma Team et al., 2024), while `Gemma-3-4B-pt` and `Gemma-3-12B-pt` belong to the multimodal Gemma 3 family (Gemma Team et al., 2025); all benchmark inputs in our experiments are text-only.

Table 3 shows that PRISM's advantage persists on both `Gemma-2-9B` and `Gemma-3-12B-pt`. Table 4 shows that PRISM remains consistently the best method across ranks $r \in \{8, 16, 32\}$ on `Gemma-3-4B-pt`. Finally, Table 5 shows that PRISM's overhead is primarily in runtime rather than memory usage, consistent with its additional geometry-aware computations.

*Table 4.* Rank sensitivity on `Gemma-3-4B-pt` under DP ($\varepsilon = 6$, $\delta = 10^{-5}$; text-only inputs). Avg is the unweighted mean over all 12 tasks. PRISM is best across all tested ranks.

| Rank | Method | GLUE8 Avg | Math Avg | Avg |
|---|---|---|---|---|
| $r = 8$ | AdamW | 0.659 | 0.522 | 0.614 |
| | LoRA+ | 0.704 | 0.543 | 0.650 |
| | PRISM | **0.744** | **0.565** | **0.684** |
| $r = 16$ | AdamW | 0.676 | 0.571 | 0.641 |
| | LoRA+ | 0.730 | 0.563 | 0.674 |
| | PRISM | **0.740** | **0.590** | **0.690** |
| $r = 32$ | AdamW | 0.682 | 0.542 | 0.636 |
| | LoRA+ | 0.721 | 0.516 | 0.653 |
| | PRISM | **0.740** | **0.566** | **0.682** |

*Table 5.* Runtime and memory profiling on `Gemma-3-4B-pt` for Math-10K ($r = 16$, $\varepsilon = 6$). Measurements use a single A100-40GB GPU with 10 warmup updates and 30 measured updates. PRISM roughly doubles step time in the current implementation, while peak memory is essentially unchanged.

| Method | Step time (s) | Peak memory (MB) |
|---|---|---|
| LoRA+ | 9.32 | 20961.1 |
| AdamW | 9.37 | 20961.1 |
| PRISM | 18.64 | 20964.3 |

**Mechanism Diagnostics (Three Issues).** We next analyze how DP clipping and noise injection propagate to the intrinsic update $Z$ during Math-10K training (300 steps), and

relate these effects to the three issues identified earlier. Figure 1 illustrates systematic amplification of intrinsic noise in factor-space DP compared to PRISM.

*Issue I: gauge sensitivity.* We perform DP training from gauge-rescaled initializations $(A, B) \mapsto (cA, c^{-1}B)$ with $c \in \{0.25, 0.5, 1, 2, 4\}$. Figure 2 shows that factor-space DP remains sensitive to this benign reparameterization (Eq. (4)), whereas PRISM quickly reduces this variability to near zero after warm-up (Appendix C.1).

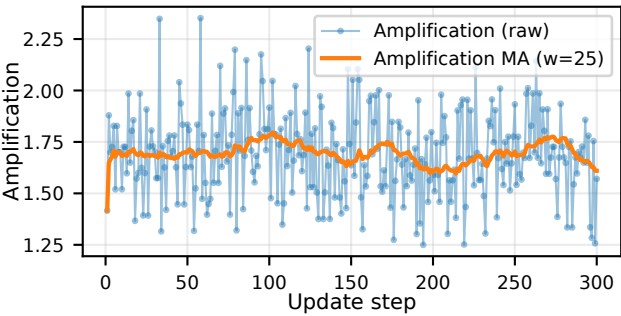

*Figure 1.* **Intrinsic DP-noise amplification during training (Math-10K).** We plot the per-step ratio $\|N_Z\|_F^{\text{fac}}/\|N_Z\|_F^{\text{PRISM}}$, where $\|N_Z\|_F$ is the Frobenius norm of the effective DP noise on the merged LoRA update $Z$. The blue curve reports the raw per-step ratio, and the orange curve reports its moving average (MA) with window size $w = 25$ updates. Values $> 1$ indicate that applying DP-SGD in factor space $(A, B)$ injects a larger intrinsic perturbation into $Z$ than PRISM under the same privacy budget.

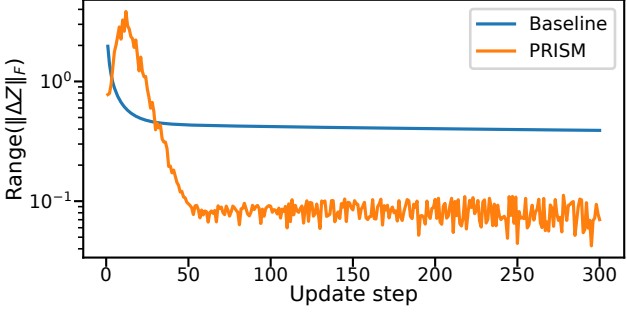

*Figure 2.* **Gauge sensitivity under DP (Math-10K).** At step $t$ we compute $\rho_t = \max_c \|\Delta Z_t\|_F - \min_c \|\Delta Z_t\|_F$ across gauge-rescaled runs; smaller is better and $\rho_t \approx 0$ indicates practical gauge invariance. PRISM drives $\rho_t$ near zero, whereas factor-space DP exhibits persistently large $\rho_t$.

*Issue II: gauge-dependent intrinsic noise.* Proposition 2.2 predicts $\mathbb{E}\|N_Z\|_F^2 = \tau^2 S_t$ for factor-space DP, where $S_t = \sum_\ell (m_\ell \|B_\ell\|_F^2 + n_\ell \|A_\ell\|_F^2)$ varies under gauge rescaling. Figure 3 confirms this linear scaling behavior for the baseline, while PRISM remains substantially lower and nearly invariant to $S_t$ (Eq. (21)). Appendix C.2 further fixes $Z$ and varies $c$ to reproduce Corollary 2.3.

*Issue III: DP under adaptive preconditioning.* Figure 4 sweeps the DP noise multiplier $\sigma$ and measures the resulting

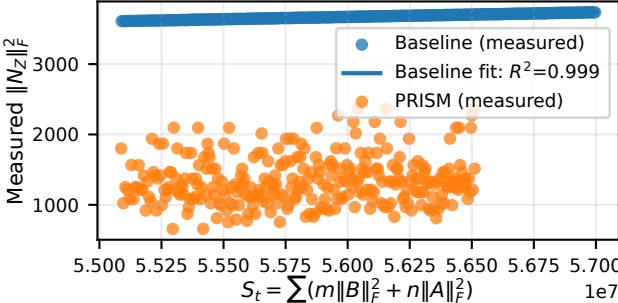

*Figure 3.* **Gauge-dependent intrinsic DP noise (Math-10K).** We plot the measured intrinsic noise energy $\|N_Z\|_F^2$ against the gauge-dependent statistic $S_t$. A strong linear trend indicates that the amount of DP noise injected into $Z$ depends on the factorization; PRISM largely removes this dependence and keeps $\|N_Z\|_F^2$ low.

preconditioned intrinsic noise magnitude (Eq. (10)). Factor-space DP-AdamW exhibits a "noise-normalization" effect (Proposition A.29), whereas PRISM with DP-aware floors (Eq. (26)) consistently reduces the perturbation across all $\sigma$, which aligns with Theorem 3.5 (Appendix C.3).

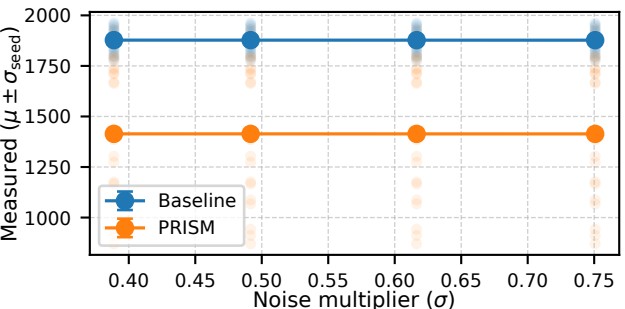

*Figure 4.* **Preconditioned intrinsic DP noise vs. $\sigma$ (Math-10K).** We report $\mathbb{E}\|P_t^{-1/2}\boldsymbol{\xi}_{\text{intr}}\|_F$; lower means the optimizer applies less stochastic perturbation after preconditioning. Factor-space DP becomes nearly $\sigma$-invariant, while PRISM keeps the preconditioned noise smaller via DP-aware floors.

**Limitations.** PRISM is tailored to LoRA-style fixed-rank updates; extending it to other PEFT methods requires deriving the corresponding intrinsic geometry. Its main practical cost is runtime overhead from geometry-aware operations, while peak memory remains nearly unchanged.

# Acknowledgements

This work was funded in part by the National Science Foundation under award number IIS2202699, IIS-2416895, IIS-2301599, CMMI2301601, and DMS-2529302.

# Impact Statement

This work develops a differentially private method for LoRA fine-tuning. We expect it to reduce privacy risks when adapt-

ing models on sensitive data; we do not foresee additional negative societal impacts beyond those typical of deploying ML systems.

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

# Appendix Roadmap

# A. Theory and Proofs

This appendix provides proofs and additional derivations. We use $\langle X, Y \rangle = \mathrm{tr}(X^\top Y)$, $\|\cdot\|_F$ for the Frobenius norm, $\mathrm{vec}(\cdot)$ for vectorization, and $\otimes$ for the Kronecker product. For a symmetric positive semidefinite matrix $X$, $X^\dagger$ denotes the Moore–Penrose pseudoinverse.

## A.1. Quotient geometry: rank-$r$ matrices as a gauge quotient

The non-identifiability $(A_\ell, B_\ell) \sim (A_\ell R, B_\ell R^{-\top})$ can be formalized as a smooth group action. Let $\tilde{\mathcal{M}} \triangleq \mathbb{R}_*^{m \times r} \times \mathbb{R}_*^{n \times r}$ be the *total space* of full-column-rank factors and let $\mathrm{GL}(r)$ act on $\tilde{\mathcal{M}}$ by

$$(A_\ell, B_\ell) \cdot R \;=\; (A_\ell R, \; B_\ell R^{-\top}), \qquad R \in \mathrm{GL}(r). \tag{28}$$

The map $\pi : \tilde{\mathcal{M}} \to \mathcal{M}_r$ given by $\pi(A_\ell, B_\ell) = A_\ell B_\ell^\top$ is constant along orbits of this action. Under mild regularity conditions, the rank-$r$ manifold is the quotient $\mathcal{M}_r \cong \tilde{\mathcal{M}}/\mathrm{GL}(r)$, and *intrinsic* quantities on $Z_\ell$ are precisely those that are invariant under (28).

**Vertical and horizontal spaces.**    Differentiating the action (28) at the identity $R = \mathbf{I}_r$ yields the *vertical space* (tangent to the gauge orbit) at $(A_\ell, B_\ell)$:

$$\mathcal{V}_{(A_\ell, B_\ell)} \;=\; \{(A_\ell \Omega, \, -B_\ell \Omega^\top) : \, \Omega \in \mathbb{R}^{r \times r}\}. \tag{29}$$

A complementary *horizontal space* $\mathcal{H}_{(A_\ell, B_\ell)}$ selects one representative lift of each intrinsic tangent direction. With the Frobenius metric on $\tilde{\mathcal{M}}$, the orthogonal complement of (29) is characterized by

$$\mathcal{H}_{(A_\ell, B_\ell)} \;=\; \left\{ (\Delta A_\ell, \Delta B_\ell) : \; A_\ell^\top \Delta A_\ell \;=\; (B_\ell^\top \Delta B_\ell)^\top \right\}, \tag{30}$$

This is exactly the gauge-fixing constraint enforced by our canonical lift (14).

**Lemma A.1** (Orthogonality to the gauge orbit). *A pair $(\Delta A_\ell, \Delta B_\ell)$ satisfies (30) if and only if it is orthogonal to every vertical direction under the Frobenius inner product on $\tilde{\mathcal{M}}$:*

$$\langle (\Delta A_\ell, \Delta B_\ell), (A_\ell \Omega, -B_\ell \Omega^\top) \rangle = \mathrm{tr}(\Delta A_\ell^\top A_\ell \Omega) - \mathrm{tr}(\Delta B_\ell^\top B_\ell \Omega^\top) = 0, \qquad \forall \Omega.$$

*Proof.* The stated inner product equals $\mathrm{tr}(\Omega A_\ell^\top \Delta A_\ell) - \mathrm{tr}(\Omega (B_\ell^\top \Delta B_\ell)^\top) = \mathrm{tr}(\Omega(A_\ell^\top \Delta A_\ell - (B_\ell^\top \Delta B_\ell)^\top))$. Since this vanishes for all $\Omega$ if and only if $A_\ell^\top \Delta A_\ell = (B_\ell^\top \Delta B_\ell)^\top$, we obtain (30). □

**Why this matters for DP.** Clipping and adding noise in factor coordinates implicitly chooses a metric on $\tilde{\mathcal{M}}$, not on the quotient $\mathcal{M}_r$. As a result, the induced intrinsic perturbation can depend on the chosen representative (i.e., the gauge), which is the source of the amplification phenomenon in Section 2. PRISM instead clips and perturbs *horizontal* (gauge-orthogonal) directions and interprets the Gaussian mechanism intrinsically in the quotient geometry. This is why the effective intrinsic noise $\mathcal{E}_{Z_\ell}$ is a fixed, controllable scalar in PRISM (Corollary A.2).

**Corollary A.2** (Effective intrinsic DP noise of PRISM). *Let $\mathcal{N}_{Z_\ell}^{PRISM} = \frac{\sigma C}{b} P_{A_\ell, B_\ell}(\Xi_\ell)$ with $\Xi_\ell \sim \mathcal{N}(0, I_{m \times n})$. Then $\mathcal{E}_{Z_\ell} = \frac{\sigma C}{b} \sqrt{r(m + n - r)}$, independent of the factor gauge.*

**Proposition A.3** (Orthogonal tangent projection). *For $Z_\ell = A_\ell B_\ell^\top \in \mathcal{M}_r$, the linear map $P_{A_\ell, B_\ell}$ in Eq. (13) is the orthogonal projector onto $T_{Z_\ell} \mathcal{M}_r$: for all $G \in \mathbb{R}^{m \times n}$, $P_{A_\ell, B_\ell}(G) \in T_{Z_\ell} \mathcal{M}_r$ and $G - P_{A_\ell, B_\ell}(G) \in (T_{Z_\ell} \mathcal{M}_r)^\perp$. Equivalently, $P_{A_\ell, B_\ell}$ is symmetric and idempotent.*

*Proof of Proposition A.3.* By Lemma A.9, any $G \in \mathbb{R}^{m \times n}$ decomposes orthogonally as

$$G = \underbrace{(G - \Pi_{A_\ell}^\perp G \Pi_{B_\ell}^\perp)}_{\in T_{Z_\ell} \mathcal{M}_r} + \underbrace{\Pi_{A_\ell}^\perp G \Pi_{B_\ell}^\perp}_{\in (T_{Z_\ell} \mathcal{M}_r)^\perp}.$$

Therefore the orthogonal projector onto $T_{Z_\ell} \mathcal{M}_r$ is $P_{A_\ell, B_\ell}(G) = G - \Pi_{A_\ell}^\perp G \Pi_{B_\ell}^\perp = \Pi_{A_\ell} G + G \Pi_{B_\ell} - \Pi_{A_\ell} G \Pi_{B_\ell}$. □

## A.2. General gauge amplification for arbitrary basis changes

Corollary 2.3 considered the scalar rescaling gauge $(A_\ell, B_\ell) \mapsto (cA_\ell, c^{-1} B_\ell)$. Here we record the corresponding expression for a general invertible basis change.

**Proposition A.4** (Gauge-dependent noise energy under general $R$). *Let $(A'_\ell, B'_\ell) = (A_\ell R, B_\ell R^{-\top})$ for invertible $R \in \mathrm{GL}(r)$ and let $\xi_{A,\ell}, \xi_{B,\ell}$ be i.i.d. Gaussian as in Proposition 2.2. Then*

$$\mathbb{E}\left[ \| \xi_{A,\ell} B'^{\top}_\ell \|_F^2 \right] = m \tau^2 \| B_\ell R^{-\top} \|_F^2, \qquad \mathbb{E}\left[ \| A'_\ell \xi_{B,\ell}^\top \|_F^2 \right] = n \tau^2 \| A_\ell R \|_F^2.$$

*Consequently, for any nonzero $Z_\ell = A_\ell B_\ell^\top$, there exist gauge matrices $R$ with arbitrarily large condition number such that the first-order noise energy on $Z_\ell$ becomes arbitrarily large, even though $Z_\ell$ is unchanged.*

*Proof.* The expressions follow by the same calculation as Proposition 2.2, with $B_\ell$ replaced by $B_\ell R^{-\top}$ and $A_\ell$ replaced by $A_\ell R$. For the final statement, take an $R$ that scales one singular direction of $A_\ell$ (and inversely scales the corresponding direction of $B_\ell$) by a large factor; the scalar rescaling case is recovered by $R = c\mathbf{I}$. □

## A.3. Second-order term magnitude and concentration

Proposition 2.2 quantifies the *expected* Frobenius energy of the bilinear term $\mathcal{N}_{Z_\ell}^{(2)} = \xi_{A,\ell} \xi_{B,\ell}^\top$. Here we record a simple high-probability bound that complements (9) and illustrates why the bilinear term can dominate when the learning rate is not extremely small.

**Lemma A.5** (High-probability bound for the bilinear term). *Let $\xi_{A,\ell} \in \mathbb{R}^{m \times r}$ and $\xi_{B,\ell} \in \mathbb{R}^{n \times r}$ have i.i.d. $\mathcal{N}(0, \tau^2)$ entries. Then for any $\delta \in (0, 1)$, with probability at least $1 - 2\delta$,*

$$\|\xi_{A,\ell}\xi_{B,\ell}^\top\|_F \leq \tau^2 \sqrt{\left(mr + 2\sqrt{mr\log(1/\delta)} + 2\log(1/\delta)\right)\left(nr + 2\sqrt{nr\log(1/\delta)} + 2\log(1/\delta)\right)}. \tag{31}$$

*In particular, the typical scale is $\|\xi_{A,\ell}\xi_{B,\ell}^\top\|_F = \Theta(\tau^2 \, r\sqrt{mn})$ up to logarithmic factors.*

*Proof.* Use $\|XY^\top\|_F \leq \|X\|_F\|Y\|_F$. Since $\|\xi_{A,\ell}\|_F^2/\tau^2 \sim \chi^2_{mr}$ and $\|\xi_{B,\ell}\|_F^2/\tau^2 \sim \chi^2_{nr}$, applying the chi-square tail bound used in Proposition A.24 to each and taking a union bound yields (31). $\square$

**Implication for DP-LoRA.** When both factors are randomized, the intrinsic parameter update contains the extra term $\eta^2\xi_{A,\ell}\xi_{B,\ell}^\top$ (Eq. (7)). Lemma A.5 implies this term has magnitude on the order of $\eta^2\tau^2r\sqrt{mn}$, which is not controlled by the intrinsic clipping threshold on the first-order tangent update. PRISM avoids this term altogether (Lemma A.26).

## A.4. Noise amplification under naive factor-space DP

*Proof of Proposition 2.2.* Consider the first-order intrinsic perturbation $\mathcal{N}_{Z_\ell}^{(1)} = \xi_{A,\ell}B_\ell^\top + A_\ell\xi_{B,\ell}^\top$. Expanding the Frobenius norm,

$$\|\mathcal{N}_{Z_\ell}^{(1)}\|_F^2 = \|\xi_{A,\ell}B_\ell^\top\|_F^2 + \|A_\ell\xi_{B,\ell}^\top\|_F^2 + 2\langle\xi_{A,\ell}B_\ell^\top, A_\ell\xi_{B,\ell}^\top\rangle.$$

The cross term has zero expectation because $\xi_{A,\ell}$ and $\xi_{B,\ell}$ are independent and centered:

$$\mathbb{E}\langle\xi_{A,\ell}B_\ell^\top, A_\ell\xi_{B,\ell}^\top\rangle = \mathbb{E}\mathrm{tr}\big((\xi_{A,\ell}B_\ell^\top)^\top(A_\ell\xi_{B,\ell}^\top)\big) = \mathbb{E}\mathrm{tr}(B_\ell\xi_{A,\ell}^\top A_\ell\xi_{B,\ell}^\top) = \mathbb{E}_{\xi_{A,\ell}}\mathrm{tr}\big(B_\ell\xi_{A,\ell}^\top A_\ell\,\mathbb{E}_{\xi_{B,\ell}}[\xi_{B,\ell}^\top]\big) = 0.$$

For the remaining terms,

$$\mathbb{E}\|\xi_{A,\ell}B_\ell^\top\|_F^2 = \mathbb{E}\mathrm{tr}(B_\ell\xi_{A,\ell}^\top\xi_{A,\ell}B_\ell^\top) = \mathrm{tr}\big(B_\ell\,\mathbb{E}[\xi_{A,\ell}^\top\xi_{A,\ell}]\,B_\ell^\top\big).$$

Since $\xi_{A,\ell}$ has i.i.d. $\mathcal{N}(0, \tau^2)$ entries, $\mathbb{E}[\xi_{A,\ell}^\top\xi_{A,\ell}] = m\tau^2\,\mathbf{I}_r$, yielding $m\tau^2\|B_\ell\|_F^2$. Similarly,

$$\mathbb{E}\|A_\ell\xi_{B,\ell}^\top\|_F^2 = \mathbb{E}\mathrm{tr}(\xi_{B,\ell}A_\ell^\top A_\ell\xi_{B,\ell}^\top) = \mathrm{tr}\big(A_\ell^\top A_\ell\,\mathbb{E}[\xi_{B,\ell}^\top\xi_{B,\ell}]\big) = n\tau^2\|A_\ell\|_F^2,$$

because $\mathbb{E}[\xi_{B,\ell}^\top\xi_{B,\ell}] = n\tau^2\,\mathbf{I}_r$. Summing proves (8).

For the bilinear term, each entry of $\xi_{A,\ell}\xi_{B,\ell}^\top$ is a sum of $r$ independent products of mean-zero Gaussians; its variance is $r\tau^4$. Summing variances over $mn$ entries yields $\mathbb{E}\|\xi_{A,\ell}\xi_{B,\ell}^\top\|_F^2 = mnr\,\tau^4$, proving (9). $\square$

**Proposition A.6** (One-sided factor noise). *Let $\xi_{A,\ell} \in \mathbb{R}^{m \times r}$ and $\xi_{B,\ell} \in \mathbb{R}^{n \times r}$ have i.i.d. entries $\mathcal{N}(0, \tau^2)$. For $Z_\ell = A_\ell B_\ell^\top$, consider the intrinsic perturbations obtained by noising only one factor: $\mathcal{N}_{Z_\ell}^{(A)} \triangleq \xi_{A,\ell}B_\ell^\top$ and $\mathcal{N}_{Z_\ell}^{(B)} \triangleq A_\ell\xi_{B,\ell}^\top$. Then*

$$\mathbb{E}\|\mathcal{N}_{Z_\ell}^{(A)}\|_F^2 = \tau^2\,m\|B_\ell\|_F^2, \qquad \mathbb{E}\|\mathcal{N}_{Z_\ell}^{(B)}\|_F^2 = \tau^2\,n\|A_\ell\|_F^2. \tag{32}$$

*Proof of Proposition A.6.* If $A_\ell$ is frozen and only $B_\ell$ is perturbed by $\xi_{B,\ell}$ with i.i.d. $\mathcal{N}(0, \tau^2)$ entries, then the induced intrinsic perturbation is $\mathcal{N}_{Z_\ell} = A_\ell\xi_{B,\ell}^\top$. Therefore

$$\mathbb{E}\|\mathcal{N}_{Z_\ell}\|_F^2 = \mathbb{E}\mathrm{tr}(\xi_{B,\ell}A_\ell^\top A_\ell\xi_{B,\ell}^\top) = \mathrm{tr}\big(A_\ell^\top A_\ell\,\mathbb{E}[\xi_{B,\ell}^\top\xi_{B,\ell}]\big) = n\tau^2\|A_\ell\|_F^2,$$

which is (32). If instead $B_\ell$ is frozen and only $A_\ell$ is perturbed, the same calculation gives $\mathbb{E}\|\mathcal{N}_{Z_\ell}\|_F^2 = m\tau^2\|B_\ell\|_F^2$. In either case there is no bilinear term because only one factor is randomized. $\square$

### A.5. Range of effective intrinsic noise under scalar gauge rescaling

Corollary 2.3 shows that naive factor-space DP can make the effective intrinsic noise arbitrarily large under the scalar gauge $(A_\ell, B_\ell) \mapsto (cA_\ell, c^{-1}B_\ell)$. For completeness, we record the *entire range* of the first-order effective noise over this one-parameter family.

**Proposition A.7** (Range over scalar gauges). *Let $Z_\ell = A_\ell B_\ell^\top \neq 0$ and consider the scalar gauge family $(A_c, B_c) = (cA_\ell, c^{-1}B_\ell)$ for $c > 0$. Let $\xi_{A,\ell}, \xi_{B,\ell}$ have i.i.d. $\mathcal{N}(0, \tau^2)$ entries as in Proposition 2.2, and define the first-order perturbation $\mathcal{N}_{Z,c}^{(1)} \triangleq \xi_{A,\ell} B_c^\top + A_c \xi_{B,\ell}^\top$. Then*

$$\mathbb{E}\big[\|\mathcal{N}_{Z,c}^{(1)}\|_F^2\big] = \tau^2 \Big(\frac{m}{c^2}\|B_\ell\|_F^2 + nc^2\|A_\ell\|_F^2\Big). \tag{33}$$

*The minimizing gauge is*

$$c^\star = \left(\frac{m\|B_\ell\|_F^2}{n\|A_\ell\|_F^2}\right)^{1/4}, \tag{34}$$

*and the minimum value is*

$$\min_{c>0} \mathbb{E}\big[\|\mathcal{N}_{Z,c}^{(1)}\|_F^2\big] = 2\tau^2 \sqrt{mn}\, \|A_\ell\|_F \|B_\ell\|_F. \tag{35}$$

*Moreover, $\sup_{c>0} \mathbb{E}[\|\mathcal{N}_{Z,c}^{(1)}\|_F^2] = \infty$.*

*Proof.* Equation (33) follows directly from (8) after substituting $A_c = cA_\ell$ and $B_c = c^{-1}B_\ell$. The objective in $c$ is strictly convex in $\log c$ and differentiating (33) yields $-2m\|B_\ell\|_F^2/c^3 + 2nc\|A_\ell\|_F^2 = 0$, giving (34). Substituting $c^\star$ into (33) yields (35). The divergence as $c \to 0$ or $c \to \infty$ gives the supremum. □

**Interpretation.** Even if one tunes the gauge once to reduce $\mathcal{E}_{Z_\ell}$, the optimizer can still drift to a different implicit scaling of $(A_\ell, B_\ell)$ over time. Therefore, under factor-space DP the effective intrinsic noise is not a fixed function of the intrinsic parameter $Z_\ell$ and is not directly controlled by $(\sigma, C, b)$ alone. PRISM removes this degree of freedom by defining the DP mechanism intrinsically on $T_{Z_\ell}\mathcal{M}_r$ (Corollary A.2).

### A.6. Tangent space orthogonal complement and orthogonal projection

**Lemma A.8** (Tangent space characterization). *Assume $A_\ell \in \mathbb{R}^{m \times r}$ and $B_\ell \in \mathbb{R}^{n \times r}$ have full column rank and $Z_\ell = A_\ell B_\ell^\top$. Then the tangent space of $\mathcal{M}_r$ at $Z_\ell$ is*

$$T_{Z_\ell}\mathcal{M}_r = \{\Delta A_\ell B_\ell^\top + A_\ell \Delta B_\ell^\top : \Delta A_\ell \in \mathbb{R}^{m \times r}, \Delta B_\ell \in \mathbb{R}^{n \times r}\}, \tag{36}$$

*which is Eq. (11) in the main text.*

*Proof.* Consider the smooth factorization map $\phi : \mathbb{R}_*^{m \times r} \times \mathbb{R}_*^{n \times r} \to \mathcal{M}_r$ defined by $\phi(A, B) = AB^\top$. For any perturbations $(\Delta A, \Delta B)$, the directional derivative at $(A_\ell, B_\ell)$ is

$$D\phi_{(A_\ell, B_\ell)}[\Delta A, \Delta B] = \Delta A\, B_\ell^\top + A_\ell\, \Delta B^\top.$$

Thus every matrix of the form $\Delta A_\ell B_\ell^\top + A_\ell \Delta B_\ell^\top$ arises as the derivative of the curve $t \mapsto (A_\ell + t\Delta A_\ell)(B_\ell + t\Delta B_\ell)^\top$ at $t = 0$, and hence lies in $T_{Z_\ell}\mathcal{M}_r$.

Conversely, any tangent vector $\Delta Z \in T_{Z_\ell}\mathcal{M}_r$ is the derivative of a smooth curve $t \mapsto Z(t) \in \mathcal{M}_r$ with $Z(0) = Z_\ell$. Because $A_\ell$ and $B_\ell$ are full column rank, rank-$r$ matrices near $Z_\ell$ admit factorizations $Z(t) = A(t)B(t)^\top$ with $A(t), B(t)$ smooth in $t$. Differentiating at $t = 0$ yields $\Delta Z = \dot{A}(0)B_\ell^\top + A_\ell \dot{B}(0)^\top$, proving (36). □

**Lemma A.9** (Tangent space orthogonal complement). *Let $Z_\ell = A_\ell B_\ell^\top$ with $\Pi_{A_\ell}, \Pi_{B_\ell}$ as in (12). Then*

$$(T_{Z_\ell}\mathcal{M}_r)^\perp = \{X \in \mathbb{R}^{m \times n} : \Pi_{A_\ell}X = 0 \text{ and } X\Pi_{B_\ell} = 0\} = \{\Pi_{A_\ell}^\perp Y \Pi_{B_\ell}^\perp : Y \in \mathbb{R}^{m \times n}\}. \tag{37}$$

*Proof.* Let $X \in (T_{Z_\ell}\mathcal{M}_r)^\perp$. For all $\Delta A_\ell, \Delta B_\ell$,

$$0 = \langle X, \Delta A_\ell B_\ell^\top + A_\ell \Delta B_\ell^\top \rangle = \langle XB_\ell, \Delta A_\ell \rangle + \langle X^\top A_\ell, \Delta B_\ell \rangle.$$

Hence $XB_\ell = 0$ and $X^\top A_\ell = 0$. Since $\Pi_{B_\ell}$ projects onto $\mathrm{col}(B_\ell)$, $XB_\ell = 0$ is equivalent to $X\Pi_{B_\ell} = 0$; similarly $X^\top A_\ell = 0$ is equivalent to $\Pi_{A_\ell}X = 0$. Conversely, if $\Pi_{A_\ell}X = 0$ and $X\Pi_{B_\ell} = 0$, the inner product above vanishes for all $\Delta A_\ell, \Delta B_\ell$. Finally, $X\Pi_{B_\ell} = 0$ implies $X = X\Pi_{B_\ell}^\perp$ and $\Pi_{A_\ell}X = 0$ implies $X = \Pi_{A_\ell}^\perp X$, hence $X = \Pi_{A_\ell}^\perp Y \Pi_{B_\ell}^\perp$ for $Y = X$. $\qquad\square$

### A.7. Gauge freedom in factor lifts and minimum-factor-norm representatives

The representation of an intrinsic tangent update $\Delta Z_\ell \in T_{Z_\ell}\mathcal{M}_r$ in factor space is not unique. This non-uniqueness is the differential analogue of the gauge symmetry $(A_\ell, B_\ell) \sim (A_\ell R, B_\ell R^{-\top})$. We record the basic "lift gauge" degrees of freedom and a canonical choice based on minimum factor norm.

**Lemma A.10** (Gauge degrees of freedom in factor lifts). *Let $Z_\ell = A_\ell B_\ell^\top$ and fix an intrinsic tangent matrix $\Delta Z_\ell \in T_{Z_\ell}\mathcal{M}_r$. If $(\Delta A_\ell, \Delta B_\ell)$ is any pair satisfying $\Delta Z_\ell = \Delta A_\ell B_\ell^\top + A_\ell \Delta B_\ell^\top$, then for any $\Omega \in \mathbb{R}^{r \times r}$,*

$$(\Delta A_\ell, \Delta B_\ell) \mapsto (\Delta A_\ell + A_\ell\Omega,\ \Delta B_\ell - B_\ell\Omega^\top) \tag{38}$$

*produces another valid lift of the* same *intrinsic update $\Delta Z_\ell$. In particular, the induced matrix $\Delta Z_\ell$ depends only on the intrinsic point $Z_\ell$ and not on the chosen factor lift.*

*Proof.* For any $\Omega$,

$$(\Delta A_\ell + A_\ell\Omega)B_\ell^\top + A_\ell(\Delta B_\ell - B_\ell\Omega^\top)^\top = \Delta A_\ell B_\ell^\top + A_\ell \Delta B_\ell^\top + A_\ell\Omega B_\ell^\top - A_\ell\Omega B_\ell^\top = \Delta Z_\ell.$$

$\qquad\square$

Equation (38) shows that there are infinitely many factor pairs that realize the *same* intrinsic tangent update. This is the differential version of the gauge symmetry (28). Depending on the optimizer implementation, it can be useful to pick a canonical representative of this equivalence class. One natural choice is the *minimum-factor-norm* lift, which can be computed by solving a small Sylvester equation of size $r \times r$.

**Proposition A.11** (Minimum-norm representative in a gauge class). *Assume $A_\ell$ and $B_\ell$ have full column rank so that $M = A_\ell^\top A_\ell \succ 0$ and $N = B_\ell^\top B_\ell \succ 0$. Fix any lift $(\Delta A_0, \Delta B_0)$ that realizes a given $\Delta Z_\ell$. Consider the gauge family*

$$\Delta A_\ell(\Omega) = \Delta A_0 + A_\ell\Omega, \qquad \Delta B_\ell(\Omega) = \Delta B_0 - B_\ell\Omega^\top.$$

*Then the unique minimizer of $f(\Omega) \triangleq \|\Delta A_\ell(\Omega)\|_F^2 + \|\Delta B_\ell(\Omega)\|_F^2$ is obtained by the unique solution $\Omega^\star$ to the Sylvester equation*

$$M\Omega + \Omega N = B_\ell^\top \Delta B_0 - A_\ell^\top \Delta A_0. \tag{39}$$

*The corresponding $(\Delta A_\ell(\Omega^\star), \Delta B_\ell(\Omega^\star))$ is the minimum-factor-norm lift of $\Delta Z_\ell$.*

*Proof.* Expand the quadratic objective:

$$\begin{aligned}
f(\Omega) &= \|\Delta A_0 + A_\ell\Omega\|_F^2 + \|\Delta B_0 - B_\ell\Omega^\top\|_F^2 \\
&= \|\Delta A_0\|_F^2 + \|\Delta B_0\|_F^2 + 2\,\mathrm{tr}(\Omega^\top A_\ell^\top \Delta A_0) - 2\,\mathrm{tr}(\Omega^\top \Delta B_0^\top B_\ell) \\
&\quad + \mathrm{tr}(\Omega^\top M\Omega) + \mathrm{tr}(\Omega N\Omega^\top),
\end{aligned}$$

where we used $\|A_\ell\Omega\|_F^2 = \mathrm{tr}(\Omega^\top M\Omega)$ and $\|B_\ell\Omega^\top\|_F^2 = \mathrm{tr}(\Omega N\Omega^\top)$. Taking the derivative and setting it to zero gives the first-order optimality condition

$$M\Omega + \Omega N = B_\ell^\top \Delta B_0 - A_\ell^\top \Delta A_0,$$

which is (39). Since $M$ and $N$ are positive definite, $f$ is strictly convex in $\Omega$ and the Sylvester equation has a unique solution, hence the minimizer is unique. $\qquad\square$

**Relation to "horizontal" conditions.** The optimality condition (39) is equivalent to orthogonality of the minimizer to the gauge (kernel) directions of the map $(\Delta A_\ell, \Delta B_\ell) \mapsto \Delta A_\ell B_\ell^\top + A_\ell \Delta B_\ell^\top$ under the Euclidean metric on factors. This is the standard minimum-norm property of least-squares solutions and is closely related to horizontal lifts in quotient-geometry treatments (Mishra et al., 2014).

### A.8. Gauge invariance of subspace projectors

**Lemma A.12** (Projectors are gauge invariant). *Let $(A'_\ell, B'_\ell) = (A_\ell R, B_\ell R^{-\top})$ for any invertible $R \in \mathbb{R}^{r \times r}$. Then $\Pi_{A'_\ell} = \Pi_{A_\ell}$ and $\Pi_{B'_\ell} = \Pi_{B_\ell}$.*

*Proof.* We prove the statement for $\Pi_{A_\ell}$; the argument for $\Pi_{B_\ell}$ is identical. Let $A'_\ell = A_\ell R$. Then $A'_\ell{}^\top A'_\ell = R^\top (A_\ell{}^\top A_\ell) R$. By Lemma A.13,

$$\left(A'_\ell{}^\top A'_\ell\right)^\dagger = \left(R^\top (A_\ell^\top A_\ell) R\right)^\dagger = R^{-1} (A_\ell^\top A_\ell)^\dagger R^{-\top}.$$

Therefore

$$\begin{aligned}
\Pi_{A'_\ell} &= A'_\ell \left(A'_\ell{}^\top A'_\ell\right)^\dagger A'_\ell{}^\top \\
&= A_\ell R \left(R^{-1} (A_\ell^\top A_\ell)^\dagger R^{-\top}\right) R^\top A_\ell^\top \\
&= A_\ell (A_\ell^\top A_\ell)^\dagger A_\ell^\top \\
&= \Pi_{A_\ell}.
\end{aligned}$$

$\square$

**Lemma A.13** (Pseudoinverse under congruence). *Let $X \succeq 0$ and let $R$ be invertible. Then $(R^\top X R)^\dagger = R^{-1} X^\dagger R^{-\top}$.*

*Proof.* Let $Y = R^\top X R$ and define $Y^* \triangleq R^{-1} X^\dagger R^{-\top}$. We verify the Moore–Penrose conditions: $YY^*Y = Y$, $Y^*YY^* = Y^*$, and both $YY^*$ and $Y^*Y$ are symmetric. All follow from substituting the definition of $Y$ and $Y^*$, using $RR^{-1} = \mathbf{I}$ and the Moore–Penrose conditions for $X$ and $X^\dagger$. $\square$

### A.9. A canonical factor lift of the tangent projection

This subsection verifies that the explicit lift defined in Eq. (14) indeed reproduces the orthogonal tangent projection (13). Non-uniqueness of factor lifts (and a minimum-norm choice) is discussed separately in Appendix A.7.

**Lemma A.14** (A convenient factor lift of the tangent projection). *Let $Z_\ell = A_\ell B_\ell^\top$ and let $G_{i,\ell} \triangleq \nabla_{Z_\ell} \ell_i$ with factor gradients $g_{A,i,\ell} = G_{i,\ell} B_\ell$ and $g_{B,i,\ell} = G_{i,\ell}^\top A_\ell$. Define $\Delta A_{i,\ell}, \Delta B_{i,\ell}$ by (14). Then the induced matrix update*

$$\Delta Z_{i,\ell} \triangleq \Delta A_{i,\ell} B_\ell^\top + A_\ell \Delta B_{i,\ell}^\top \tag{40}$$

*equals the tangent projection $P_{A_\ell, B_\ell}(G_{i,\ell})$.*

*Proof.* **Step 1 (matching the matrix update).** Using $B_\ell N^\dagger B_\ell^\top = \Pi_{B_\ell}$ and $A_\ell M^\dagger A_\ell^\top = \Pi_{A_\ell}$,

$$g_{A,i,\ell} N^\dagger = G_{i,\ell}(B_\ell N^\dagger) = G_{i,\ell} \Pi_{B_\ell}, \qquad g_{B,i,\ell} M^\dagger = G_{i,\ell}^\top (A_\ell M^\dagger) = G_{i,\ell}^\top \Pi_{A_\ell}.$$

Hence

$$\begin{aligned}
\Delta A_{i,\ell} B_\ell^\top &= \left(G_{i,\ell} \Pi_{B_\ell} - \tfrac{1}{2} \Pi_{A_\ell}(G_{i,\ell}\Pi_{B_\ell})\right) B_\ell^\top = G_{i,\ell}\Pi_{B_\ell} - \tfrac{1}{2}\Pi_{A_\ell} G_{i,\ell}\Pi_{B_\ell}, \\
A_\ell \Delta B_{i,\ell}^\top &= A_\ell \left(G_{i,\ell}^\top \Pi_{A_\ell} - \tfrac{1}{2}\Pi_{B_\ell}(G_{i,\ell}^\top \Pi_{A_\ell})\right)^\top = \Pi_{A_\ell} G_{i,\ell} - \tfrac{1}{2}\Pi_{A_\ell} G_{i,\ell}\Pi_{B_\ell}.
\end{aligned}$$

Summing yields $\Delta Z_{i,\ell} = \Pi_{A_\ell} G_{i,\ell} + G_{i,\ell}\Pi_{B_\ell} - \Pi_{A_\ell} G_{i,\ell}\Pi_{B_\ell} = P_{A_\ell, B_\ell}(G_{i,\ell})$. $\square$

## A.10. Frobenius norm formula for tangent updates

**Proposition A.15** (Frobenius norm of a factorized update)**.** *Let $Z_\ell = A_\ell B_\ell^\top$ and let $\Delta Z_\ell = \Delta A_\ell B_\ell^\top + A_\ell \Delta B_\ell^\top$ for arbitrary $\Delta A_\ell \in \mathbb{R}^{m \times r}$ and $\Delta B_\ell \in \mathbb{R}^{n \times r}$. With $M = A_\ell^\top A_\ell$ and $N = B_\ell^\top B_\ell$,*

$$\|\Delta Z_\ell\|_F^2 = \text{tr}(\Delta A_\ell^\top \Delta A_\ell \, N) + \text{tr}(\Delta B_\ell^\top \Delta B_\ell \, M) + 2\,\text{tr}\big((A_\ell^\top \Delta A_\ell)(B_\ell^\top \Delta B_\ell)\big). \tag{41}$$

*Proof of Proposition A.15.* Expand $\|\Delta Z_\ell\|_F^2 = \langle \Delta A_\ell B_\ell^\top + A_\ell \Delta B_\ell^\top, \Delta A_\ell B_\ell^\top + A_\ell \Delta B_\ell^\top \rangle$ and use cyclicity of trace:

$$\|\Delta A_\ell B_\ell^\top\|_F^2 = \text{tr}(\Delta A_\ell^\top \Delta A_\ell \, B_\ell^\top B_\ell) = \text{tr}(\Delta A_\ell^\top \Delta A_\ell \, N), \quad \|A_\ell \Delta B_\ell^\top\|_F^2 = \text{tr}(\Delta B_\ell^\top \Delta B_\ell \, A_\ell^\top A_\ell) = \text{tr}(\Delta B_\ell^\top \Delta B_\ell \, M),$$

and $\langle \Delta A_\ell B_\ell^\top, A_\ell \Delta B_\ell^\top \rangle = \text{tr}((A_\ell^\top \Delta A_\ell)(B_\ell^\top \Delta B_\ell))$. Summing yields (41). $\square$

## A.11. Specialization to rank-1 per-example gradients

**Lemma A.16** (Norm for rank-1 gradients)**.** *Let $G_{i,\ell} = uv^\top$ and let $\Delta Z_{i,\ell} = P_{A_\ell, B_\ell}(G_{i,\ell})$. Define $\widehat{u} = \Pi_{A_\ell} u$ and $\widehat{v} = \Pi_{B_\ell} v$. Then*

$$\|\Delta Z_{i,\ell}\|_F^2 = \|\widehat{u}\|_2^2 \|v\|_2^2 + \|u\|_2^2 \|\widehat{v}\|_2^2 - \|\widehat{u}\|_2^2 \|\widehat{v}\|_2^2. \tag{42}$$

*Proof.* Use $\Delta Z_{i,\ell} = \widehat{u}v^\top + u\widehat{v}^\top - \widehat{u}\widehat{v}^\top$ and the identities $\|ab^\top\|_F^2 = \|a\|_2^2\|b\|_2^2$ and $\langle ab^\top, cd^\top \rangle = (a^\top c)(b^\top d)$. Expanding and simplifying yields (42). $\square$

## A.12. Factorized tangent noise equals a projected dense Gaussian

For convenience, define the *whitened* factors

$$\widehat{A}_\ell \triangleq A_\ell M^{-1/2}, \qquad \widehat{B}_\ell \triangleq B_\ell N^{-1/2}, \tag{43}$$

so that $\widehat{A}_\ell^\top \widehat{A}_\ell = \widehat{B}_\ell^\top \widehat{B}_\ell = I_r$ and $\Pi_{A_\ell} = \widehat{A}_\ell \widehat{A}_\ell^\top, \Pi_{B_\ell} = \widehat{B}_\ell \widehat{B}_\ell^\top$ when $A_\ell, B_\ell$ have full column rank.

**Lemma A.17** (Distributional equivalence of the sampler)**.** *Let $U \in \mathbb{R}^{m \times r}$ and $V \in \mathbb{R}^{n \times r}$ have i.i.d. standard normal entries and define $\widehat{A}_\ell, \widehat{B}_\ell$ as in (43). Then*

$$(\mathbf{I} - \Pi_{A_\ell}) U \, \widehat{B}_\ell^\top + \widehat{A}_\ell V^\top \stackrel{d}{=} P_{A_\ell, B_\ell}(\Xi_\ell),$$

*where $\Xi_\ell$ has i.i.d. $\mathcal{N}(0,1)$ entries.*

*Proof.* We show that $\text{vec}(\Delta Z_{\text{noise}})$ is a zero-mean Gaussian with covariance equal to (46). Using the identity $\text{vec}(XY^\top) = (Y \otimes \mathbf{I})\text{vec}(X)$, we have

$$\text{vec}\big((\mathbf{I} - \Pi_{A_\ell})U\widehat{B}_\ell^\top\big) = (\widehat{B}_\ell \otimes (\mathbf{I} - \Pi_{A_\ell})) \, \text{vec}(U),$$
$$\text{vec}(\widehat{A}_\ell V^\top) = (\mathbf{I} \otimes \widehat{A}_\ell) \, \text{vec}(V).$$

Since $\text{vec}(U)$ and $\text{vec}(V)$ are independent standard Gaussians, their images under fixed linear maps are independent Gaussians and the covariances add:

$$\text{Cov}\big[\text{vec}(\Delta Z_{\text{noise}})\big] = (\widehat{B}_\ell \widehat{B}_\ell^\top \otimes (\mathbf{I} - \Pi_{A_\ell})) + (\mathbf{I} \otimes \widehat{A}_\ell \widehat{A}_\ell^\top).$$

Substituting $\widehat{B}_\ell \widehat{B}_\ell^\top = \Pi_{B_\ell}$ and $\widehat{A}_\ell \widehat{A}_\ell^\top = \Pi_{A_\ell}$ yields

$$\text{Cov}\big[\text{vec}(\Delta Z_{\text{noise}})\big] = (\Pi_{B_\ell} \otimes \mathbf{I}) - (\Pi_{B_\ell} \otimes \Pi_{A_\ell}) + (\mathbf{I} \otimes \Pi_{A_\ell}),$$

which matches (46). Therefore $\Delta Z_{\text{noise}} \stackrel{d}{=} P_{A_\ell, B_\ell}(\Xi_\ell)$. $\square$

## A.13. Low-dimensional noise sampler

The intrinsic PRISM noise for module $\ell$ is $N_{Z_\ell} = \tau \mathcal{P}_{A_\ell, B_\ell}(\Xi_\ell)$ with $\Xi_\ell \sim \mathcal{N}(0, I_{m \times n})$ and $\tau = \sigma C/b$ (Theorem 3.1). Directly sampling the dense matrix $\Xi_\ell$ is unnecessary.

Let $U \in \mathbb{R}^{m \times r}$ and $V \in \mathbb{R}^{n \times r}$ have i.i.d. $\mathcal{N}(0, 1)$ entries, and define

$$\Xi_{A,\ell} = (I - \Pi_{A_\ell}) U N^{-1/2}, \qquad \Xi_{B,\ell} = V M^{-1/2},$$

where $M = A_\ell^\top A_\ell$ and $N = B_\ell^\top B_\ell$. Then the intrinsic perturbation induced by these factor noises is

$$\Xi_{A,\ell} B_\ell^\top + A_\ell \Xi_{B,\ell}^\top,$$

which matches the low-dimensional sampler in Eq. (19) (with $(U, V)$ corresponding to $(\Omega_{A,\ell}, \Omega_{B,\ell})$). By Lemma A.17 (Appendix A.12), this intrinsic perturbation is distributed exactly as $\mathcal{P}_{A_\ell, B_\ell}(\Xi_\ell)$.

Computationally, this reduces random number generation from $O(mn)$ to $O((m + n)r)$ per module. Stable computation of $\Pi_{A_\ell}, \Pi_{B_\ell}$ and the inverse square roots $M^{-1/2}, N^{-1/2}$ is discussed in Appendix A.14. The resulting intrinsic noise distribution is also invariant to the choice of factorization, as formalized in Appendix A.15.

## A.14. Stable computation of projectors and orthonormal bases

For numerical stability, it is often preferable to compute the projectors $\Pi_{A_\ell}, \Pi_{B_\ell}$ and orthonormal bases of $\mathrm{col}(A_\ell)$ and $\mathrm{col}(B_\ell)$ without forming Gram-matrix inverses explicitly. We record simple equivalences.

**Lemma A.18** (Projector from a thin QR factorization). *Assume $A_\ell \in \mathbb{R}^{m \times r}$ has full column rank and let $A_\ell = Q_A R_A$ be a thin QR factorization with $Q_A^\top Q_A = \mathbf{I}_r$ and $R_A$ invertible. Then the orthogonal projector onto $\mathrm{col}(A_\ell)$ satisfies $\Pi_{A_\ell} = Q_A Q_A^\top$. An analogous statement holds for $B_\ell$.*

*Proof.* Since $A_\ell^\top A_\ell = R_A^\top R_A$, we have

$$\Pi_{A_\ell} = A_\ell (A_\ell^\top A_\ell)^{-1} A_\ell^\top = Q_A R_A (R_A^\top R_A)^{-1} R_A^\top Q_A^\top = Q_A Q_A^\top,$$

because $R_A (R_A^\top R_A)^{-1} R_A^\top = \mathbf{I}_r$ when $R_A$ is invertible. $\qquad\square$

**Lemma A.19** (Noise sampling is invariant to the choice of orthonormal bases). *Let $Q_A, Q'_A \in \mathbb{R}^{m \times r}$ be orthonormal bases of the same subspace $\mathrm{col}(A_\ell)$ and let $Q_B, Q'_B \in \mathbb{R}^{n \times r}$ be orthonormal bases of $\mathrm{col}(B_\ell)$. Define $\Pi_{A_\ell} = Q_A Q_A^\top = Q'_A Q'^\top_A$ and $\Pi_{B_\ell} = Q_B Q_B^\top = Q'_B Q'^\top_B$. If $U \in \mathbb{R}^{m \times r}$ and $V \in \mathbb{R}^{n \times r}$ have i.i.d. $\mathcal{N}(0, 1)$ entries, then*

$$(\mathbf{I} - \Pi_{A_\ell}) U Q_B^\top + Q_A V^\top \overset{d}{=} (\mathbf{I} - \Pi_{A_\ell}) U Q'^\top_B + Q'_A V^\top.$$

*Proof.* Because $Q_A$ and $Q'_A$ are orthonormal bases of the same subspace, there exists an orthogonal matrix $O_A \in \mathbb{R}^{r \times r}$ such that $Q'_A = Q_A O_A$. Similarly, $Q'_B = Q_B O_B$ for some orthogonal $O_B$. Then

$$(\mathbf{I} - \Pi_{A_\ell}) U Q'^\top_B + Q'_A V^\top = (\mathbf{I} - \Pi_{A_\ell}) U O_B^\top Q_B^\top + Q_A O_A V^\top.$$

Since $U$ and $V$ are i.i.d. standard Gaussian matrices, $U O_B^\top \overset{d}{=} U$ and $O_A V \overset{d}{=} V$. The claim follows. $\qquad\square$

**Regularization.** When Gram matrices are nearly singular, one can obtain stable orthonormal bases via QR/SVD and use Lemma A.18 to form $\Pi_{A_\ell}, \Pi_{B_\ell}$. If one instead regularizes Gram inverses directly (e.g., spectral truncation or damping), the resulting operators are still deterministic functions of $(A_\ell, B_\ell)$ and therefore DP-safe by post-processing, but may not preserve *exact* gauge invariance unless the regularization is itself defined in a gauge-consistent way.

## A.15. Gauge invariance of the factorized noise sampler

We recall the factorized sampler used to generate the tangent noise lifts in Eq. (18):

$$\Delta A_{\mathrm{noise}} = (I - \Pi_{A_\ell}) U N^{-1/2}, \qquad \Delta B_{\mathrm{noise}} = V M^{-1/2}, \tag{44}$$

where $U \in \mathbb{R}^{m \times r}$ and $V \in \mathbb{R}^{n \times r}$ have i.i.d. $\mathcal{N}(0, 1)$ entries, and $M = A_\ell^\top A_\ell$, $N = B_\ell^\top B_\ell$. The induced intrinsic perturbation is $\Delta Z_{\mathrm{noise}} = \Delta A_{\mathrm{noise}} B_\ell^\top + A_\ell \Delta B_{\mathrm{noise}}^\top$.

The intrinsic mechanism (18) is gauge invariant by construction. Here we make explicit that the *implementation-level* sampler (44) inherits the same invariance: regardless of which factorization of $Z_\ell$ is used internally, the induced distribution on the intrinsic noise $\Delta Z_{\mathrm{noise}}$ is unchanged.

**Proposition A.20** (Sampler invariance under gauge transforms). *Let $(A_\ell', B_\ell') = (A_\ell R, B_\ell R^{-\top})$ for some $R \in \mathrm{GL}(r)$. Construct $\Delta A_{\mathrm{noise}}$ and $\Delta B_{\mathrm{noise}}$ from $(A_\ell, B_\ell)$ via (44), and construct $\Delta A_{\ell,\mathrm{noise}}'$ and $\Delta B_{\ell,\mathrm{noise}}'$ from $(A_\ell', B_\ell')$ via the same formula (with the corresponding projectors and Gram matrices). Then the induced intrinsic noises*

$$\Delta Z_{\mathrm{noise}} = \Delta A_{\mathrm{noise}} B_\ell^\top + A_\ell (\Delta B_{\mathrm{noise}})^\top, \qquad \Delta Z_{\mathrm{noise}}' = \Delta A_{\ell,\mathrm{noise}}' (B_\ell')^\top + A_\ell' (\Delta B_{\ell,\mathrm{noise}}')^\top.$$

*have the same distribution.*

*Proof.* By Lemma A.17, both $\Delta Z_{\mathrm{noise}}$ and $\Delta Z_{\mathrm{noise}}'$ are distributed as $P_{A_\ell, B_\ell}(\Xi_\ell)$ and $P_{A', B'}(\Xi_\ell)$, respectively, for a dense standard Gaussian $\Xi_\ell$. By Lemma A.12, the subspace projectors are gauge invariant and thus $P_{A', B'} = P_{A_\ell, B_\ell}$. Therefore $P_{A', B'}(\Xi_\ell)$ and $P_{A_\ell, B_\ell}(\Xi_\ell)$ have identical distributions. $\square$

**Contrast with naive factor noise.** If one instead adds i.i.d. Gaussian noise directly to $\Delta A_\ell$ and $\Delta B_\ell$ without the whitening and projection factors in (44), the induced intrinsic perturbation depends on the chosen gauge through $\|A_\ell\|_F$ and $\|B_\ell\|_F$ (Proposition 2.2). The role of $M^{-1/2}$ and $N^{-1/2}$ in (44) is precisely to compensate for this coordinate dependence and yield an isotropic Gaussian in the intrinsic tangent space.

## A.16. Proof of Theorem 3.1

*Proof of Theorem 3.1.* Let $\Xi_\ell \sim \mathcal{N}(0, I_{m \times n})$ be a dense standard Gaussian and write $N_{Z_\ell} = \tau \mathcal{P}_{A_\ell, B_\ell}(\Xi_\ell)$ with $\tau = \sigma C / b$.

**Gaussianity and support.** Vectorizing gives $\mathrm{vec}(\Xi_\ell) \sim \mathcal{N}(0, I_{mn})$ and $\mathrm{vec}(\mathcal{P}_{A_\ell, B_\ell}(\Xi_\ell)) = \mathbf{P}\,\mathrm{vec}(\Xi_\ell)$, where $\mathbf{P}$ is the matrix representation of the orthogonal projector $\mathcal{P}_{A_\ell, B_\ell}$ under $\mathrm{vec}(\cdot)$. Since $\mathbf{P}$ is linear, symmetric, and idempotent, $\mathbf{P}\,\mathrm{vec}(\Xi_\ell)$ is Gaussian with covariance $\mathbf{P}$ and is supported on $\mathrm{range}(\mathbf{P})$, which corresponds to the tangent subspace $T_{Z_\ell}\mathcal{M}_r$ (Eq. (13)).

**Isotropy on the tangent space.** Because the covariance equals the orthogonal projector onto $T_{Z_\ell}\mathcal{M}_r$, the distribution is isotropic within that subspace; a self-contained verification is given in Lemma A.21 (Appendix A.17).

**Expected energy.** Using $\|X\|_F^2 = \|\mathrm{vec}(X)\|_2^2$ and $\mathbb{E}\|G\|_2^2 = \mathrm{tr}(\mathrm{Cov}[G])$ for a zero-mean Gaussian vector $G$, we obtain

$$\mathbb{E}\big\|\mathcal{P}_{A_\ell, B_\ell}(\Xi_\ell)\big\|_F^2 = \mathbb{E}\big\|\mathbf{P}\mathrm{vec}(\Xi_\ell)\big\|_2^2 = \mathrm{tr}(\mathbf{P}) = \mathrm{rank}(\mathbf{P}) = \dim(T_{Z_\ell}\mathcal{M}_r) = r(m + n - r),$$

which is Eq. (20). Multiplying by $\tau^2$ yields $\mathbb{E}\|N_{Z_\ell}\|_F^2 = \tau^2 r(m + n - r)$ and thus Eq. (21).

**Gauge invariance.** Finally, $\mathcal{P}_{A_\ell, B_\ell}$ depends only on $\Pi_{A_\ell}$ and $\Pi_{B_\ell}$ (Eq. (13)), and these projectors are invariant under the gauge transform $(A_\ell, B_\ell) \mapsto (A_\ell R, B_\ell R^{-\top})$ by Lemma A.12 (Appendix A.8). $\square$

## A.17. Isotropy of the projected Gaussian on the tangent space

**Lemma A.21** (Isotropy within $T_{Z_\ell}\mathcal{M}_r$). *Let $\Xi_\ell \in \mathbb{R}^{m \times n}$ have i.i.d. $\mathcal{N}(0, 1)$ entries and let $P_{A_\ell, B_\ell}$ be the orthogonal projector onto $T_{Z_\ell}\mathcal{M}_r$. For any $U, V \in T_{Z_\ell}\mathcal{M}_r$,*

$$\mathbb{E}\big[\langle U, P_{A_\ell, B_\ell}(\Xi_\ell)\rangle \langle V, P_{A_\ell, B_\ell}(\Xi_\ell)\rangle\big] = \langle U, V\rangle. \tag{45}$$

*Equivalently, $P_{A_\ell, B_\ell}(\Xi_\ell)$ is an isotropic Gaussian in the tangent space under the Frobenius inner product.*

*Proof.* Because $P_{A_\ell, B_\ell}$ is an orthogonal projector, it is self-adjoint: $\langle U, P_{A_\ell, B_\ell}(X)\rangle = \langle P_{A_\ell, B_\ell}(U), X\rangle$ for all $U, X$. For $U \in T_{Z_\ell}\mathcal{M}_r$, $P_{A_\ell, B_\ell}(U) = U$. Therefore $\langle U, P_{A_\ell, B_\ell}(\Xi_\ell)\rangle = \langle U, \Xi_\ell\rangle$ and similarly for $V$. Since $\Xi_\ell$ has i.i.d. standard normal entries, $\langle U, \Xi_\ell\rangle$ is a centered Gaussian with variance $\|U\|_F^2$, and

$$\mathbb{E}[\langle U, \Xi_\ell\rangle \langle V, \Xi_\ell\rangle] = \langle U, V\rangle.$$

$\square$

### A.18. Projected Gaussian covariance and intrinsic dimension

**Lemma A.22** (Covariance of a projected dense Gaussian). *Let $\Xi_\ell \in \mathbb{R}^{m \times n}$ have i.i.d. $\mathcal{N}(0,1)$ entries and let $Z_\ell = A_\ell B_\ell^\top \in \mathcal{M}_r$. Then $P_{A_\ell,B_\ell}(\Xi_\ell)$ is a centered Gaussian supported on $T_{Z_\ell}\mathcal{M}_r$ with vectorized covariance*

$$\mathrm{Cov}\big[\mathrm{vec}\big(P_{A_\ell,B_\ell}(\Xi_\ell)\big)\big] = (\mathbf{I}_n \otimes \Pi_{A_\ell}) + (\Pi_{B_\ell} \otimes \mathbf{I}_m) - (\Pi_{B_\ell} \otimes \Pi_{A_\ell}). \tag{46}$$

*If $A_\ell$ and $B_\ell$ have full column rank, the covariance operator in Eq. (46) is an orthogonal projector of rank $r(m+n-r)$, which implies $\mathbb{E}\|P_{A_\ell,B_\ell}(\Xi_\ell)\|_F^2 = r(m+n-r)$ (Eq. (20)).*

*Proof of Lemma A.22.* Write $P(\Xi_\ell) = \Pi_{A_\ell}\Xi_\ell + \Xi_\ell\Pi_{B_\ell} - \Pi_{A_\ell}\Xi_\ell\Pi_{B_\ell}$ and apply vectorization: $\mathrm{vec}(\Pi_{A_\ell}\Xi_\ell) = (\mathbf{I} \otimes \Pi_{A_\ell})\mathrm{vec}(\Xi_\ell)$, $\mathrm{vec}(\Xi_\ell\Pi_{B_\ell}) = (\Pi_{B_\ell} \otimes \mathbf{I})\mathrm{vec}(\Xi_\ell)$, $\mathrm{vec}(\Pi_{A_\ell}\Xi_\ell\Pi_{B_\ell}) = (\Pi_{B_\ell} \otimes \Pi_{A_\ell})\mathrm{vec}(\Xi_\ell)$. Thus

$$\mathrm{vec}(P(\Xi_\ell)) = \big(\mathbf{I} \otimes \Pi_{A_\ell} + \Pi_{B_\ell} \otimes \mathbf{I} - \Pi_{B_\ell} \otimes \Pi_{A_\ell}\big)\mathrm{vec}(\Xi_\ell).$$

Since $\mathrm{vec}(\Xi_\ell) \sim \mathcal{N}(0,\mathbf{I})$, the covariance is (46). When $A_\ell, B_\ell$ are full column rank, this covariance is an orthogonal projector with rank $r(m+n-r)$ (Edelman et al., 1998). The expected squared norm equals the trace, giving (20). $\quad\square$

### A.19. Concentration of the effective intrinsic noise in PRISM

Because $P_{A_\ell,B_\ell}(\Xi_\ell)$ is an isotropic Gaussian in the tangent space (Lemma A.21), its Frobenius norm concentrates sharply. This provides high-probability control beyond the expectation in (20).

**Lemma A.23** (Chi-square form). *Assume $A_\ell$ and $B_\ell$ are full column rank and let $d \triangleq \dim(T_{Z_\ell}\mathcal{M}_r) = r(m+n-r)$. Let $\Xi_\ell \in \mathbb{R}^{m \times n}$ have i.i.d. $\mathcal{N}(0,1)$ entries and set $G = P_{A_\ell,B_\ell}(\Xi_\ell)$. Then $\|G\|_F^2$ has a chi-square distribution with $d$ degrees of freedom:*

$$\|G\|_F^2 \sim \chi_d^2. \tag{47}$$

*Proof.* Let $\{E_1, \ldots, E_d\}$ be any orthonormal basis of $T_{Z_\ell}\mathcal{M}_r$ under $\langle \cdot, \cdot \rangle$. Since $P_{A_\ell,B_\ell}$ is the orthogonal projector onto $T_{Z_\ell}\mathcal{M}_r$, we may write $G = \sum_{k=1}^d \langle \Xi_\ell, E_k \rangle E_k$. By orthonormality and independence of Gaussian linear functionals, the coefficients $\{\langle \Xi_\ell, E_k \rangle\}_{k=1}^d$ are i.i.d. $\mathcal{N}(0,1)$. Therefore $\|G\|_F^2 = \sum_{k=1}^d \langle \Xi_\ell, E_k \rangle^2$ is chi-square with $d$ degrees of freedom. $\quad\square$

**Proposition A.24** (High-probability bound for PRISM noise). *Let $\mathcal{N}_{Z_\ell} = \frac{\sigma C}{b}P_{A_\ell,B_\ell}(\Xi_\ell)$ be the intrinsic Gaussian perturbation in DP-PRISM. Let $d = r(m+n-r)$. Then for any $\delta \in (0,1)$,*

$$\Pr\bigg(\|\mathcal{N}_{Z_\ell}\|_F \leq \frac{\sigma C}{b}\sqrt{d + 2\sqrt{d\log(1/\delta)} + 2\log(1/\delta)}\bigg) \geq 1 - \delta. \tag{48}$$

*Proof.* By Lemma A.23, $\|\mathcal{N}_{Z_\ell}\|_F^2 = (\sigma C/b)^2 X$ where $X \sim \chi_d^2$. A standard chi-square concentration inequality gives $\Pr(X - d \geq 2\sqrt{dt} + 2t) \leq e^{-t}$ for all $t \geq 0$. Setting $t = \log(1/\delta)$ yields (48). $\quad\square$

**Remark.** Proposition A.24 shows that the realized intrinsic noise magnitude in PRISM concentrates around its mean $\mathcal{E}_{Z_\ell}$ with relative fluctuations $O(1/\sqrt{d})$. This is useful when interpreting the privacy–utility trade-off in large layers, where $d = r(m+n-r)$ is large.

### A.20. Retraction and rank-$r$ approximation

We justify Proposition 3.2 and record standard facts about truncated SVD retractions.

**Lemma A.25** (Eckart–Young–Mirsky theorem). *Let $X \in \mathbb{R}^{m \times n}$ have singular values $s_1 \geq \cdots \geq s_{\min(m,n)}$. Let $X_r$ be the truncated SVD keeping the top $r$ singular values. Then $X_r$ is a best rank-$r$ approximation in Frobenius norm:*

$$X_r \in \arg\min_{\mathrm{rank}(Y)\leq r} \|X - Y\|_F, \qquad \|X - X_r\|_F^2 = \sum_{k>r} s_k^2.$$

*Proof.* See (Eckart & Young, 1936; Mirsky, 1960). $\quad\square$

*Proof of Proposition 3.2.* Let $X_\eta = Z - \eta\Delta Z$. Define

$$Y_\eta = (A - \eta\Delta A)(B - \eta\Delta B)^\top.$$

Then $\mathrm{rank}(Y_\eta) \le r$, and

$$Y_\eta = Z - \eta(\Delta AB^\top + A\Delta B^\top) + \eta^2\Delta A\Delta B^\top = X_\eta + \eta^2\Delta A\Delta B^\top.$$

By Lemma A.25, $\mathrm{Retr}_r(X_\eta)$ is a best rank-$r$ approximation to $X_\eta$. Since $Y_\eta$ is a rank-$r$ candidate,

$$\|X_\eta - \mathrm{Retr}_r(X_\eta)\|_F \le \|X_\eta - Y_\eta\|_F = \eta^2\|\Delta A\Delta B^\top\|_F.$$

This proves Eq. (23). Finally, $\|\Delta A\Delta B^\top\|_F \le \|\Delta A\|_F\|\Delta B\|_F$, giving the stated second-order distortion bound. $\qquad\square$

### A.21. Absence of bilinear second-order DP noise in PRISM

**Lemma A.26** (PRISM noise is additive in the intrinsic parameter)**.** *Consider one PRISM iteration for a single LoRA module. Conditioned on the minibatch and on the Gaussian randomness used in* (18)*, the update takes the intrinsic additive form*

$$Z_\ell^+ = \mathrm{Retr}_r\left(Z_\ell - \eta\left(\bar{\Delta Z}_\ell + \tfrac{\sigma C}{b}P_{A_\ell,B_\ell}(\Xi_\ell)\right)\right),$$

*where $\bar{\Delta Z}_\ell$ is the clipped mean tangent update. In particular, the only randomness in the intrinsic update is the* linear *Gaussian term $P_{A_\ell,B_\ell}(\Xi_\ell)$; there is no bilinear product of independent noises analogous to $\xi_{A,\ell}\xi_{B,\ell}^\top$ in* (7)*.*

*Proof.* This is immediate from the definition of PRISM in (18)–(22) and the linearity of $P_{A_\ell,B_\ell}$. Retraction $\mathrm{Retr}_r$ and any subsequent factorization/gauge alignment are deterministic post-processing steps. $\qquad\square$

### A.22. Proof of Theorem 3.3

*Proof of Theorem 3.3.* Part (i) is Lemma A.12. Part (ii) follows from Proposition A.3 since $P_{A_\ell,B_\ell}$ depends only on $(\Pi_{A_\ell}, \Pi_{B_\ell})$. For (iii), $P_{A_\ell,B_\ell}(\Xi_\ell)$ is a measurable function of $(\Pi_{A_\ell}, \Pi_{B_\ell}, \Xi_\ell)$ and $(\Pi_{A_\ell}, \Pi_{B_\ell})$ are unchanged under gauge transformations, hence the induced distribution is unchanged. Retraction, refactorization, and gauge alignment are deterministic maps of the intrinsic quantities, so they preserve gauge invariance. $\qquad\square$

### A.23. Gaussian mechanism on a linear subspace

DP analyses are often stated for outputs in $\mathbb{R}^d$ with full-dimensional Gaussian noise. PRISM adds Gaussian noise supported on the tangent subspace $T_{Z_\ell}\mathcal{M}_r$. This is still a standard Gaussian mechanism once the output space is identified with the subspace.

**Lemma A.27** (Gaussian mechanism restricted to a subspace)**.** *Let $S \subseteq \mathbb{R}^d$ be a linear subspace with orthogonal projector $\Pi_S$. Let $f : \mathcal{D} \mapsto S$ be a function with $\ell_2$ sensitivity at most $\Delta$: $\|f(\mathcal{D}) - f(\mathcal{D}')\|_2 \le \Delta$ for all adjacent $\mathcal{D}, \mathcal{D}'$. Let $g \sim \mathcal{N}(0, \mathbf{I}_d)$ and define the mechanism*

$$\mathcal{M}(\mathcal{D}) \triangleq f(\mathcal{D}) + \sigma\Delta\,\Pi_S g.$$

*Then $\mathcal{M}$ is $(\varepsilon, \delta)$-DP for the same $(\varepsilon, \delta)$ guarantee as the standard Gaussian mechanism in dimension $\dim(S)$ (with noise multiplier $\sigma$).*

*Proof.* Let $k = \dim(S)$ and let $U \in \mathbb{R}^{d \times k}$ have orthonormal columns spanning $S$ so that $\Pi_S = UU^\top$. Write $f(\mathcal{D}) = U\alpha(\mathcal{D})$ for some $\alpha(\mathcal{D}) \in \mathbb{R}^k$. Then $\Pi_S g = UU^\top g \overset{d}{=} Uh$ where $h \sim \mathcal{N}(0, \mathbf{I}_k)$. Therefore $\mathcal{M}(\mathcal{D}) \overset{d}{=} U(\alpha(\mathcal{D}) + \sigma\Delta h)$. Since $U$ is an isometry on $S$, the DP guarantee for $\alpha(\mathcal{D}) + \sigma\Delta h$ (a standard Gaussian mechanism in $\mathbb{R}^k$) transfers directly to $\mathcal{M}$. $\qquad\square$

### A.24. Procrustes alignment is a gauge transform

**Lemma A.28** (Orthogonal alignment is gauge preserving)**.** *Let $Z_\ell = A_\ell B_\ell^\top$ with $A_\ell, B_\ell$ full column rank and let $Q$ be orthogonal. Then $(A_\ell', B_\ell') = (A_\ell Q, B_\ell Q)$ satisfies $A_\ell' B_\ell'^\top = Z_\ell$ and leaves $\Pi_{A_\ell}, \Pi_{B_\ell}$ unchanged.*

*Proof.* $A'_\ell {B'_\ell}^\top = (A_\ell Q)(B_\ell Q)^\top = A_\ell QQ^\top B_\ell^\top = A_\ell B_\ell^\top$. For the projector,

$$A'_\ell \big((A'_\ell)^\top A'_\ell\big)^\dagger (A'_\ell)^\top = A_\ell Q \big(Q^\top A_\ell^\top A_\ell Q\big)^\dagger Q^\top A_\ell^\top = A_\ell (A_\ell^\top A_\ell)^\dagger A_\ell^\top = \Pi_{A_\ell},$$

because $\big(Q^\top X Q\big)^\dagger = Q^\top X^\dagger Q$ for orthogonal $Q$.

$\square$

## A.25. DP guarantee details

We provide a proof of Theorem 3.4.

*Proof of Theorem 3.4.* Write the per-example intrinsic tangent update (concatenated across all LoRA modules) as $\Delta \mathbf{Z}_{\ell i} \in \mathbb{T}$, where $\mathbb{T}$ denotes the direct-sum tangent space equipped with the Frobenius inner product. PRISM applies intrinsic clipping (Eq. (16)) to obtain $\Delta \tilde{\mathbf{Z}}_{\ell i} = \alpha_i \, \Delta \mathbf{Z}_{\ell i}$ with $\|\Delta \tilde{\mathbf{Z}}_{\ell i}\|_F \le C$. Hence the $\ell_2$ sensitivity of the minibatch average is bounded by

$$\left\| \frac{1}{b} \sum_{i=1}^b \Delta \tilde{\mathbf{Z}}_{\ell i}(\mathcal{D}) - \frac{1}{b} \sum_{i=1}^b \Delta \tilde{\mathbf{Z}}_{\ell i}(\mathcal{D}') \right\|_F \le \frac{C}{b}$$

for any adjacent datasets $\mathcal{D}, \mathcal{D}'$.

Next, PRISM adds Gaussian noise of standard deviation $\sigma C / b$ in $\mathbb{T}$. Concretely, each module samples a dense Gaussian matrix and applies the orthogonal tangent projector (Eq. (13)), so the resulting noise is a Gaussian restricted to a linear subspace. By Lemma A.27, the released vector

$$\widehat{\Delta \mathbf{Z}}_\ell = \frac{1}{b} \sum_{i=1}^b \Delta \tilde{\mathbf{Z}}_{\ell i} + \frac{\sigma C}{b} \, \mathbf{G}, \qquad \mathbf{G} \sim \mathcal{N}(0, \Pi_{\mathbb{T}}),$$

is an instance of the Gaussian mechanism with sensitivity $C/b$.

Finally, under Poisson subsampling with rate $q = b/N$, each iteration is a *subsampled* Gaussian mechanism. The overall $(\varepsilon, \delta)$ guarantee after $T$ steps follows from standard privacy-loss composition for subsampled Gaussian mechanisms, and PRISM uses the PRV accountant implemented in Opacus to compute $\varepsilon$ for a target $\delta$ (Gopi et al., 2021; Yousefpour et al., 2022; Opacus Contributors, 2026). All subsequent operations (adaptive post-processing, factorization, alignment, and retraction) are deterministic post-processing and therefore do not weaken DP. $\square$

## A.26. Rank-space moments of isotropic tangent noise

This section derives the rank-space second moments of the isotropic tangent noise used by PRISM (Eq. (19)), which motivates the DP-aware floors in Eq. (26). Let $U \sim \mathcal{N}(0, I_{m \times r})$ and $V \sim \mathcal{N}(0, I_{n \times r})$, and define $\Xi_{A,\ell} = (I - \Pi_{A_\ell}) U \, N^{-1/2}$ and $\Xi_{B,\ell} = V \, M^{-1/2}$ with $M = A_\ell^\top A_\ell$ and $N = B_\ell^\top B_\ell$. Then

$$\mathbb{E}[\Xi_{A,\ell}^\top \Xi_{A,\ell}] = N^{-1/2} \, \mathbb{E}[U^\top (I - \Pi_{A_\ell}) U] \, N^{-1/2}, \qquad \mathbb{E}[\Xi_{B,\ell}^\top \Xi_{B,\ell}] = M^{-1/2} \, \mathbb{E}[V^\top V] \, M^{-1/2}. \tag{49}$$

Since $U$ has i.i.d. standard normal entries and $(I - \Pi_{A_\ell})$ is an orthogonal projector of rank $\mathrm{tr}(I - \Pi_{A_\ell}) = m - r$, we have $\mathbb{E}[U^\top (I - \Pi_{A_\ell}) U] = (m - r) I_r$. Similarly, $\mathbb{E}[V^\top V] = n I_r$. Substituting into (49) yields

$$\mathbb{E}\left[ \frac{\Xi_{A,\ell}^\top \Xi_{A,\ell}}{m} \right] = \frac{m - r}{m} \, N^{-1}, \qquad \mathbb{E}\left[ \frac{\Xi_{B,\ell}^\top \Xi_{B,\ell}}{n} \right] = M^{-1}. \tag{50}$$

Thus the typical eigenvalues of the rank-space noise covariance scale with $M^{-1}$ and $N^{-1}$, explaining why inverse-square-root preconditioning can explode when $M$ or $N$ is ill-conditioned. PRISM's floors in Eq. (26) are gauge invariant because $\mathrm{tr}(M^{-1})$ and $\mathrm{tr}(N^{-1})$ are invariant under $(A_\ell, B_\ell) \mapsto (A_\ell R, B_\ell R^{-\top})$.

## A.27. Adaptive preconditioning and DP noise amplification

This subsection complements Section 3 and proves Theorem 3.5. We also record a simple identity showing how rank-space normalization can "cancel" the DP noise scale when the second moment is dominated by noise.

*Proof of Theorem 3.5.* Since $V \succeq 0$, write its eigendecomposition $V = U\Lambda U^\top$ with $\Lambda = \text{diag}(\lambda_1, \ldots, \lambda_r)$ and $\lambda_i \geq 0$. Then $\mathsf{P} = V + \lambda I = U(\Lambda + \lambda I)U^\top$ and thus $\mathsf{P}^{-1/2} = U(\Lambda + \lambda I)^{-1/2}U^\top$.

$$\|\mathsf{P}^{-1/2}\|_2 = \max_i (\lambda_i + \lambda)^{-1/2} \leq \lambda^{-1/2}.$$

For any $X$, submultiplicativity of the Frobenius norm gives $\|X\mathsf{P}^{-1/2}\|_F \leq \|X\|_F \|\mathsf{P}^{-1/2}\|_2 \leq \lambda^{-1/2}\|X\|_F$. Squaring yields Eq. (27). $\square$

**Proposition A.29** (Noise normalization under naive rank-space preconditioning)**.** *Let $G \in \mathbb{R}^{m \times r}$ have i.i.d. $\mathcal{N}(0, 1)$ entries and define the (uncentered) second moment $V \triangleq \frac{1}{m}G^\top G$. Then the preconditioned matrix $Q \triangleq GV^{-1/2}$ satisfies*

$$Q^\top Q = mI_r \qquad \text{and hence} \qquad \|Q\|_F^2 = mr.$$

*Equivalently, if $\widehat{m} = \tau G$ for any $\tau > 0$ and $V = \frac{1}{m}\widehat{m}^\top \widehat{m}$, then $\widehat{m}V^{-1/2}$ has Frobenius norm $\sqrt{mr}$ independent of $\tau$.*

*Proof.* By definition, $Q^\top Q = V^{-1/2}G^\top G V^{-1/2} = V^{-1/2}(mV)V^{-1/2} = mI_r$. Taking traces gives $\|Q\|_F^2 = \text{tr}(Q^\top Q) = mr$. The final claim follows from $V = \tau^2(\frac{1}{m}G^\top G)$, which implies $\widehat{m}V^{-1/2} = G(\frac{1}{m}G^\top G)^{-1/2} = Q$. $\square$

Proposition A.29 explains the issue in Issue III: when an adaptive method forms $V$ directly from a DP-sanitized gradient whose energy is dominated by DP noise, the right-multiplication by $V^{-1/2}$ can make the stochastic component insensitive to the DP noise scale. PRISM avoids this via (i) DP-aware floors and condition-number clamping (Eq. (26)), which bound $\|V^{-1/2}\|_2$ and prevent ill-conditioned amplification, and (ii) debiasing of the second moment by subtracting the known DP noise covariance in rank space (Appendix A.26).

**Lemma A.30** (Orthogonal gauge equivariance of rank-space preconditioning)**.** *Let $R \in \mathbb{R}^{r \times r}$ be orthogonal and consider the restricted gauge transform $(A_\ell, B_\ell) \mapsto (A_\ell R, B_\ell R)$. If $\widehat{m}_A, \widehat{m}_B$ transform as $\widehat{m}'_A = \widehat{m}_A R$ and $\widehat{m}'_B = \widehat{m}_B R$, and the second moments transform as $V'_A = R^\top V_A R$ and $V'_B = R^\top V_B R$, then the preconditioned directions from Eq. (25) satisfy $U'_A = U_A R$ and $U'_B = U_B R$, and therefore the intrinsic update $U_A B_\ell^\top + A_\ell U_B^\top$ is invariant.*

*Proof.* For orthogonal $R$, similarity equivariance of matrix functions yields $(V'_A + \lambda I)^{-1/2} = (R^\top(V_A + \lambda I)R)^{-1/2} = R^\top(V_A + \lambda I)^{-1/2}R$. Thus $U'_A = \widehat{m}'_A(V'_A + \lambda I)^{-1/2} = \widehat{m}_A R R^\top(V_A + \lambda I)^{-1/2}R = U_A R$, and similarly $U'_B = U_B R$. Finally, with $A'_\ell = A_\ell R$ and $B'_\ell = B_\ell R$ we have

$$\begin{aligned}
U'_A(B'_\ell)^\top + A'_\ell(U'_B)^\top &= (U_A R)(B_\ell R)^\top + (A_\ell R)(U_B R)^\top \\
&= (U_A R)(R^\top B_\ell^\top) + (A_\ell R)(R^\top U_B^\top) \\
&= U_A(RR^\top)B_\ell^\top + A_\ell(RR^\top)U_B^\top \\
&= U_A B_\ell^\top + A_\ell U_B^\top,
\end{aligned}$$

$\square$

# B. Experimental Setup

## B.1. Experimental Details

This section reports dataset split sizes, hyperparameters, and the hardware/software environment used in our experiments.

**Compute.** All experiments were run on a single NVIDIA **A100-PCIE-40GB** GPU.

**Datasets and splits.** Table 6 reports the training sizes we used and evaluation split sizes. GLUE8 is derived from GLUE (Wang et al., 2018) (excluding WNLI), converted to an instruction-format JSON dataset, and sub-sampled with a fixed number of training examples per task. Math-10K is the LLM-Adapters mixture (Hu et al., 2023; AGI-Edgerunners, 2023) and is evaluated on the standard test splits of its component datasets.

**Common fine-tuning hyperparameters.** Unless otherwise stated, all methods share the same backbone, LoRA configuration, and DP settings in Table 7. For DP runs, the noise multiplier is calibrated with Opacus `make_private_with_epsilon` using the default PRV accountant (Gopi et al., 2021; Opacus Contributors, 2026).

*Table 6.* Dataset splits used in our experiments.

| DATASET / SPLIT | TRAIN | EVAL |
|---|---|---|
| **GLUE8** | | |
| CoLA | 1,250 | 1,043 |
| SST-2 | 1,250 | 872 |
| MRPC | 1,250 | 408 |
| STS-B | 1,250 | 1,500 |
| QQP | 1,250 | 40,430 |
| MNLI (MATCHED / MISMATCHED) | 1,250 | 9,815 / 9,832 |
| QNLI | 1,250 | 5,463 |
| RTE | 1,250 | 277 |
| **GLUE8 TOTAL** | 10,000 | – |
| **MATH-10K** | | |
| MATH-10K TRAIN (MIXTURE) | 9,919 | – |
| GSM8K TEST | – | 1,319 |
| AQUA TEST | – | 254 |
| MAWPS TEST | – | 238 |
| SVAMP TEST | – | 1,000 |

*Table 7.* Common hyperparameters shared across methods.

| SETTING | GLUE8 | MATH-10K |
|---|---|---|
| BACKBONE MODEL | GOOGLE/GEMMA-3-4B-PT (GEMMA TEAM ET AL., 2025) | SAME |
| LoRA RANK $r$ | 16 | 16 |
| LoRA SCALING $\alpha_{\text{LoRA}}$ | 16 | 16 |
| LoRA DROPOUT | 0.05 | 0.05 |
| TARGET MODULES | {Q_PROJ,K_PROJ,V_PROJ,UP_PROJ,DOWN_PROJ} | SAME |
| UPDATE STEPS | 500 | 300 |
| EFFECTIVE BATCH SIZE | 64 | 64 |
| MICRO-BATCH SIZE | 4 | 4 |
| MAX SEQUENCE LENGTH | 384 | 256 |
| TRAIN ON INPUTS | FALSE | TRUE |
| RANDOM SEED | 42 | 42 |
| DP BUDGETS | $\varepsilon \in \{3,6\}, \delta = 10^{-5}$ | SAME |
| CLIPPING NORM $C$ | 1.0 | 1.0 |
| DP GRAD-SAMPLE BACKEND | FUNCTORCH (FALLBACK TO HOOKS) | SAME |
| DP ACCOUNTANT | PRV (OPACUS DEFAULT) (GOPI ET AL., 2021; OPACUS CONTRIBUTORS, 2026) | SAME |

## C. Additional Diagnostics and Analysis

### C.1. Additional diagnostics for Issue I

**Diagnostic protocol and hyperparameters.** We evaluate gauge sensitivity by running *DP training* under multiple equivalent LoRA factorizations of the same intrinsic update $Z_\ell = A_\ell B_\ell^\top$. For each gauge $c$, we apply the reparameterization $(A_\ell, B_\ell) \leftarrow (cA_\ell, c^{-1}B_\ell)$, which leaves $Z_\ell$ unchanged but alters factor-space norms. We train for $T = 300$ update steps on Math-10K and log (i) clipping fraction dp_clip_frac, (ii) mean clipping coefficient dp_coef_mean, and (iii) realized intrinsic step magnitude $\|\Delta Z_t\|_F$, where $\Delta Z_t \equiv Z_{t+1} - Z_t$ is computed from the *actual parameter update* (not a formula-level proxy). **Gauges:** $c \in \{0.25, 0.5, 1.0, 2.0, 4.0\}$.

**Metrics and theoretical link.** In baseline factor-space DP-SGD, per-example clipping uses the factor norm $s_i^{\text{fact}} = \sqrt{\|g_{A,i}\|_F^2 + \|g_{B,i}\|_F^2}$ and $\alpha_i = \min\{1, C_{\text{fact}}/s_i^{\text{fact}}\}$ (cf. (3)). Under gauge rescaling, the norm transforms as (4), so $\alpha_i$ and the induced intrinsic update distribution depend on $c$ (Issue-I). PRISM instead forms intrinsic directions via the tangent construction and projectors (e.g., (13), (14)), clips using the intrinsic norm (16), and adds isotropic tangent noise (17); these operations are designed to depend on $Z_\ell$ rather than on a particular factorization, so gauge dependence should be strongly reduced (up to stochastic variability from DP noise).

**Interpretation (Figure 5).** This plot summarizes how the *clipping coefficients* vary across gauges during training. For

*Table 8.* Method-specific hyperparameters.

| METHOD | OPTIMIZER | LR (GLUE8) | LR (MATH) | SUPPLEMENTARY |
|---|---|---|---|---|
| ADAMW | ADAMW (KINGMA & BA, 2015; LOSHCHILOV & HUTTER, 2019) | $2\times10^{-4}$ | $3\times10^{-4}$ | – |
| FFA | ADAMW + FREEZE $A_\ell$ (SUN ET AL., 2024) | $2\times10^{-4}$ | $3\times10^{-4}$ | – |
| RITE | LORA-RITE (YEN ET AL., 2025) | $2\times10^{-4}$ | $3\times10^{-4}$ | – |
| LoRA+ | ADAMW SPLIT LRS (HAYOU ET AL., 2024) | $2\times10^{-4}$ | $3\times10^{-4}$ | RATIO $\rho = 6.0$. |
| LAMB | LAMB (YOU ET AL., 2020) | $5\times10^{-3}$ | $5\times10^{-3}$ | – |
| PRISM | PRISM (OURS) | $2\times10^{-4}$ | $3\times10^{-4}$ | – |

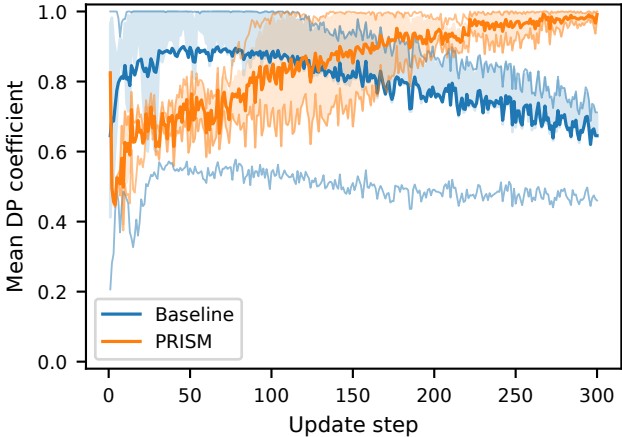

*Figure 5.* Over-time bands of dp_coef_mean across gauges (mean with IQR band; min/max lines).

baseline, the spread between min/max (and the IQR band) remains wide for most of training, indicating that the same DP configuration produces materially different clipping behavior depending on $(A_\ell, B_\ell)$'s gauge. This matches the mechanism in (4): changing $c$ reweights $\|g_{A,i}\|_F$ versus $\|g_{B,i}\|_F$, hence changes $s_i^{\mathrm{fact}}$ and pushes different gauges into different clipping regimes (larger/smaller $\alpha_i$). For PRISM, dp_coef_mean concentrates near 1 after the transient, and the across-gauge dispersion shrinks, consistent with clipping being controlled by the intrinsic norm $s_i$ in (16), which depends on $\Delta Z_{i,\ell}$ rather than on factor scaling. Importantly, the remaining non-zero dispersion is expected in finite runs because DP noise makes $\alpha_i$ and $\Delta Z_\ell$ stochastic, but PRISM's variability is markedly smaller than baseline's.

**Interpretation (Figure 6).** This figure measures the *actual intrinsic update* applied to $Z_\ell$ at each step (computed from $Z_{t+1} - Z_t$), thus directly reflecting the DP perturbation that matters in the intrinsic space. Baseline exhibits a persistent and relatively wide band across gauges: some gauges yield substantially larger $\|\Delta Z_t\|_F$ than others. This is the operational manifestation of Issue-I: once $\alpha_i$ is gauge-dependent via (4), the clipped-and-noised factor update implies a gauge-dependent induced update on $Z_\ell$, so $\|\Delta Z_t\|_F$ cannot be predicted from $Z_\ell$ alone. PRISM shows a transient early phase (optimizer/moment warm-up plus DP stochasticity) and then stabilizes to a smaller, tighter band; this is consistent with intrinsic clipping (16) and tangent noise (17) controlling the intrinsic step directly. The remaining oscillations are natural: even a gauge-invariant *distribution* will yield non-identical single-run trajectories under DP noise, but PRISM suppresses the systematic gauge effect visible in baseline.

**Interpretation (Figure 7).** We compress the multi-gauge experiment into a single diagnostic: $\mathrm{range}_c(\mathrm{dp\_coef\_mean}) = \max_c \mathrm{dp\_coef\_mean}(c) - \min_c \mathrm{dp\_coef\_mean}(c)$. A gauge-invariant DP mechanism should make this range small (up to stochastic fluctuations). Baseline remains high throughout training, directly supporting the theoretical failure mode: factor-space clipping depends on $c$ because of (4), so the average clipping coefficient changes substantially across equivalent parameterizations. PRISM yields a much smaller range after the initial transient, consistent with using the intrinsic norm

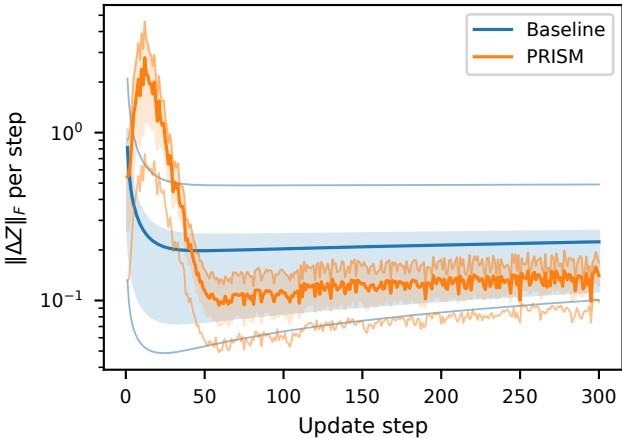

*Figure 6.* Over-time bands of realized intrinsic step magnitude $\|\Delta Z_t\|_F$ across gauges (mean with IQR band; min/max lines).

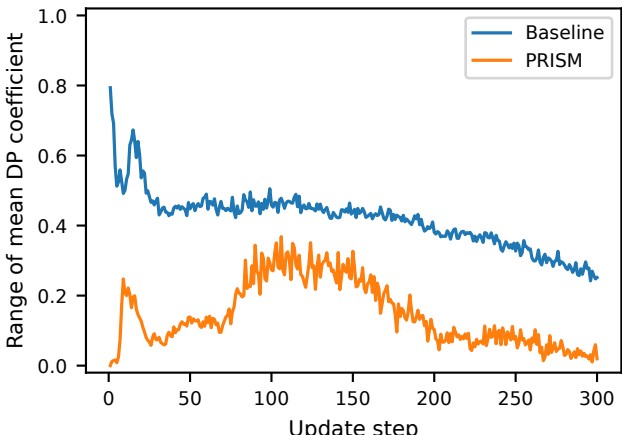

*Figure 7.* Gauge-sensitivity index for clipping: $\mathrm{range}_c(\mathrm{dp\_coef\_mean})$ over time.

(16) and tangent construction ((13), (14)) to decouple DP sensitivity control from gauge.

**Interpretation (Figure 8).** This is the most direct "mechanism check" for Issue-I: the x-axis varies only the gauge $c$, while all intrinsic quantities are initially identical. At step 1, baseline's clipping fraction moves from almost fully clipped (small $c$) to almost never clipped (large $c$), i.e., a qualitative regime change triggered purely by reparameterization. This matches (4): for small $c$ the $c^{-2}\|g_{A,i}\|_F^2$ term can dominate, inflating norms and forcing $\alpha_i \ll 1$; for larger $c$ the norm shrinks and clipping disengages. PRISM is approximately flat across gauges, aligning with intrinsic clipping (16): the clipping decision depends on $\|\Delta Z_{i,\ell}\|_F$ rather than on factor scaling.

**Interpretation (Figure 9).** The mean clipping coefficient $\mathrm{dp\_coef\_mean}$ is a smoother counterpart of Figure 8: it directly measures the average shrinkage induced by DP clipping, $\alpha_i = \min\{1, C/s_i\}$. Baseline increases monotonically with $c$ at step 1, showing that the same DP algorithm injects different effective shrinkage (hence different intrinsic update distributions) purely due to gauge, as predicted by (4). PRISM remains roughly constant across gauges, consistent with controlling sensitivity in intrinsic space (16) and therefore avoiding this reparameterization artifact.

**Interpretation (Figure 10).** By step 300, baseline's $\mathrm{dp\_clip\_frac}$ is close to 1 for all gauges, indicating a *clipping-saturated* regime in factor space: most examples are clipped regardless of $c$. This supports (but also partially limits) diagnostic interpretability: once clipping saturates, the mechanism becomes less sensitive to further changes in the factor norms, so a flatter curve here does *not* imply gauge invariance. In contrast, PRISM shows a near-zero clipping fraction at step 300 for all gauges, suggesting that (under the intrinsic threshold $C_{\mathrm{int}}$) optimization has entered a stable region where intrinsic

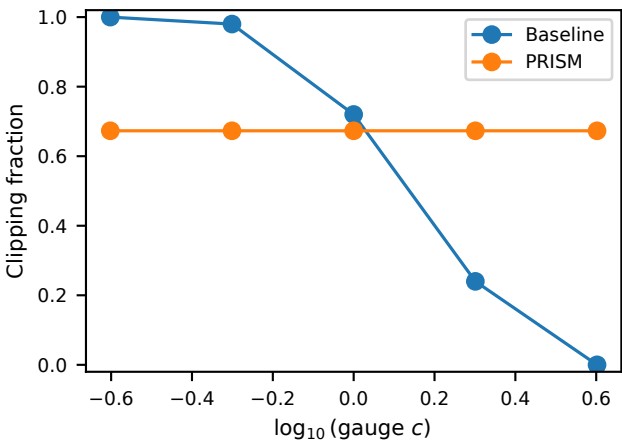

*Figure 8.* Discrete gauge sweep of dp_clip_frac at step 1.

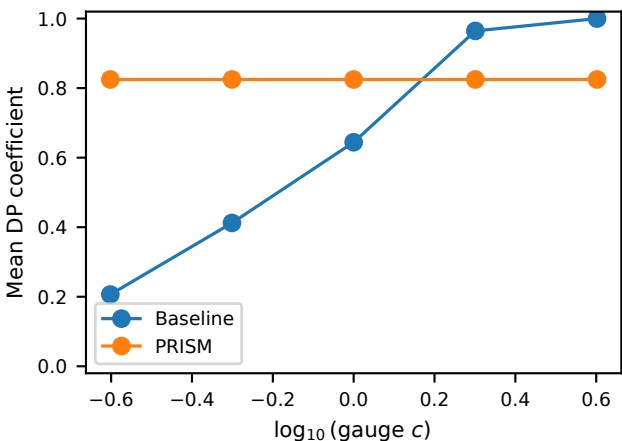

*Figure 9.* Discrete gauge sweep of dp_coef_mean at step 1.

per-example norms mostly lie below the clip bound in (16). Thus, the step-300 sweep is best viewed as confirming that late training can enter a stable/saturated regime, rather than as the primary evidence for Issue-I (which is better captured at step 1 and by Figures 2 and 7).

**Interpretation (Figure 11).** Consistent with Figure 10, baseline's dp_coef_mean still varies with gauge at step 300, but the variability is reduced relative to step 1 because clipping is already heavily engaged for all gauges (many $\alpha_i < 1$). This illustrates a subtle but important point: Issue-I is fundamentally about the *mechanism's dependence on reparameterization* (here seen sharply at step 1), and saturation can mask that dependence by collapsing the algorithm into an always-clipped regime. PRISM's dp_coef_mean concentrates near 1 across gauges at step 300, consistent with intrinsic clipping being mostly inactive and the DP perturbation being governed primarily by tangent noise (17) rather than by gauge-dependent shrinkage. Together with the intrinsic step sensitivity in Figure 2, these late-step diagnostics suggest PRISM's intrinsic control yields a more stable intrinsic update distribution across equivalent factorizations.

### C.2. Additional diagnostics for Issue II

**Gauge sweep protocol.** We snapshot the LoRA layer with the largest $S_t$ at the end of training and sweep the gauge $(A_\ell, B_\ell) \mapsto (cA_\ell, c^{-1}B_\ell)$, which keeps $Z_\ell = A_\ell B_\ell^\top$ fixed. We evaluate $\log_{10} c \in \text{linspace}(-3, 3, 61)$ and draw 64 Monte-Carlo samples per $c$; we plot the median and the 10–90% band.

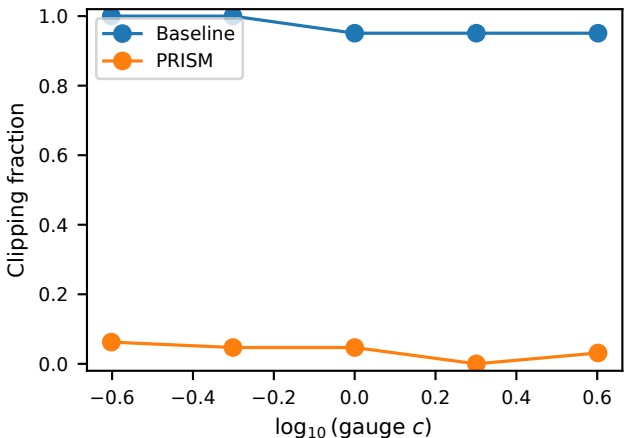

*Figure 10.* Discrete gauge sweep of $\mathrm{dp\_clip\_frac}$ at step 300.

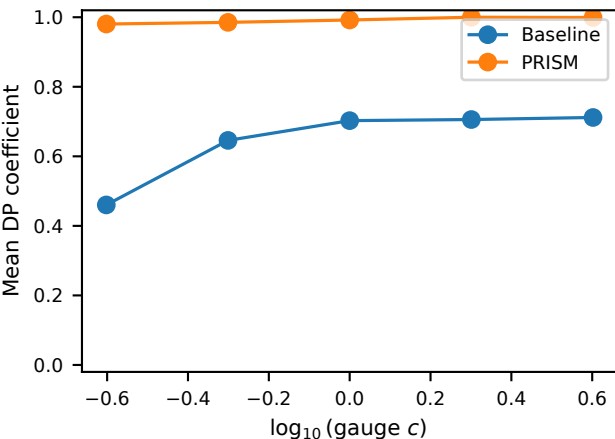

*Figure 11.* Discrete gauge sweep of $\mathrm{dp\_coef\_mean}$ at step 300.

**What Figure 12 tests.** Issue II predicts that factor-space DP can inject a *gauge-dependent* intrinsic noise even when the intrinsic parameter $Z_\ell = A_\ell B_\ell^\top$ (and thus the model function) is held fixed. From Eq. (8), the first-order intrinsic noise satisfies

$$\mathbb{E}\|\xi_{A,\ell} B_\ell^\top + A_\ell \xi_{B,\ell}^\top\|_F^2 = \tau^2\big(m\|B_\ell\|_F^2 + n\|A_\ell\|_F^2\big),$$

so under $(A_\ell, B_\ell) \mapsto (cA_\ell, c^{-1}B_\ell)$ the coefficient becomes $S(c) = mc^{-2}\|B_\ell\|_F^2 + nc^2\|A_\ell\|_F^2$, which is minimized at $c_{\mathrm{th}} = \left(\frac{m\|B_\ell\|_F^2}{n\|A_\ell\|_F^2}\right)^{1/4}$ and diverges as $c \to 0$ or $c \to \infty$. PRISM instead samples isotropic tangent noise $P_{A_\ell, B_\ell}(\Xi_\ell)$, whose distribution depends only on the tangent projector, and whose energy is controlled by the intrinsic dimension (cf. $\mathcal{E}_{Z_\ell} = (\sigma C/b)\sqrt{r(m + n - r)}$ in the main text). Empirically, Figure 12 matches this dichotomy: the baseline (factor-space DP) curve varies by orders of magnitude across $c$ despite fixed $Z_\ell$, while PRISM remains essentially flat up to Monte-Carlo variability. The baseline "bilinear" component (from the $\eta\,\xi_{A,\ell}\xi_{B,\ell}^\top$ term in Eq. (7)) is comparatively small and gauge-invariant, indicating that the dominant instability here comes from the *linear* term's gauge dependence.

**Amplification factors.** Figure 13 converts the sweep into an explicit amplification ratio. Because PRISM's intrinsic noise is gauge-invariant, the ratio inherits the V-shaped dependence of $S(c)$: even benign reparameterizations that leave $Z_\ell$ unchanged can inflate factor-space intrinsic DP noise by large factors. This complements the training-time observation in Figure 1: during optimization, the implicit gauge chosen by the optimizer already yields a consistent $> 1$ amplification, and the controlled gauge sweep shows that, in principle, the same model state (same $Z_\ell$) admits much larger effective intrinsic noise under factor-space DP. Together with Figure 3, these results empirically validate Issue II's core claim: *factor-space*

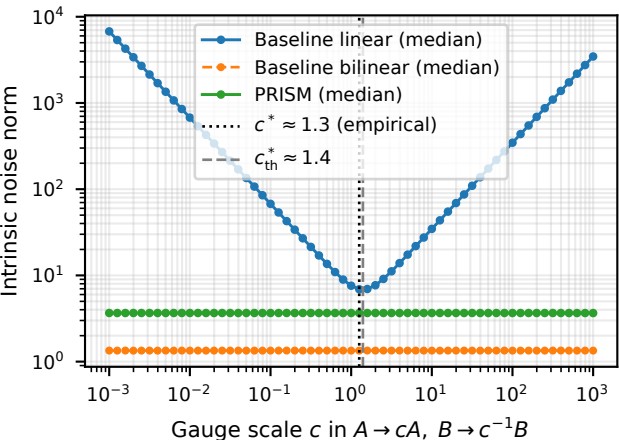

*Figure 12.* Gauge sweep at fixed $Z_\ell$: intrinsic-noise medians with 10–90% band.

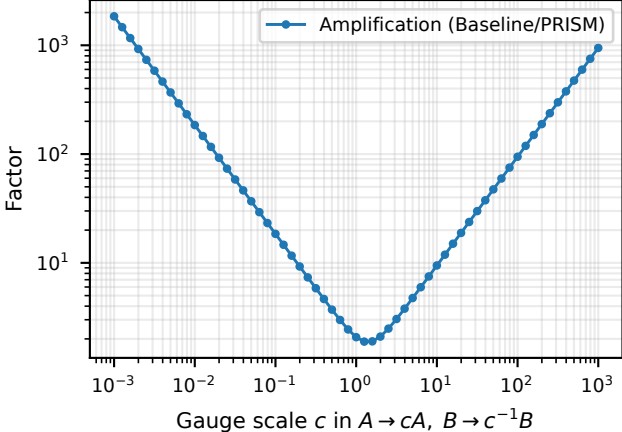

*Figure 13.* Gauge sweep: amplification factor (median baseline / median PRISM) vs. $c$.

*DP produces gauge-dependent, potentially highly amplified intrinsic noise*, while PRISM keeps the intrinsic DP noise scale controlled and gauge-invariant.

### C.3. Additional Issue III diagnostics

**Protocols.** (**Sigma sweep.**) For Figures 4, 14 and 15, we sweep $\epsilon \in \{1.5, 3, 6, 12\}$ at fixed $(C, \delta)$, run 120 optimizer steps, discard the first 10 steps as burn-in, and report means over the remaining steps. The x-axis uses the *realized* noise multiplier $\sigma$ returned by the privacy engine. (**Step-wise diagnostics.**) For Figures 16 to 20, we run 300 steps at $\epsilon = 3$ (hence $\sigma \approx 0.62$ in this setup).

**Analysis (raw noise scaling).** For clipped DP-SGD-style noise, the injected intrinsic noise satisfies $\boldsymbol{\xi}_{\text{intr}} \sim \mathcal{N}\big(0, (\sigma C/b)^2 \mathbf{I}\big)$ (up to the intrinsic parameterization), so $\mathbb{E}\|\boldsymbol{\xi}_{\text{intr}}\|_F$ should grow approximately linearly with $\sigma$ at fixed clipping norm $C$ and batch size $b$. Figure 14 matches this expectation for both methods, serving as a sanity check that (i) the privacy engine responds correctly to the $\epsilon$ sweep and (ii) our measurement pipeline is consistent. Notably, PRISM exhibits a larger *raw* intrinsic noise norm than factor-space DP-AdamW; this is expected because PRISM injects noise directly in the intrinsic update space, whereas factor-space perturbations are first applied to the LoRA factors and then mapped into the intrinsic update, which can reduce the resulting $\|\boldsymbol{\xi}_{\text{intr}}\|_F$ via the low-rank geometry.

**Analysis (noise normalization + reduced amplification).** Define the amplification factor $a(\sigma) \equiv \mathbb{E}\big[\|\mathsf{P}^{-1/2}\boldsymbol{\xi}_{\text{intr}}\|_F / \|\boldsymbol{\xi}_{\text{intr}}\|_F\big]$, which isolates how much the preconditioner scales DP noise in Eq. (10). A key predic-

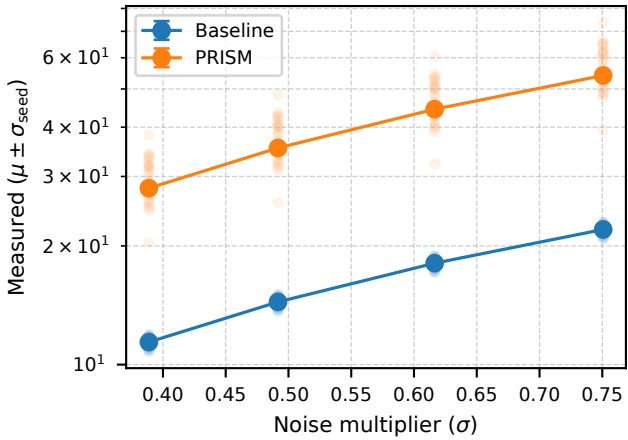

*Figure 14.* Mean raw intrinsic DP noise $\|\boldsymbol{\xi}_{\text{intr}}\|_F$ vs. $\sigma$.

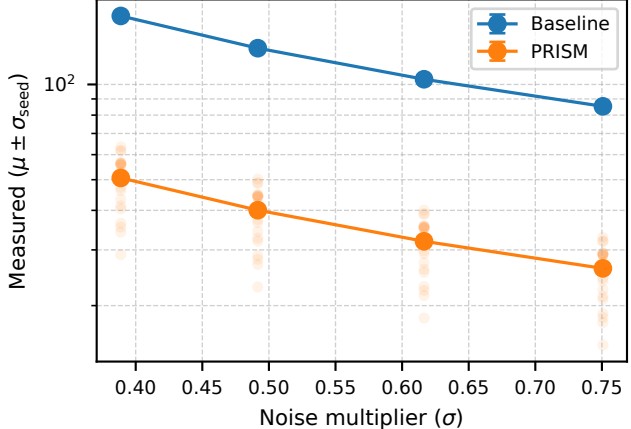

*Figure 15.* Mean amplification $\|\mathrm{P}^{-1/2}\boldsymbol{\xi}\|_F/\|\boldsymbol{\xi}\|_F$ vs. $\sigma$.

tion of Issue III is *noise normalization*: when the second-moment estimator is dominated by DP noise, $\mathbf{V} \propto \sigma^2$ so $(\mathbf{V} + \lambda\mathbf{I})^{-1/2} \propto 1/\sigma$, and the *preconditioned* noise becomes nearly $\sigma$-invariant (Prop. A.29). This is exactly what Figure 4 shows; combining it with the near-linear growth of raw noise in Figure 14 implies $a(\sigma)$ should decrease roughly as $1/\sigma$, which is what Figure 15 exhibits. Crucially, PRISM's amplification is much smaller across $\sigma$ (roughly 18–50 vs. 90–275 for DP-AdamW), supporting the claim that DP-aware floors (Eq. (26)) and the bound in Eq. (27) control worst-case scaling of DP noise under adaptive preconditioning.

**Analysis (why amplification can be large).** Figure 16 plots a "max scaling" proxy for preconditioning strength, roughly corresponding to the largest coordinate-wise scaling (e.g., $\max_i(\widehat{v}_i + \epsilon_{\text{adam}})^{-1/2}$ for Adam-like diagonals, and an operator-norm proxy for low-rank preconditioners). This quantity upper-bounds how much a preconditioner can magnify any input vector, and thus should track amplification of DP noise. DP-AdamW begins with extremely large aggressiveness (orders of magnitude larger than PRISM) and only gradually decays, which is consistent with Issue III: early noisy second-moment estimates can have very small entries/eigenvalues, yielding very large inverse-square-root scaling. PRISM stays in a much tighter range (tens to $\sim 100$), consistent with explicitly enforcing a noise-calibrated floor (Eq. (26)), which prevents the smallest eigenvalues from collapsing and keeps the effective scaling bounded as suggested by Eq. (27).

**Analysis (ill-conditioning in the low-rank core).** Issue III also has a *numerical* face: when preconditioning is implemented through low-rank structure, stability is controlled by the smallest eigenvalues of the relevant Gram/second-moment objects. Let $\mathbf{M}$ denote the (measured) Gram proxy in the low-rank core; then $\|\mathbf{M}^{-1/2}\|_2 = 1/\sqrt{\lambda_{\min}(\mathbf{M})}$, so large values indicate severe ill-conditioning and stress both optimization and numerics. Figure 17 shows DP-AdamW exhibits substantially

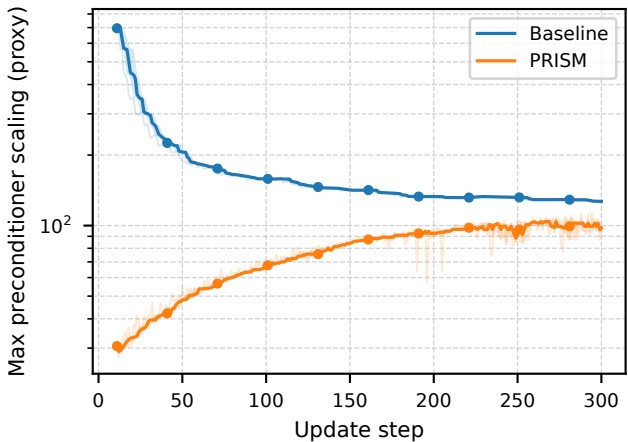

*Figure 16.* Preconditioner aggressiveness (proxy) over training steps.

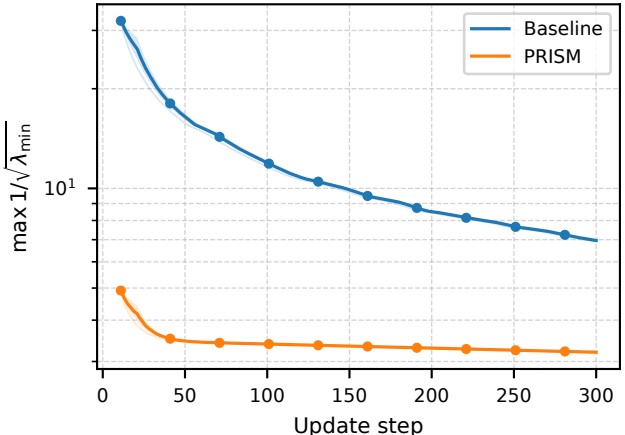

*Figure 17.* Low-rank numerics stress: $\max \|\mathbf{M}^{-1/2}\|_2$ (Gram proxy).

higher stress (large inverse-square-root operator norms) throughout training, whereas PRISM remains in a low-stress regime. This supports the theoretical motivation behind DP-aware floors and condition-number control: by preventing near-singular directions in the low-rank core, PRISM reduces the regimes in which Eq. (27) would otherwise allow very large scaling (small effective $\lambda$).

**Analysis (direct evidence for Issue III and PRISM mitigation).** Figure 18 tracks the realized amplification $\|\mathsf{P}_t^{-1/2}\boldsymbol{\xi}_{t,\mathrm{intr}}\|_F/\|\boldsymbol{\xi}_{t,\mathrm{intr}}\|_F$ during training. Under adaptive preconditioning, this factor can be large when (i) the second-moment (or Gram) has tiny eigenvalues and/or (ii) the preconditioner becomes overly aggressive, precisely the failure mode summarized by Issue III. Empirically, DP-AdamW sits near a large constant amplification ($\sim 10^2$) over the entire run, which explains why its effective update noise (after preconditioning) can remain large even when the raw intrinsic noise is comparatively small (Figure 14). PRISM's amplification is materially smaller (tens rather than hundreds) and evolves smoothly, consistent with a preconditioner whose smallest eigenvalues are protected by noise-aware floors (Eq. (26)) and whose worst-case scaling is constrained in the sense of Eq. (27). Together with Figures 16 and 17, this plot provides direct empirical support that PRISM mitigates the amplification aspect of Issue III.

**Analysis (mechanism: amplification is controlled by scaling).** Figure 19 relates amplification to the max-scaling proxy. In general, for any linear preconditioner $\mathbf{H}_\ell$, $\|\mathbf{H}_\ell\boldsymbol{\xi}\|/\|\boldsymbol{\xi}\|$ concentrates between the singular values of $\mathbf{H}_\ell$; Thus a max-scaling (operator-norm) proxy should strongly correlate with realized amplification. PRISM exhibits this expected monotonic relationship: as the preconditioner becomes more aggressive over training (larger x), the measured amplification (y) rises accordingly, and the color gradient shows this evolution over steps. DP-AdamW, in contrast, occupies a regime with much

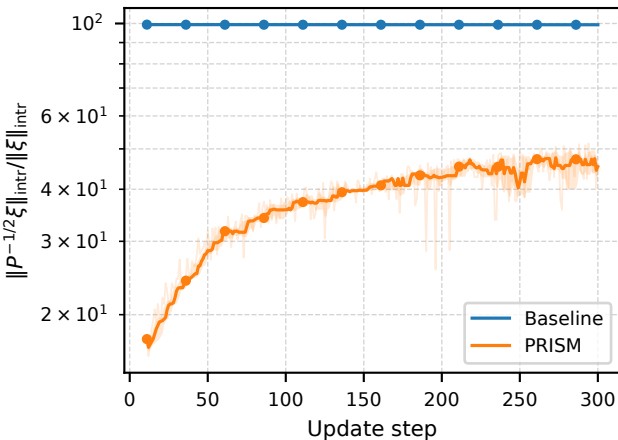

*Figure 18.* Amplification over training steps: $\|\mathrm{P}^{-1/2}\boldsymbol{\xi}\|/\|\boldsymbol{\xi}\|$.

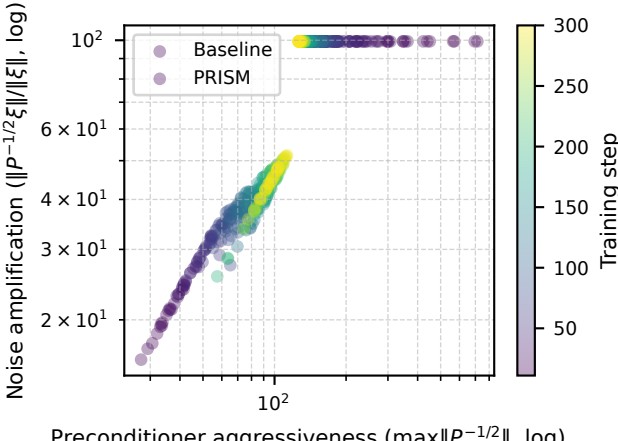

*Figure 19.* Amplification vs. aggressiveness (color = step).

larger aggressiveness yet saturates at a high amplification level, suggesting the run spends most of its time near a hard constraint (e.g., clipping/conditioning caps) rather than smoothly trading off scaling. This plot supports the interpretation that PRISM's improvements are driven by controlling the preconditioner's effective scaling, exactly the control knob targeted by Eq. (26) and bounded by Eq. (27).

**Analysis (mechanism: PRISM de-couples amplification from ill-conditioning).** Figure 20 links amplification to low-rank ill-conditioning via the Gram proxy stress $\|\mathbf{M}^{-1/2}\|_2$. Without safeguards, increasing stress (smaller $\lambda_{\min}(\mathbf{M})$) would typically increase amplification because inverse-square-root operations magnify components in near-null directions. DP-AdamW concentrates in a high-stress regime, consistent with Figure 17, while maintaining a high amplification level. PRISM stays in a low-stress regime and, importantly, shows that when stress grows, amplification does not explode; rather, amplification can even decrease as safeguards activate (floors/clamps effectively reduce the preconditioner's usable gain in ill-conditioned regimes). This is the intended behavior of Issue III mitigation: the algorithm should *avoid* coupling worst-case scaling to unstable low-rank directions, in line with the floor-based control in Eq. (26) and the bound in Eq. (27).

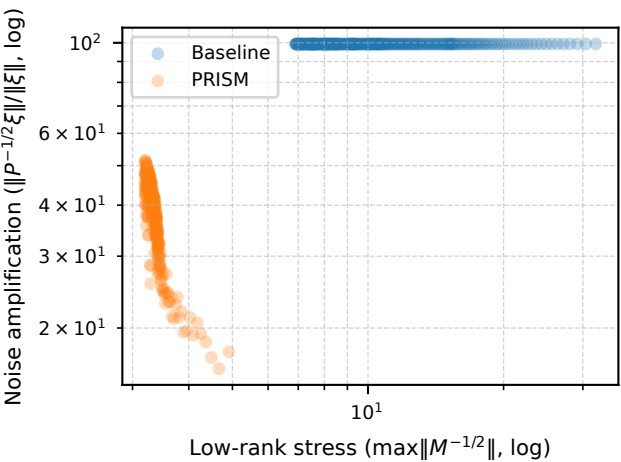

*Figure 20.* Amplification vs. low-rank stress (Gram proxy).

