# OpenReview forum: "PRISM: Gauge-Invariant Tangent-Space Differentially Private LoRA"
_ICML.cc/2026/Conference — ICML 2026 spotlight_

### Official Review · Reviewer_eAaW · 2026-02-26

**Soundness:** 2
**Presentation:** 3
**Significance:** 3
**Originality:** 3
**Overall Recommendation:** 4
**Confidence:** 3

**Summary:**

The paper consider an alternative to differentially private low rank adaptation of LLMs, where the pre-trained weight matrices X are fixed and rank-r matrices A and B are fine-tuned with the layer parameterization change X -> X + AB^T. The proposed method injects noise in the tangent space defined by A and B and then retracts back to rank-r matrices, instead of directly adding noise to the gradient of the loss w.r.t. A and B. It shown mathematically, that the proposed is "gauge invariant" in a sense that both the deterministic model update (the model update independent of DP noise) and the DP noise distributions are invariant to scaling A and B with any rank-r matrix R: A -> AR,
B = BR^{-T}. Notice that this scaling keeps the factor AB^T unchanged however can dramatically change the DP updates in the naive DP-SGD + LORA. Experiments illustrate the effectiveness of the proposed method.

**Compliance With Llm Reviewing Policy:**

Affirmed.

**Final Justification:**

The rebuttal proposed a correction for erroneous Proposition 3.2 and also claimed (and gave experimental results showing) that the proposed method is better than LORA+, a natural baseline for the method of this paper.

**Key Questions For Authors:**

- Can you comment of Prop. 3.2: do you agree with my comment and what would potentially be a fix to this result? How would the proof look like?

- Can you find experiments where this method is clearly better than LORA+? Perhaps high noise regime or situation where you want to avoid expensive hyperparameter tuning (and it's privacy cost) for LORA+ (arguably this method could be more robust) ?

**Limitations:**

yes

**Strengths And Weaknesses:**

Strengths:

- I think this is a nice idea and has clear novelty and potential.
- Geometric interpretation, avoids the potential pitfalls of naive "DP-SGD + LORA" approach.

Weaknesses:

While I think the idea is strong and the method has great potential, I have two major critique points:

1. One results that is central to the story looks wrong to me: Proposition 3.2. First of all, I cannot find a proof for the statement in the paper. Second, I don't think it is true as stated, mainly due to the fact that the matrix $Z + E_T$ is not generally of rank $r$ even if $Z \in \mathcal{M}_r$ and $E_T$ is in the tangent space determined by $Z$. This due to the fact that $E_T$ can have rank up to 2r even if it is in the tangent space of $Z$ (where $Z$ has rank $r$).

2. The experiments are not convincing. The clear competing baseline, LORA+, is also a method that adapts to the geometry by having layer-wise learning rates and avoids ill-conditioning in this heuristic way (that works in practice). Although being a clearly simpler approach, it seems there is no clear overall difference between these two methods in these experiments.

I think some of the math could be moved to the appendix to make the main text easier to read, however I think that is just matter of taste.

My main point is: to me this looks like a strong idea and a strong start, however it looks to me that the story might have to be refined by fixing Prop. 3.2 (the gauge invariance still holds, the DP guarantees hold independent of Prop. 3.2), and also one should find experiments where this more refined geometric methods is benefical compared to LORA+.

---

> ### Author Rebuttal · Authors · 2026-03-31
>
> >W1/Q1: Proposition 3.2
>
> Thank you for pointing this out. It is true that $Z + E_T$ need not stay rank $r$, so Eq. (23) should not compare $\operatorname{Retr}_r(Z + E)$ directly to $Z + E_T$ using an Eckart–Young argument. What PRISM actually needs is only the tangent-direction case used in Eq. (22), where the update is a lifted tangent perturbation $\Delta Z = \Delta A B^\top + A \Delta B^\top$ in $T_Z \mathcal{M}_r$. We will revise Proposition 3.2 by replacing Eq. (23) with
>
> $$
> \lVert \operatorname{Retr}_r(Z - \eta \Delta Z) - (Z - \eta \Delta Z)\rVert_F \le \eta^2 \lVert \Delta A \Delta B^\top\rVert_F
> $$
>
> and therefore
>
> $$
> \operatorname{Retr}_r(Z - \eta \Delta Z) = Z - \eta \Delta Z + O(\eta^2).
> $$
>
> The proof follows directly: setting
>
> $$
> Y_{\eta} = (A - \eta \Delta A)(B - \eta \Delta B)^\top.
> $$
>
> which has rank at most $r$, gives
>
> $$
> Y_{\eta} = Z - \eta \Delta Z + \eta^2 \Delta A \Delta B^\top.
> $$
>
> from which the truncated SVD yields the bound immediately. We will also revise the surrounding text to explicitly scope the first-order exactness claim to the lifted tangent perturbation used by PRISM.
>
> It's worth noting that this change does not affect the main claims. The core mechanism story still comes from the additive intrinsic update in Eq. (22): unlike factor-space noising, it does not create a bilinear factor-product noise term, and the retraction is deterministic post-processing. So Theorem 3.3 (gauge invariance) and Theorem 3.4 (DP guarantee) are unchanged.
>
> >W2/Q2: Comparison with LoRA+
>
> PRISM is fundamentally different from LoRA+: although LoRA+ also adapts to the geometry and helps optimization heuristically via split learning rates, when DP clipping and noise injection are still performed in factor space, it does **not** make the randomized DP mechanism gauge invariant and is vulnerable to gauge-sensitive clipping/noise (Figs. 2, 8–11), gauge-dependent intrinsic noise scaling (Figs. 3, 12–13), and DP-noise amplification under adaptive preconditioning (Figs. 4, 15–20).
>
> To further compare the two methods, we added two more experiments:
>
> (1) We compare PRISM and LoRA+ under different ranks at $r=8,16,32$:
>
> | 4B rank sweep | LoRA+ Avg | PRISM Avg | AdamW Avg |
> |---|---:|---:|---:|
> | r=8 | 0.650 | **0.684** | 0.614 |
> | r=16 | 0.674 | **0.690** | 0.641 |
> | r=32 | 0.653 | **0.682** | 0.636 |
>
> It shows that **PRISM is more robust as rank $r$ changes while LoRA+ is more sensitive to the choice of $r$.** Particularly, PRISM is higher on **11/12 tasks at r=8**, **10/12 tasks at r=16**, and **11/12 tasks at r=32**.
>
>
>
> (2) We compare methods under larger-backbone at Math-10K ($\epsilon=6,\ r=16$), including **Gemma-2-9B (text-only)** and **Gemma-3-12B-pt (from the multimodal Gemma 3 family)**:
>
> | Math-10K, r=16 | GSM8K | AQuA | MAWPS | SVAMP | Avg |
> |---|---:|---:|---:|---:|---:|
> | LoRA+, Gemma-3-4B-pt | 0.446 | 0.409 | 0.786 | 0.611 | 0.563 |
> | **PRISM, Gemma-3-4B-pt** | **0.469** | 0.445 | **0.819** | **0.626** | **0.590** |
> | AdamW, Gemma-3-4B-pt | 0.441 | **0.465** | 0.761 | 0.615 | 0.571 |
> | LoRA+, Gemma-2-9B | 0.6293 | 0.4409 | 0.8067 | 0.6970 | 0.6435 |
> | **PRISM, Gemma-2-9B** | **0.6603** | **0.5197** | **0.8487** | **0.7790** | **0.7019** |
> | AdamW, Gemma-2-9B | 0.6473 | 0.4979 | 0.8093 | 0.7570 | 0.6779 |
> | LoRA+, Gemma-3-12B-pt | 0.6346 | 0.5039 | **0.8193** | 0.7460 | 0.6760 |
> | **PRISM, Gemma-3-12B-pt** | **0.6535** | **0.5315** | **0.8193** | **0.7820** | **0.6966** |
> | AdamW, Gemma-3-12B-pt | 0.5807 | 0.4764 | 0.7311 | 0.6870 | 0.6188 |
>
> It shows that PRISM remains best on Avg on both larger backbones: On the 9B backbone, PRISM is better than LoRA+ on all four Math-10K tasks and improves the average from 0.6435 to 0.7019. On the 12B backbone, PRISM is better on three tasks and tied on MAWPS, improving the average from 0.6760 to 0.6966.
>
> On the high-noise regime, Fig. 4 / Appendix C.3 show that PRISM yields smaller preconditioned DP noise and lower amplification as $\sigma$ increases. We agree that avoiding privacy-costly retuning is a practical advantage of a mechanism-level fix, although we do not yet quantify hyperparameter-search cost explicitly in the current paper.

---

> > ### Author Rebuttal · Reviewer_eAaW · 2026-04-03
> >
> > Thank you for the rebuttal, this addresses my concerns. Please do correct Proposition 3.2 for the final version as proposed.

---

> > > ### Author Response · Authors · 2026-04-06
> > >
> > > Thank you for your time and constructive feedback. We are glad the rebuttal addressed your concerns. We will revise Proposition 3.2 in the final version as discussed in the rebuttal.

---

### Official Review · Reviewer_nRin · 2026-03-11

**Soundness:** 4
**Presentation:** 4
**Significance:** 4
**Originality:** 4
**Overall Recommendation:** 5
**Confidence:** 3

**Summary:**

The paper studies differentially private LoRA fine-tuning, where updates are parameterized as a low-rank matrix $Z = AB^T$. The authors observe that applying DP-SGD directly to the LoRA factors leads to gauge-dependent clipping and noise, because the factorization is non-identifiable, which can cause noise amplification and instability. To address this, they propose PRISM, a DP mechanism that performs gradient clipping and noise injection in the tangent space of the intrinsic update matrix $Z$, ensuring gauge-invariant updates and controlled noise. The paper provides theoretical analysis showing gauge invariance and standard $(\epsilon, \delta)$-DP guarantees, and introduces a DP-aware adaptive update to limit noise amplification. Experiments on GLUE and Math-10K show improved average performance across tasks under differential privacy compared to several baselines.

**Compliance With Llm Reviewing Policy:**

Affirmed.

**Final Justification:**

The rebuttal sufficiently addressed my concerns.

**Key Questions For Authors:**

1. Would it be possible to discuss the computational overhead of PRISM compared to vanilla LoRA?

2. How sensitive is PRISM’s performance in terms of the LoRA rank $r$? How does the privacy-utility trade-offs of PRISM change as $r$ increases?

**Limitations:**

The paper does not explicitly discuss limitations. Potential discussion could include, for example, the additional computational complexity of PRISM, and the relatively limited scale of the current experiments compared to a real-world deployment setting.

**Strengths And Weaknesses:**

Soundness

The submission appears technically sound. The paper clearly formulates the problem of applying differential privacy to LoRA-based fine-tuning and provides a well-motivated analysis of why applying DP-SGD directly to LoRA factors can lead to undesirable behavior due to the non-identifiability of the factorization. The proposed PRISM mechanism is grounded in a principled approach that operates in the intrinsic space of the low-rank update matrix, and the design choices are supported by theoretical analysis. The empirical evaluation on GLUE8 and Math-10K supports the theoretical claims and includes diagnostic experiments that illustrate the identified issues with factor-space DP and how PRISM mitigates them.

Presentation

The paper is well written and clearly structured. The motivation and problem formulation are clearly presented, and the narrative makes it easy to understand why applying DP-SGD directly to LoRA factors leads to undesirable behavior. The paper also does a good job of organizing the technical contributions, first identifying the key issues with factor-space DP and then presenting PRISM as a principled solution. The theoretical analysis, algorithm description, and experimental evaluation are logically connected, which helps the reader follow the development of the method.

Significance
The paper addresses an important and timely problem: how to perform differentially private fine-tuning for large language models using parameter-efficient methods such as LoRA. The paper studies a meaningful problem and provides insights that could be useful for both the differential privacy and large-model fine-tuning communities.

Originality

The paper demonstrates strong originality. It identifies three previously underexplored issues in differentially private LoRA fine-tuning. The paper provides new insights into how this reparameterization freedom can lead to noise amplification and instability, and proposes PRISM, a mechanism that operates in the intrinsic tangent space of the low-rank update to ensure gauge-invariant DP training. This perspective offers a novel way of reasoning about differential privacy in parameter-efficient fine-tuning and is well articulated both theoretically and empirically.

---

> ### Author Rebuttal · Authors · 2026-03-31
>
> >Q1/limitation: Computational overhead
>
> The runtime overhead is expected. Compared with vanilla factor-space LoRA trained with AdamW, PRISM performs additional geometry-aware operations at each step: tangent-space and lifted update construction, intrinsic-norm computation for clipping, low-dimensional tangent-noise sampling, and rank-$r$ retraction. These operations add computation but remain LoRA-scale: the extra geometric cost is $O((m+n)r^2)$ per module, while the tangent-noise sampler avoids dense $O(mn)$ noise generation by instead using a low-dimensional $O((m+n)r)$ construction.
>
> To quantify this, we profiled the main Math-10K / $\epsilon=6$ / $r=16$ setup on a single A100-40GB (10 warmup + 30 measured updates):
>
> | Method | Step time mean (s) | Peak allocated max (MB) |
> |---|---:|---:|
> | LoRA+ | 9.32 | 20961.1 |
> | AdamW | 9.37 | 20961.1 |
> | **PRISM** | **18.64** | **20964.3** |
>
> This is consistent with the theoretical analysis: the main overhead is runtime (~2x in the current implementation), while peak memory is essentially unchanged (+3.2 MB). We will add this discussion and profiling table to the revision.
>
>
> >Q2: Performance & privacy-utility trade-off under varied rank $r$
>
> Following the reviewer's suggestion, we conducted a rank sweep on the 4B backbone ($\epsilon=6$):
>
> | 4B rank sweep | LoRA+ | PRISM | AdamW |
> |---|---:|---:|---:|
> | r=8, GLUE8 Avg | 0.704 | **0.744** | 0.659 |
> | r=8, Math Avg | 0.543 | **0.565** | 0.522 |
> | r=8, Avg | 0.650 | **0.684** | 0.614 |
> | r=16, GLUE8 Avg | 0.730 | **0.740** | 0.676 |
> | r=16, Math Avg | 0.563 | **0.590** | 0.571 |
> | r=16, Avg | 0.674 | **0.690** | 0.641 |
> | r=32, GLUE8 Avg | 0.721 | **0.740** | 0.682 |
> | r=32, Math Avg | 0.516 | **0.566** | 0.542 |
> | r=32, Avg | 0.653 | **0.682** | 0.636 |
>
> PRISM achieves the best performance at every rank, and its advantage over the baselines is stable across $r=8, 16, 32$. PRISM's average varies by only 0.008 across ranks (0.682–0.690), compared to 0.024 for LoRA+ and 0.027 for AdamW, indicating greater robustness to rank selection under DP. Notably, larger $r$ is not guaranteed to improve utility under DP; this is consistent with theorem: while increasing $r$ expands model capacity, it simultaneously enlarges the intrinsic tangent space — whose dimension grows as $r(m+n-r)$ — and therefore the subspace on which clipping and noise operate.
> On the privacy side, Fig. 4 and Appendix C.3 already sweep the noise level $\sigma$ (hence different $\epsilon$ values) and show that, as privacy is tightened, PRISM yields smaller preconditioned DP noise and lower amplification. So while we do not yet have a full $(r,\epsilon)$ task-accuracy grid, the current paper already contains mechanism-level evidence for the privacy--utility trade-off, and the new rank sweep complements it with task-level utility at fixed $\epsilon=6$.
>
>
> >Limitation: limited scale
>
> Following the reviewer's comment, we added larger-backbone models ($\epsilon=6,\ r=16$) on Math-10K, including **Gemma-2-9B (text-only)** and **Gemma-3-12B-pt (from the multimodal Gemma 3 family)**:
>
> | Backbone | LoRA+ Avg | PRISM Avg | AdamW Avg |
> |---|---:|---:|---:|
> | Gemma-3-4B-pt | 0.563 | **0.590** | 0.571 |
> | Gemma-2-9B (text-only) | 0.6435 | **0.7019** | 0.6779 |
> | Gemma-3-12B-pt (multimodal family) | 0.6760 | **0.6966** | 0.6188 |
>
>
> Results show that PRISM can consistently improve across different backbones. We will add the above results to the paper.

---

> > ### Author Rebuttal · Reviewer_nRin · 2026-04-03
> >
> > Thank you very much for the response. I'll keep my score.

---

> > > ### Author Response · Authors · 2026-04-06
> > >
> > > Thank you for your time and constructive feedback. We are glad the rebuttal addressed your concerns.

---

### Official Review · Reviewer_96ni · 2026-03-12

**Soundness:** 3
**Presentation:** 3
**Significance:** 3
**Originality:** 3
**Overall Recommendation:** 4
**Confidence:** 3

**Summary:**

This paper studies differential privacy for parameter-efficient fine-tuning using Low-Rank Adaptation (LoRA). The authors observe that applying DP-SGD directly to the LoRA factors $(A,B)$ is fundamentally misaligned with the intrinsic update $Z = AB^{\top}$ because the factorization is non-identifiable. So factor-space DP can lead to gauge-dependent clipping and noise injection, and also bilinear noise amplification. To solve this issue, the paper proposes PRISM, a gauge-invariant DP mechanism that operates in the intrinsic geometry of low-rank updates. The key idea is to project per-example gradients onto the tangent space of the fixed-rank manifold, apply global intrinsic clipping and isotropic Gaussian noise in this tangent space, and then retract the update back to rank r.

The authors provide theoretical analysis shows that gauge invariance of the induced updates, bounded intrinsic noise energy, and standard $(\varepsilon,\delta)$-DP guarantees via subsampled Gaussian mechanisms. The method uses a gauge-invariant tangent projection, a low-dimensional noise sampler with complexity $O((m+n)r^2)$, and a DP-aware adaptive update rule that stabilizes the preconditioner based on the privacy noise level. This paper evaluate this method on private LoRA fine-tuning for instruction tuning tasks using a Gemma-3 backbone on the GLUE8 and Math-10K benchmarks. The results indicate that it improves robustness and higher average accuracy under privacy constraints compared with several baselines.

**Compliance With Llm Reviewing Policy:**

Affirmed.

**Final Justification:**

I maintain my recommendation of Weak Accept. The authors provided a high-quality rebuttal that addressed all my technical questions.

**Key Questions For Authors:**

1. Could the authors provide empirical measurements of training time or memory overhead compared with standard factor-space DP-LoRA?

2.How does the behavior of PRISM change as the LoRA rank r increases?

3. In practice, do the authors observe numerical instability or rank deficiency during training?

4. The proposed geometric DP mechanism is tied to the rank-r manifold structure of LoRA updates. Could a similar approach be extended to other PEFT methods such as adapters or prefix tuning?

**Limitations:**

Yes. The authors discuss the limitations of their approach and note that the method primarily targets LoRA-based PEFT settings. The paper could further elaborate on practical computational overhead and potential numerical stability issues when operating on low-rank manifolds during training.

**Strengths And Weaknesses:**

Strengths
The core problem identification is valuable. The observation that LoRA's gauge non-identifiability makes factor-space DP fundamentally ill-posed is clearly articulated and formally supported. Corollary 2.3 demonstrating unbounded noise amplification under scalar gauge rescaling is a crisp, and it convince the result that motivates the approach well. The theoretical development is thorough: Theorem 3.1 (isotropic tangent noise with closed-form energy), Theorem 3.3 (gauge invariance), and Theorem 3.4 (DP guarantee) collectively form a complete and rigorous foundation. The low-dimensional noise sampler reducing complexity from O(mn) to $O((m+n)r^2)$ is important. The diagnostic experiments (Figures 1–4, 12–13) directly validate the three identified issues.

Weakness
The experimental evaluation is the main limitation of this work. It is confined to a single backbone (Gemma-3-4B) and two task suites; There should be some more results on larger models, varied ranks, or a broader range of PEFT settings to strengthen the empirical case and demonstrate broader applicability. Furthermore, the non-DP performance of PRISM is noticeably below LoRA-RITE and LoRA+ (0.737 vs. 0.782 average), so that the tangent-space constraint or the adaptive rule introduces optimization overhead beyond what DP requires. This gap also deserves more thorough discussion.

---

> ### Author Rebuttal · Authors · 2026-03-31
>
> >W1/Q2: Evaluation on larger models, varied ranks
>
> Following the reviewer's suggestions, we added two more experiments:
>
> (1) Comparison of different methods under varied ranks ($\epsilon=6$)
>
> | 4B rank sweep | LoRA+ | PRISM | AdamW |
> |---|---:|---:|---:|
> | r=8, GLUE8 Avg | 0.704 | **0.744** | 0.659 |
> | r=8, Math Avg | 0.543 | **0.565** | 0.522 |
> | r=8, Avg | 0.650 | **0.684** | 0.614 |
> | r=16, GLUE8 Avg | 0.730 | **0.740** | 0.676 |
> | r=16, Math Avg | 0.563 | **0.590** | 0.571 |
> | r=16, Avg | 0.674 | **0.690** | 0.641 |
> | r=32, GLUE8 Avg | 0.721 | **0.740** | 0.682 |
> | r=32, Math Avg | 0.516 | **0.566** | 0.542 |
> | r=32, Avg | 0.653 | **0.682** | 0.636 |
>
> Results show PRISM achieves the best average performance across all ranks $r=8, 16, 32$. This is consistent with theory: while increasing $r$ expands model capacity, it simultaneously enlarges the intrinsic tangent space — whose dimension grows as $r(m+n-r)$ — and therefore the subspace on which clipping and noise operate. Larger $r$ is thus not guaranteed to improve DP utility, which is precisely what the non-monotone rank sweep reflects.
>
> (2) Larger-backbone models ($\epsilon=6,\ r=16$) on Math-10K, including **Gemma-2-9B (text-only)** and **Gemma-3-12B-pt (from the multimodal Gemma 3 family)**:
>
> | Backbone | LoRA+ Avg | PRISM Avg | AdamW Avg |
> |---|---:|---:|---:|
> | Gemma-3-4B-pt | 0.563 | **0.590** | 0.571 |
> | Gemma-2-9B | 0.644 | **0.702** | 0.678 |
> | Gemma-3-12B-pt | 0.676 | **0.697** | 0.619 |
>
> Results show that PRISM can consistently improve across different backbones. We will add the above results to the paper.
>
>
>
> >W2: Non-DP performance of PRISM
>
> We emphasize that PRISM is designed to address **DP-specific** issues in the randomized optimization mechanism, not to serve as the strongest non-private optimizer. The challenges it targets — gauge-sensitive clipping and noise injection, gauge-dependent intrinsic noise scaling, and DP-noise amplification under adaptive preconditioning — are unique to the DP setting. This is precisely why the gains are meaningful under DP. Our results are consistent with this design intent: without DP, stronger non-private optimizers can outperform PRISM; under DP, PRISM achieves the best average accuracy at both privacy budgets tested. We will add this clarification to the paper.
>
> >Q1: Training time or memory overhead?
>
> The runtime overhead is expected. Compared with vanilla factor-space LoRA trained with AdamW, PRISM performs additional geometry-aware operations at each step: tangent-space and lifted update construction, intrinsic-norm computation for clipping, low-dimensional tangent-noise sampling, and rank-$r$ retraction. These operations add computation but remain LoRA-scale: the extra geometric cost is $O((m+n)r^2)$ per module, while the tangent-noise sampler avoids dense $O(mn)$ noise generation by instead using a low-dimensional $O((m+n)r)$ construction.
>
> To quantify this, we profiled the main Math-10K / $\epsilon=6$ / $r=16$ setup on a single A100-40GB (10 warmup + 30 measured updates):
>
> | Method | Step time mean (s) | Peak allocated max (MB) |
> |---|---:|---:|
> | LoRA+ | 9.32 | 20961.1 |
> | AdamW | 9.37 | 20961.1 |
> | **PRISM** | **18.64** | **20964.3** |
>
> This is consistent with the theoretical analysis: the main overhead is runtime (~2x in the current implementation), while peak memory is essentially unchanged (+3.2 MB). We will add this discussion and profiling table to the revision.
>
>
>
>
> >Q3: Numerical instability or rank deficiency
>
> What we observe in practice is not hard rank collapse, but **ill-conditioning and DP-noise amplification** in adaptive factor-space baselines. This is already visible in the original diagnostics: Fig. 16 shows much larger preconditioner aggressiveness for the baseline, Fig. 17 shows higher low-rank Gram stress, and Fig. 18 shows persistently larger noise amplification during training. These are precisely the failure modes that the DP-aware floors are designed to control. PRISM itself does not exhibit catastrophic rank deficiency; the practical challenge is amplified noise under ill-conditioned adaptive updates, not numerical singularity in the manifold operations.
>
> >Q4: Extension to other PEFT methods
>
> The geometric recipe should transfer to any PEFT method that admits a redundant parameterization and an intrinsic state on which clipping, isotropic noise, and retraction can be defined. LoRA is the canonical case because its intrinsic object is a fixed-rank matrix with a tractable manifold structure. Adapters introduce a bottleneck parameterization whose intrinsic geometry differs from the rank-r manifold, while prefix tuning operates on a sequence of continuous vectors with no natural low-rank structure; both would require deriving a new intrinsic geometry. We view these as interesting and non-trivial directions for future work.

---

> > ### Author Rebuttal · Reviewer_96ni · 2026-04-03
> >
> > Thanks for your response. I keep my previous evaluation.

---

> > > ### Author Response · Authors · 2026-04-06
> > >
> > > Thank you for your time and constructive feedback. We are glad the rebuttal addressed your concerns.

---

### Official Review · Reviewer_HxgE · 2026-03-12

**Soundness:** 4
**Presentation:** 4
**Significance:** 3
**Originality:** 3
**Overall Recommendation:** 4
**Confidence:** 2

**Summary:**

This paper proposes PRISM, a gauge-invariant differentially private mechanism for LoRA fine-tuning, aiming to solve how to properly apply differential privacy to low-rank factorization. PRISM projects per-example gradients to the tangent space, applies global intrinsic clipping, injects isotropic tangent Gaussian noise, and retracts back to rank r. The paper provides theoretical guarantees showing gauge invariance, (ε,δ)-DP compliance, and bounded noise amplification. Experiments on GLUE8 and Math-10K demonstrate that PRISM achieves the best average performance under DP constraints (ε ∈ {3,6}), particularly on multi-step reasoning tasks.

**Compliance With Llm Reviewing Policy:**

Affirmed.

**Final Justification:**

solved my problems.

**Key Questions For Authors:**

1. **Scalability to larger settings:** Can you provide experiments on larger models (>10B parameters) or larger datasets (>100k examples)? It's crucial to demonstrate that the advantages persist at scale, especially given that current experiments use relatively small setups and only achieve small improvements.
2. **When does gauge-dependence actually matter?** In practice, if one simply runs naive DP-LoRA once with a fixed random initialization, does the gauge-dependence issue cause measurable harm? Or is it primarily a theoretical concern? Given that the improvements is small (which i thought should be more significant since the original methods leads to unbounded noise), i am wondering whether this is a practical problem.
3. **Ablation studies:** Can you isolate the contributions of the three components of PRISM: (i) tangent projection + intrinsic clipping, (ii) DP-aware adaptive floors, (iii) retraction without bilinear noise? Which component contributes most to the improvements? This would help practitioners understand which parts are essential.

**Limitations:**

No. N/A

**Strengths And Weaknesses:**

## Strengths

- **Clear problem formulation and elegant solution.** The paper identifies a genuine and important issue with factor-space DP for LoRA, the direct application eads to fundamentally ill-defined private updates. Then the paper provides an elegant solution that is both mathematically elegant and practically motivated.
- **Rigorous theoretical analysis.** The paper provides comprehensive theoretical characterization.
- **Comprehensive evaluation and nice orgnization.** The paper is well-written and well-organized, all the formats usage make this paper clear to find the key points.

## Weakness

- **Experimental evaluation limited to small-scale settings.** GLUE8 uses only 1,250 training examples per task (10k total); Math-10K has 9,919 examples. These are relatively small for modern LLM fine-tuning. Also, Gemma-3-4b-pt is a 4B parameter model, which is moderate by current standards.
- **Modest improvements in some settings.** While PRISM achieves the best average scores under DP, the absolute gains are often small. In Table2, we can see that the PRISM achieves very small improvement over the second method when it is the best. And in many cases ($\approx$ 1/3), the utility of PRISM is not the best.

---

> ### Author Rebuttal · Authors · 2026-03-31
>
> >W1 & Q1: Experiments on larger model and datasets
>
> We chose GLUE8 (NLU) and Math10K (multi-step numerical reasoning) because they are representative to assess robustness across tasks, covering diverse linguistic phenomena and demanding multi-step computation.  We added additional experiments on larger model ($\epsilon=6,\ r=16$) on Math-10K, including **Gemma-2-9B (text-only)** and **Gemma-3-12B-pt (from the multimodal Gemma 3 family)**:
>
> | Math-10K, r=16 | GSM8K | AQuA | MAWPS | SVAMP | Avg |
> |---|---:|---:|---:|---:|---:|
> | AdamW, Gemma-3-4B-pt | 0.441 | **0.465** | 0.761 | 0.615 | 0.571 |
> | LoRA+, Gemma-3-4B-pt | 0.446 | 0.409 | 0.786 | 0.611 | 0.563 |
> | **PRISM, Gemma-3-4B-pt** | **0.469** | 0.445 | **0.819** | **0.626** | **0.590** |
> | AdamW, Gemma-2-9B | 0.6473 | 0.4979 | 0.8093 | 0.7570 | 0.6779 |
> | LoRA+, Gemma-2-9B | 0.6293 | 0.4409 | 0.8067 | 0.6970 | 0.6435 |
> | **PRISM, Gemma-2-9B** | **0.6603** | **0.5197** | **0.8487** | **0.7790** | **0.7019** |
> | AdamW, Gemma-3-12B-pt | 0.5807 | 0.4764 | 0.7311 | 0.6870 | 0.6188 |
> | LoRA+, Gemma-3-12B-pt | 0.6346 | 0.5039 | **0.8193** | 0.7460 | 0.6760 |
> | **PRISM, Gemma-3-12B-pt** | **0.6535** | **0.5315** | **0.8193** | **0.7820** | **0.6966** |
>
> These results show that for larger models, PRISM is better than LoRA+ on all four tasks and improves the average from 0.6435 to 0.7019. On Gemma-3-12B-pt, PRISM is better on three tasks and tied on MAWPS, improving the average from 0.6760 to 0.6966. We will add these results to the paper.
>
>
> >W2: Modest improvements in some settings
>
> The reason that PRISM does not win across all tasks is because GLUE8 and Math-10K in our paper are trained as **two task suites, not as 12 independent datasets**, i.e., we tune the parameter for each task suite based on the average performance across all tasks, and apply those parameters for all tasks within the same suite. This setup (train on whole suite while evaluating on each task separately) is consistent with the LLM-Adapters protocol  (Hu et al., 2023; AGI-Edgerunners, 2023) and indeed more challenging than tuning and evaluating on different tasks separately, because different tasks may not converge synchronously and DP noise can amplify that mismatch. For this challenging setting, PRISM attains the best performance on majority tasks in private settings.
>
> If compare average performance across all tasks, PRISM can consistently improve across different backbones. Indeed, we can further show that the improvement is **robust** as rank $r$ changes:
>
>
> | 4B rank sweep | GLUE8 Avg | Math Avg | Avg |
> |---|---:|---:|---:|
> | AdamW, r=8 | 0.659 | 0.522 | 0.614 |
> | LoRA+, r=8 | 0.704 | 0.543 | 0.650 |
> | **PRISM, r=8** | **0.744** | **0.565** | **0.684** |
> | AdamW, r=16 | 0.676 | 0.571 | 0.641 |
> | LoRA+, r=16 | 0.730 | 0.563 | 0.674 |
> | **PRISM, r=16** | **0.740** | **0.590** | **0.690** |
> | AdamW, r=32 | 0.682 | 0.542 | 0.636 |
> | LoRA+, r=32 | 0.721 | 0.516 | 0.653 |
> | **PRISM, r=32** | **0.740** | **0.566** | **0.682** |
>
> Results show that PRISM is best on both task suites at $r=8, 16, 32$, with performance variation is much smaller compare to the baselines.
>
>
> >Q2: When does gauge-dependence actually matter?
>
> Gauge-dependence is not merely a theoretical concern, it is a real practical problem. Even a single naive DP-LoRA run can drift into very different effective clipping and noising behavior as factor norms evolve. This is directly evidenced in our experiments: Fig. 1 shows persistent intrinsic-noise amplification under factor-space DP during ordinary training; Fig. 2 shows that realized intrinsic updates remain gauge-sensitive; Fig. 3 shows that measured intrinsic noise tracks the gauge-dependent statistic $S_t$​; and App. Figs. 8–9 show that changing only the gauge alters factor-space DP's clipping behavior materially, while PRISM remains nearly flat.
>
>
>
>
> >Question: Ablation studies
>
> Indeed, we have evaluated the contributions of the three components of PRISM in the paper: Tangent projection + intrinsic clipping target Issue I (Fig. 2; App. Figs. 8–9). Additive intrinsic noising with retraction addresses Issue II (Figs. 1 and 3; App. Figs. 12–13). DP-aware floors target Issue III (Fig. 4; App. Figs. 15–20). We will make this decomposition more explicit in the revision.

---

> > ### Author Rebuttal · Reviewer_HxgE · 2026-04-03
> >
> > Thanks for your response. I have improved my score.

---

> > > ### Author Response · Authors · 2026-04-06
> > >
> > > Thank you for your time and constructive feedback. We are glad the rebuttal addressed your concerns.

---

### Decision · Program_Chairs · 2026-04-30

**Decision:**

Accept (spotlight)

**Comment:**

This paper introduces PRISM, a gauge-invariant differentially private (DP) mechanism for Low-Rank Adaptation (LoRA) that operates in the intrinsic tangent space. This approach prevents the noise amplification and instability that may arise when applying DP-SGD with LoRA.

Reviewers unanimously praised the work for identifying a fundamental flaw in how DP is applied to non-identifiable LoRA factorizations and for providing an elegant, theoretically rigorous solution with strong privacy guarantees. The method effectively mitigates the pitfalls of DP-SGD with LoRA. They specifically highlighted the submission's high technical quality, clear presentation, and significance for the fine-tuning community.

During the discussion, all reviewer concerns were addressed, including those regarding the scalability of the method, computational overhead, and sensitivity to the rank r and the privacy-utility trade-off. One major concern was that Proposition 3.2 appeared to be erroneous, but the authors provided an alternative proof, which satisfied the reviewer who raised this issue.

Because the authors demonstrated that PRISM’s advantages persist across a variety of settings, and given that all reviewers remained satisfied after the revisions, I recommend this paper for acceptance. I ask the authors to correct the proof of Proposition 3.2 and incorporate the additional experiments and discussion into the final manuscript.